# Scaffold with Stochastic Gradients: New Analysis with Linear Speed-Up

**Paul Mangold** [1]   **Alain Durmus** [1]   **Aymeric Dieuleveut** [1]   **Eric Moulines** [1]

## Abstract

This paper proposes a novel analysis for the Scaffold algorithm, a popular method for dealing with data heterogeneity in federated learning. While its convergence in deterministic settings—where local control variates mitigate client drift—is well established, the impact of stochastic gradient updates on its performance is less understood. To address this problem, we first show that its global parameters and control variates define a Markov chain that converges to a stationary distribution in the Wasserstein distance. Leveraging this result, we prove that Scaffold achieves linear speed-up in the number of clients up to higher-order terms in the step size. Nevertheless, our analysis reveals that Scaffold retains a higher-order bias, similar to FedAvg, that does not decrease as the number of clients increases. This highlights opportunities for developing improved stochastic federated learning algorithms.

## 1. Introduction

This paper focuses on the federated optimization, in which $N$ agents collaborate to solve a problem of the form

$$\theta^\star \in \arg\min_{\theta \in \mathbb{R}^d} f(\theta) = \frac{1}{N} \sum_{c=1}^{N} f_{(c)}(\theta) \ , \qquad (1)$$

where for each $c \in \{1, \ldots, N\}$, $f_{(c)}(\theta) = \mathbb{E}[F_{(c)}^{Z_{(c)}}(\theta)]$ is a local risk function of agent $c$ for some function $(z_{(c)}, \theta) \mapsto F_{(c)}^{z_{(c)}}(\theta)$ and local observation $Z_{(c)}$ with distribution $\nu_{(c)}$ over a measurable space $(\mathsf{Z}, \mathcal{Z})$.

One of the most popular methods for solving (1) is FEDAVG (McMahan et al., 2017), where clients perform multiple local stochastic gradient updates, and send their updated parameters to a central server, that aggregates them. Although FEDAVG's local training reduces the number of communications in certain settings, client heterogeneity can significantly hinder its convergence. When the number of local iterations increases, clients lean towards their local minimums, which differ from the global one due to heterogeneity. This phenomenon, called *client drift*, can induce bias in FEDAVG. To control this bias, clients must communicate frequently, requiring at least $\Omega(1/\epsilon)$ communication rounds to reach a mean squared error of $\epsilon^2$ when objective functions are strongly-convex (Karimireddy et al., 2020).

A key method for mitigating client drift is SCAFFOLD (Karimireddy et al., 2020). In this algorithm, each client updates its local model by performing gradient updates, adjusted using local control variates. After each aggregation step, clients update their local control variates based on the global model received from the server, effectively removing heterogeneity bias. SCAFFOLD was first theoretically studied by Karimireddy et al. (2020), reducing communications from $O(1/\epsilon)$ to $O(\log(1/\epsilon))$ for strongly-convex objectives, where $\epsilon > 0$ is a precision target. Later, Mishchenko et al. (2022); Hu & Huang (2023) proved that (a variant of) SCAFFOLD reaches $O(\log(1/\epsilon))$ communication cost with an improved dependence on the problem's condition number. Unfortunately, in all these results, *the number of gradients computed by each client does not decrease with the number of clients.*[1] Yet, a fundamental promise of federated learning is to reduce training cost through collaboration, a phenomenon called linear speed-up (Yu et al., 2019a).

In this paper, we show for the first time, to our knowledge, that *SCAFFOLD achieves linear speed-up.* To this end, we develop a novel point of view on SCAFFOLD, showing that its global iterates and control variates jointly form a Markov chain, similarly to SGD (Dieuleveut et al., 2020) and FE-DAVG (Mangold et al., 2025). For strongly-convex and smooth objectives, we show that this Markov chain converges geometrically to a unique stationary distribution. A careful examination of the pairwise covariances of the global parameters and control variate reveals that, in this stationary distribution, SCAFFOLD's global parameters' variance reduces linearly with the number of clients, up to a maximum number of clients. We then leverage this result to give a new

---

[1]Ecole Polytechnique, CMAP, UMR 7641, France. Correspondence to: Paul Mangold <paul.mangold@polytechnique.edu>.

*Proceedings of the 42$^{nd}$ International Conference on Machine Learning*, Vancouver, Canada. PMLR 267, 2025. Copyright 2025 by the author(s).

[1]We note that, although Karimireddy et al. (2020) obtain such speed-up, they do using a global step size, which significantly departs from common practice. See discussions in Remark 2.1.

Table 1: Communications and local iterations required for SCAFFOLD to reach $\mathbb{E}[\|\theta^t - \theta^\star\|^2] \leq \epsilon^2$, for $\epsilon > 0$, according to multiple analyses of SCAFFOLD with stochastic gradients for $\mu$-strongly convex and $L$-smooth functions.

| | Communication | Local Iterations | Linear Speed-Up | Acceleration[3] Det. | Sto. | General objective |
|---|---|---|---|---|---|---|
| Karimireddy et al. (2020)[1] | $O(\log(1/\epsilon))$ | $O(1/\epsilon^2)$ | ✗[1] | ✗ | ✓ | ✓ |
| Mishchenko et al. (2022)[2] | $O(1/\epsilon)$ | $O(1/\epsilon)$ | ✗ | ✓ | ✗ | ✓ |
| Hu & Huang (2023)[2] | $O(\log(1/\epsilon))$ | $O(1/\epsilon^2)$ | ✗ | ✓ | ✓ | ✓ |
| Mangold et al. (2024) | $O(\log(1/\epsilon))$ | $O(1/N\epsilon^2)$ | ✓ | ✗ | ✓ | ✗[4] |
| **Ours** | $O(\log(1/\epsilon))$ | $O(1/N\epsilon^2)$ | ✓ | ✗ | ✓ | ✓ |

(1) they obtain a linear speed-up by introducing a global step size: in practical implementations, there is no global step size and their analysis loses linear speed-up (see Remark 2.1); (2) based on a stochastic communication scheme; (3) acceleration means that the algorithm benefits from local steps, when gradients are deterministic (Det.), or stochastic (Sto.); (4) only holds for quadratic functions.

non-asymptotic convergence rate for SCAFFOLD, highlighting the speed-up property. Our analytical framework also allows to derive first-order (in the step size) expansions of this covariances, and unveils that, despite its bias-correction mechanism, SCAFFOLD's global iterates still suffer from a small bias. Our contributions are:

- **SCAFFOLD's iterates converge.** The global iterates and control variates of SCAFFOLD form a Markov chain that converges linearly to a stationary distribution in Wasserstein distance, with a faster rate with more local steps.

- **SCAFFOLD has linear speed-up.** We give a new non-asymptotic convergence rate for SCAFFOLD, showing that the number of gradients computed by each client to reach a given precision decreases linearly with the number of clients (up to a limit that we characterize). To our knowledge, this is the first result of this kind for SCAFFOLD; see Table 1 for a comparison with existing works.

- **SCAFFOLD is still biased.** We give first-order expansions, in the step size, of the covariances of SCAFFOLD's iterates in the stationary distribution. Surprisingly, while SCAFFOLD corrects *heterogeneity bias*, it still suffers from another bias due to its stochastic updates.

**Notations.** We denote by $\nabla f$ the gradient of a differentiable function $f : \mathbb{R}^d \to \mathbb{R}$. If $f$ is $i$-times differentiable for $i \geq 1$, we denote its $i$-th derivative by $\nabla^i f$. We use $\langle \cdot, \cdot \rangle$ to denote the Euclidean dot product. Vectors are columns, and their Euclidean norm is $\|\cdot\|$. For matrices, $\|\cdot\|$ is the operator norm, Id is the identity matrix in $\mathbb{R}^d$. For two matrices $A, B$, we define the Kronecker-type linear operator $A \otimes B$ as $A \otimes B : M \mapsto AMB$ where $A, M$, and $B$ have compatible dimensions for multiplication. For a tensor $X$, we denote by $X^{\otimes k}$ its $k$-th tensor power. For a sequence of matrices $M_1, \ldots, M_k$, we define their ordered product as $\prod_{i=1}^{k} M_i = M_k M_{k-1} \cdots M_1$. Let $\mathcal{B}(\mathbb{R}^d)$ be the Borel $\sigma$-algebra of $\mathbb{R}^d$. For two probability measures $\rho_1, \rho_2$ over $\mathcal{X}$ such that $\int \rho_i(\mathrm{dX})\|\mathrm{X}\|_\Lambda^2 < \infty$,

$i = 1, 2$, we define the second-order Wasserstein distance as $\mathbf{W}_2^2(\rho_1, \rho_2) = \inf_{\xi \in \Pi(\rho_1, \rho_2)} \int \|\mathrm{X} - \mathrm{X}'\|_\Lambda^2 \xi(\mathrm{dX}, \mathrm{dX}')$, with $\Pi(\rho_1, \rho_2)$ the set of probability measures on $\mathcal{X} \times \mathcal{X}$ such that $\xi(\mathsf{A} \times \mathcal{X}) = \rho_1(\mathsf{A}), \xi(\mathcal{X} \times \mathsf{A}) = \rho_2(\mathsf{A})$ for $\mathsf{A} \in \mathcal{B}(\mathcal{X})$.

## 2. Federated Learning and SCAFFOLD

The main challenge in federated learning arises from the fact that each client $c \in \{1, \ldots, N\}$ only has access to its own local function $f_{(c)}$, rather than the full sum in (1). Since these functions typically differ across clients, this induces *heterogeneity*, making optimization more complex.

**Assumptions.** Throughout this paper, we consider the following assumptions. The first assumptions A1, A2 and A3 define the regularity of the local objective functions.

*A* 1 (Strong Convexity). For every $c \in \{1, \ldots, N\}$, the function $f_{(c)}$ is twice differentiable and $\mu$-strongly-convex. In particular, we have $\nabla^2 f_{(c)}(\theta) \succcurlyeq \mu \mathrm{Id}$ for any $\theta \in \mathbb{R}^d$.

*A* 2 (Smoothness). For every $c \in \{1, \ldots, N\}$ and $z \in \mathsf{Z}$, the function $F_{(c)}^z$ is twice differentiable and $L$-smooth. In particular, we have $\nabla^2 F_{(c)}^z(\theta) \preccurlyeq L\mathrm{Id}$ for $\theta \in \mathbb{R}^d$.

*A* 3 (Third Derivative). For every $c \in \{1, \ldots, N\}$, $z \in \mathsf{Z}$, the function $f_{(c)}$ is thrice differentiable with bounded third derivative, i.e., there exists $Q \geq 0$ such that for any $u \in \mathbb{R}^d$ and $\theta \in \mathbb{R}^d$, $\|\nabla^3 f_{(c)}(\theta)u^{\otimes 2}\| \leq Q\|u\|^2$.

These assumptions are classical in stochastic optimization (Nesterov, 2013; Dieuleveut & Bach, 2016). We discuss the main consequences of A1 and A2 in Appendix A.1.

To measure heterogeneity of the problem, we rely on the gradients and Hessians of local functions at the solution.

*A* 4 (Heterogeneity Measure). There exist $\zeta_1, \zeta_2 \geq 0$ such that, with $\theta^\star$ as in (1)

$$\frac{1}{N} \sum_{c=1}^{N} \|\nabla^i f_{(c)}(\theta^\star) - \nabla^i f(\theta^\star)\|^2 \leq \zeta_i^2 \quad \text{for } i \in \{1, 2\} \ .$$

Finally, for a parameter $\theta \in \mathbb{R}^d$ and $z \in \mathsf{Z}$, we define the

**Algorithm 1** SCAFFOLD

---

**Input**: initial $\theta^0 \in \mathbb{R}^d$ and $\xi_{(1)}^0, \ldots, \xi_{(N)}^0 \in \mathbb{R}^d$, step size $\gamma > 0$, number of rounds $T > 0$, number of clients $N > 0$, number of local steps $H > 0$

1:  **for** $t = 0$ to $T - 1$ **do**
2:      **for** $c = 1$ to $N$ **do**
3:          Initialize $\theta_{(c)}^{t,0} = \theta^t$
4:          **for** $h = 0$ to $H - 1$ **do**
5:              Receive random state $Z_{(c)}^{t,h+1}$
6:              Set $\theta_{(c)}^{t,h+1} = \theta_{(c)}^{t,h} - \gamma\left\{\nabla F_{(c)}^{Z_{(c)}^{t,h+1}}(\theta_{(c)}^{t,h}) + \xi_{(c)}^t\right\}$
7:          **end for**
8:      **end for**
9:      Update: $\theta^{t+1} = \frac{1}{N}\sum_{c=1}^N \theta_{(c)}^{t,H}$
10:      Update: $\xi_{(c)}^{t+1} = \xi_{(c)}^t + \frac{1}{\gamma H}(\theta_{(c)}^{t,H} - \theta^{t+1})$
11: **end for**
12: **Return:** $\theta^T$

---

stochastic part of the gradient and its covariance as

$$\varepsilon_{(c)}^z(\theta) \triangleq \nabla F_{(c)}^z(\theta) - \nabla f_{(c)}(\theta) \ , \tag{2}$$

$$\mathcal{C}_c(\theta) \triangleq \mathbb{E}\big[\varepsilon_{(c)}^z(\theta)\varepsilon_{(c)}^z(\theta)^\top\big] \ . \tag{3}$$

We assume in A5 that $\varepsilon_{(c)}^z(\theta)$ has bounded sixth moment.

**A5** (Gradient's Variance). There exist constants $\sigma_\star^2, \beta \geq 0$ such that for $\theta \in \mathbb{R}^d$, $p \in \{1, 2, 3\}$, and $c \in \{1, \ldots, N\}$,

$$\mathbb{E}^{1/p}\big[\|\varepsilon_{(c)}^{Z_{(c)}}(\theta)\|^{2p}\big] \leq \sigma_\star^2 + \beta\|\theta - \theta^\star\|^2 \ ,$$

where $Z_{(c)}$ has values in $\mathsf{Z}$ and distribution $\nu_{(c)}$.

**FEDAVG.** A now very popular algorithm to solve (1) is Federated Averaging (FEDAVG) (McMahan et al., 2017). This method leverages local training to reduce communications, by letting each client perform a number of local stochastic gradient updates. Each final iterate of these updates are then sent to a central server, which aggregates the model received by all clients. More precisely, FEDAVG defines a sequence of global iterates $(\vartheta_t)_{t \in \mathbb{N}}$ as follows. At a global time step $t \geq 0$, each client $c \in \{1, \ldots, N\}$ performs $H > 0$ local iterations, starting from $\vartheta_{(c)}^0 = \vartheta_t$, where $\vartheta_t$ is the current global parameter received from the server. This writes as, for $h \in \{0, \ldots, H-1\}$,

$$\vartheta_{(c)}^{t,h+1} = \vartheta_{(c)}^{t,h} - \gamma \nabla F_{(c)}^{Z_{(c)}^{t,h+1}}(\vartheta_{(c)}^{t,h}) \ ,$$

where $\{Z_{(c)}^{t,h+1}\}_{h=1}^H$ are i.i.d. random variables independent among clients and from the previous iterations, with distribution $\nu_{(c)}$. After these local updates, the parameters are aggregated by the server $\vartheta^{t+1} = N^{-1}\sum_{c=1}^N \vartheta_{(c)}^{t,H}$.

**SCAFFOLD.** The SCAFFOLD algorithm (Karimireddy et al., 2020) uses control variates to mitigate *client drift* by replacing the local gradient updates of FEDAVG for $h \in \{0, \ldots, H-1\}$, by

$$\theta_{(c)}^{t,h+1} = \theta_{(c)}^{t,h} - \gamma\Big(\nabla F_{(c)}^{Z_{(c)}^{t,h+1}}(\theta_{(c)}^{t,h}) + \xi_{(c)}^t\Big) \ . \tag{4}$$

These parameters are then aggregated by a central server as in FEDAVG: $\theta^{t+1} = N^{-1}\sum_{c=1}^N \theta_{(c)}^{t,H}$. After aggregation, each client $c$ locally updates its control variate as

$$\xi_{(c)}^{t+1} = \xi_{(c)}^t + \frac{1}{\gamma H}(\theta_{(c)}^{t,H} - \theta^{t+1}) \ . \tag{5}$$

We give the pseudo-code of this algorithm in Algorithm 1. Learning $\xi_{(c)}^t$ corresponds to estimating a linear correction of the gradient of the local functions so that the corrected gradient is zero at $\theta^\star$. The *ideal control variate* for client $c$ is thus $\xi_{(c)}^\star = -\nabla f_{(c)}(\theta^\star)$, as this correction ensures that all clients converge toward the same optimum.

*Remark* 2.1. In this paper, we aim to study the SCAFFOLD algorithm as it is commonly used. Thus, contrarily to (Karimireddy et al., 2020; Yang et al., 2021), we do not consider two-sided step sizes. While this yields the desired linear speed-up by dividing the local step size by $\sqrt{N}$, and increasing the global one, it essentially reduces the algorithm to mini-batch SGD, and does not give much insights on SCAFFOLD itself. Thus, we consider in Table 1 the rate of Karimireddy et al. (2020) without global step size.

## 3. Related Work

**Analysis of FEDAVG.** Early analyses of FEDAVG were conducted under homogeneity assumptions on the gradients (Stich, 2019; Wang & Joshi, 2018; Haddadpour & Mahdavi, 2019; Patel & Dieuleveut, 2019; Yu et al., 2019b; Li et al., 2019b; Woodworth et al., 2020b). Subsequent studies have shown that FEDAVG exhibits a fundamental bias in heterogeneous settings (Li et al., 2019a; Malinovskiy et al., 2020; Charles & Konečný, 2021; Pathak & Wainwright, 2020; Karimireddy et al., 2020): due to client drift, the iterates of FEDAVG do not converge to the true solution $\theta_\star$, but to a biased limit point.

In fact, even in homogeneous settings, FEDAVG remains biased due to its stochastic updates. This appears in the analyses of Khaled et al. (2020); Woodworth et al. (2020a); Glasgow et al. (2022); Wang et al. (2024).

**Heterogeneity mitigation.** Karimireddy et al. (2020) proposed SCAFFOLD, which reduces client drift with control variates, alike variance reduction methods (Schmidt et al., 2017), and proved its convergence. Subsequently, Mitra et al. (2021); Gorbunov et al. (2021) established similar rates in the smooth and strongly convex case. However,

in all these works, the number of communication rounds required to achieve mean squared error of order $\epsilon^2$, scales as $O(\kappa \log(1/\epsilon))$, where $\kappa$ is the problem's condition number.

Mishchenko et al. (2022) then introduced PROXSKIP, which, in a deterministic setting, achieves accelerated communication complexity, reducing the average number of communication rounds to $O(\sqrt{\kappa} \log(\epsilon^{-1}))$ for reaching MSE of order $\epsilon^2$. However, when gradients are stochastic, their analysis requires $O(1/\epsilon)$ rounds. Later on, Hu & Huang (2023) fixed this, reaching $O(\sqrt{\kappa} \log(1/\epsilon))$ rounds even in the stochastic setting. Nonetheless, neither of these analyses achieve linear speed-up with respect to the number of clients. Several extensions of these methods have been proposed (Malinovsky et al., 2022; Condat et al., 2022; Condat & Richtárik, 2022; Sadiev et al., 2022). However, the sample complexity results established in these works do not exhibit linear speed-up either. A notable exception is the work of Mangold et al. (2024), who achieves linear speed-up for SCAFFOLD *for quadratic objectives*; they consider an extended version of SCAFFOLD for linear approximation, named SCAFFLSA, requiring $O(\kappa^2 \log(1/\epsilon))$ communications with a number of local updates scaling in $O(1/N\epsilon^2)$, effectively achieving linear speed-up. In this work, we present a more general analysis that holds beyond the quadratic setting.

**SGD in a Markovian setup.** Unlike SGD with a diminishing step size, which converges to the true optimum under convexity assumptions, constant step-size SGD does not converge pointwise and instead oscillates around $\theta_\star$ (Chee & Toulis, 2018), introducing an inherent bias. To address this problem, Dieuleveut et al. (2020), following a stream of works by (Pflug, 1986; Fort & Pages, 1999; Bach & Moulines, 2013), analyze SGD with a constant step size as a Markov chain, leveraging randomly perturbed dynamical systems to characterize its convergence and limiting behavior. Recently, Mangold et al. (2025) proposed to view FEDAVG's iterates as a Markov chain. They establish that FEDAVG's iterates converge towards a unique stationary distribution, and give explicit first-order expansion of the bias in $O(\gamma H)$. This bias decomposes into two components: one due to heterogeneity, and one due to stochasticity of the local gradients. Remarkably, this second bias vanishes when optimizing quadratic functions.

# 4. New Convergence Rate for SCAFFOLD

In this section, we present our first main theoretical contribution: *SCAFFOLD achieves linear speed-up with respect to the number of agents*. To establish this result, we introduce a new analytical framework for the study of SCAFFOLD.

First, we show in Section 4.1 that the global iterates and control variates of SCAFFOLD define a Markov chain. We then establish that this Markov chain geometrically converges

to a unique stationary distribution in Wasserstein distance. Next, we analyze the covariance structure of this stationary distribution in Section 4.2. The detailed analysis of this covariance matrix provides important insights into the behavior of SCAFFOLD in the stationary regime. Finally, based on these results, we derive a non-asymptotic convergence rate in Section 4.3, proving the linear speed-up for a range of step-sizes and horizons.

## 4.1. Convergence of Global Iterates

**Iterates of SCAFFOLD.** We define the following operators, that generate the iterates of SCAFFOLD. For a value $\theta \in \mathbb{R}^d$, define the local update operator on client $c$ as

$$\mathsf{T}_{(c)}(\theta; \xi_{(c)}, z_{(c)}) = \theta - \gamma \{\nabla F_{(c)}^{z_{(c)}}(\theta) + \xi_{(c)}\} \ ,$$

for $z_{(c)} \in \mathsf{Z}$. Set $\mathsf{T}_{(c)}^0(\theta; \xi_{(c)}, z) = \theta$ and define recursively the local parameter updates

$$\mathsf{T}_{(c)}^{h+1}(\theta; \xi_{(c)}, z_{(c)}^{1:h+1}) = \mathsf{T}_{(c)}(\mathsf{T}_{(c)}^h(\theta; \xi_{(c)}, z_{(c)}^{1:h}); \xi_{(c)}, z_{(c)}^{h+1}) \ ,$$

where $z_{(c)}^{1:h} = [z_{(c)}^1, \dots, z_{(c)}^h]$, for $c \in [N]$ and $h \in [H]$. This allows to define the global update operator

$$\mathsf{T}(\theta; \xi_{(1:N)}, z_{(1:N)}^{1:H}) = \frac{1}{N} \sum_{c=1}^N \mathsf{T}_{(c)}^H(\theta; \xi_{(c)}, z_{(c)}^{1:H}) \ ,$$

Similarly, for $\theta \in \mathbb{R}^d$, we define the operator that updates the control variates as

$$\mathsf{V}_{(c)}(\xi_{(c)}; \theta, z_{(1:N)}^{1:H}) = \xi_{(c)}$$
$$+ \frac{1}{\gamma H}\big(\mathsf{T}_{(c)}^H(\theta; \xi_{(c)}, z_{(c)}^{1:H}) - \mathsf{T}(\theta; \xi_{(1:N)}, z_{(1:N)}^{1:H})\big) \ .$$

Thus, we can define the update of the SCAFFOLD algorithm

$$\mathsf{S}: \big(\theta, \xi_{(1:N)}; z_{(1:N)}^{1:H}\big) \mapsto \big(\mathsf{T}(\theta; \xi_{(1:N)}, z_{(1:N)}^{1:H}),$$
$$\mathsf{V}_{(1)}(\xi_{(1)}; \theta, z_{(1:N)}^{1:H}), \dots, \mathsf{V}_{(N)}(\xi_{(N)}; \theta, z_{(1:N)}^{1:H}))\big) \ .$$

Note that for all $z_{(1:N)}^{1:H}$, $\mathsf{S}(\cdot, z_{(1:N)}^{1:H})$ is a mapping from

$$\mathcal{X} = \{(\mathrm{X}_{(0)}, \dots, \mathrm{X}_{(N)}) \in \mathbb{R}^{(N+1)d} : \textstyle\sum_{c=1}^N \mathrm{X}_{(c)} = 0\} \ ,$$

into itself. We equip $\mathcal{X}$ with the norm $\|\mathrm{X}\|_\Lambda^2 = \langle \mathrm{X}, \Lambda \mathrm{X}\rangle$, where $\Lambda = (\mathrm{Id}, \frac{\gamma^2 H^2}{N}\mathrm{Id}, \dots, \frac{\gamma^2 H^2}{N}\mathrm{Id})$, or more explicitly,

$$\|\mathrm{X}\|_\Lambda^2 = \|\mathrm{X}_{(0)}\|^2 + \frac{\gamma^2 H^2}{N} \sum_{c=1}^N \|\mathrm{X}_{(c)}\|^2 \ . \tag{6}$$

With these notations, the SCAFFOLD updates of the parameters and the control variates—see Algorithm 1—writes

$$\mathrm{X}^{t+1} = \mathsf{S}\big(\mathrm{X}^t; Z_{(1:N)}^{t+1, 1:H}\big) \ , \tag{7}$$

where $\mathrm{X}^t = [\theta^t, \xi_{(1)}^t, \dots, \xi_{(N)}^t]$ and $\{Z_{(1:N)}^{t,1:H}\}_{t \in \mathbb{N}}$ is an i.i.d. sequence with $Z_{(c)}^{t,h} \sim \nu_{(c)}$ for $c \in \{1, \dots, N\}$ and $h \in \{0, \dots, H\}$.

**SCAFFOLD's iterates as a Markov chain.** SCAFFOLD updates form an iterated random function, a specific class of Markov chains that have been extensively studied (see Diaconis & Freedman (1999) and the references therein). The Markov property is clear: given the present state of the $X^t = (\theta^t, \xi^t_{(1)}, \ldots, \xi^t_{(N)})$, the conditional distribution of the future state does not depend on the past. Hence SCAFFOLD's global iterates define a time-homogeneous Markov chain on $\mathcal{X}$ equipped with its Borel $\sigma$-algebra $\mathcal{B}(\mathcal{X})$. We denote by $K_{(\gamma,H)}$ the corresponding Markov kernel on $\mathcal{X}$. We define, for $t \geq 1$, the iterates of $K_{(\gamma,H)}$ as $K^t_{(\gamma,H)}$. For any probability measure $\rho$ on $\mathcal{X}$ and $t \in \mathbb{N}$, the distribution of SCAFFOLD's iterates $X^t$ started from $X^0 \sim \rho$ is $\rho K^t_{(\gamma,H)}$. We show below that the iterates of SCAFFOLD converge to a unique stationary distribution. This requires a contraction in average (see Diaconis & Freedman (1999), Theorem 1): the next lemma shows that $S$ defines a contractive map over $\mathcal{X}$.

**Lemma 4.1.** *Assume A 1 and A 2. Let $Z = Z^{1:H}_{(1:N)}$ be i.i.d. random variables satisfying A 5. Let the step size $\gamma > 0$ and number of local updates $H > 0$ satisfy $\gamma \leq 1/(2L)$ and $\gamma H(L + \mu) \leq 1$. Then, for any $\theta, \theta' \in \mathbb{R}^d$ and $\{\xi_{(c)}, \xi'_{(c)}\}^N_{c=1} \in \mathbb{R}^d$ such that $\sum^N_{c=1} \xi_{(c)} = \sum^N_{c=1} \xi'_{(c)} = 0$, it holds that*

$$\mathbb{E}\left[\|S(X; Z) - S(X'; Z)\|^2_\Lambda\right] \leq \left(1 - \frac{\gamma\mu}{4}\right)^H \|X - X'\|^2_\Lambda , \quad (8)$$

*with $X = (\theta, \xi_{(1)}, \ldots, \xi_{(N)})$, and $X' = (\theta', \xi'_{(1)}, \ldots, \xi'_{(N)})$.*

We prove this lemma in Appendix B.1. A major consequence of this lemma is that SCAFFOLD's iterates and control variates converge to a unique stationary distribution.

**Theorem 4.2.** *Assume A1, A2, and A5. Let $\gamma > 0$, $H > 0$, such that $\gamma \leq 1/(2L)$ and $\gamma H(L + \mu) \leq 1$. Let $\{X^t\}^\infty_{t=0}$, with $X^t = (\theta^t, \xi^t_{(1)}, \ldots, \xi^t_{(N)})$, be SCAFFOLD's iterates with step size $\gamma$ and $H$ local steps and $X^0 \sim \rho$, where $\rho$ is a probability measure on $\mathcal{X}$ such that $\int \|X\|^2_\Lambda \rho(dX) < \infty$. Then, the distribution $\rho K^t_{(\gamma,H)}$ of $X^t$ converges to a unique stationary distribution $\pi_{(\gamma,H)}$ satisfying $\int \|X\|^2_\Lambda \pi_{(\gamma,H)}(dX) < \infty$, and for any $t \in \mathbb{N}$,*

$$\mathbf{W}^2_2(\rho K^t_{(\gamma,H)}, \pi_{(\gamma,H)}) \leq \left(1 - \frac{\gamma\mu}{4}\right)^{Ht} \mathbf{W}^2_2(\rho, \pi_{(\gamma,H)}) .$$

We prove this theorem in Appendix B.1. In the following, we indifferently write $\pi_{(\gamma,H)}(d\theta, d\Xi)$ and $\pi_{(\gamma,H)}(dX)$.

Theorem 4.2 shows that the Markov kernel $K_{(\gamma,H)}$ is geometrically ergodic in 2-Wasserstein distance. Moreover, the distribution of $X^t$ converges to the limiting distribution $\pi_{(\gamma,H)}$ at a linear rate $(1 - \gamma\mu/4)$, with the exponent given by the number of *effective* steps $H \times t$. As with the deterministic algorithm, for a given step size $\gamma$, a larger number of local steps $H$ speeds up the convergence to stationarity. We will show below that it leads to additional bias. Define

the optimal vector $X^\star = (\theta^\star, \xi^\star_{(1)}, \ldots, \xi^\star_{(N)})$, where the optimal control variates are given by $\xi^\star_{(c)} = -\nabla f_{(c)}(\theta^\star)$.

**Lemma 4.3.** *Assume A1, A2, A5. Let $Z = Z^{1:H}_{(1:N)}$ be i.i.d. random variables satisfying A 5. Assume the step size $\gamma$ and the number of local updates $H$ satisfy $\gamma H(L + \mu) \leq 1$. Then, for all $\theta \in \mathbb{R}^d$ and $\{\xi_{(c)}\}^N_{c=1} \subset \mathbb{R}^d$ such that $\sum^N_{c=1} \xi_{(c)} = 0$,*

$$\mathbb{E}\left[\|S(X; Z) - X^\star\|^2_\Lambda\right]$$
$$\leq \left(1 - \frac{\gamma\mu}{4}\right)^H \|X - X^\star\|^2_\Lambda + 2\gamma^2 H \sigma^2_\star ,$$

*with the global iterate vector $X = (\theta, \xi_{(1)}, \ldots, \xi_{(N)})$.*

We prove this lemma in Appendix B.2. Thus, for any $X \in \mathcal{X}$, a single iteration of SCAFFOLD brings $X$ closer to a neighborhood of the optimal solution $X^\star$, as long as $\|X - X^\star\|^2_\Lambda$ is sufficiently large. In Markov chain theory, this implies that $\|X - X^\star\|^2_\Lambda$ serves as a Foster-Lyapunov function for the kernel $K_{(\gamma,H)}$. From this Foster-Lyapunov condition, we may retrieve a first rough bound on the fluctuation of the estimator around $X^\star$.

**Theorem 4.4.** *Assume A1, A2 and A5. Let $\gamma > 0$ be the step size and $H > 0$ the number of local updates. Assume that $\gamma \leq 1/4L$ and $\gamma H(L + \mu) \leq 1$. Then, for any $T > 0$ and any $X^0 \in \mathcal{X}$, the iterates and control variates of SCAFFOLD, $X^T = (\theta^T, \xi^T_{(1)}, \ldots, \xi^T_{(N)})$, satisfy the inequality*

$$\mathbb{E}\left[\|X^T - X^\star\|^2_\Lambda\right]$$
$$\leq \left(1 - \frac{\gamma\mu}{4}\right)^{HT} \|X^0 - X^\star\|^2_\Lambda + \frac{8\gamma}{\mu}\sigma^2_\star ,$$

*where $X^\star$ is the global optimal vector.*

The proof of this theorem is given in Appendix B.2. This preliminary bound is very similar to the ones established in Karimireddy et al. (2020, Lemma 14) and Mishchenko et al. (2022, Theorem 5.5) (for PROXSKIP). We include it for completeness, to underline that a major limitation is that it does not achieve linear speedup in the number of clients. Nonetheless, this result is crucial to bound the higher-order terms that appear in all our subsequent analysis. Indeed, taking $T \to \infty$, a consequence of Theorem 4.2 and Theorem 4.4 is $\int \|X^T - X^\star\|^2_\Lambda \pi^{(\gamma,H)}(dX) \leq 8\gamma\sigma^2_\star/\mu$, which gives the following Corollary.

**Corollary 4.5.** *Assume A1, A2 and A5. Let $Z = Z^{1:H}_{(1:N)}$ be i.i.d. random variables satisfying A5. Let $\gamma > 0$ be the step size and $H > 0$ the number of local updates of SCAFFOLD. Assume that $\gamma \lesssim 1/L$ and $\gamma H(L + \mu) \lesssim 1$. Then, for all $h \in \{0, \ldots, H\}$, it holds that*

$$\int \|\theta - \theta^\star\|^2 \pi_{(\gamma,H)}(d\theta, d\Xi) \leq \frac{8\gamma}{\mu}\sigma^2_\star ,$$

*where $\Xi = (\xi_{(1)}, \ldots, \xi_{(N)}) \in \mathbb{R}^{N \times d}$.*

We give a proof and a more complete version of this Corollary in Appendix B.2, Corollary B.1. We may also obtain a similar bound for local updates and control variates.

**Lemma 4.6.** *Assume A 1, A 2. Let $Z = Z_{(1:N)}^{1:H}$ be i.i.d. random variables satisfying A 5. Assume the step size $\gamma$ and the number of local updates $H$ satisfy $\gamma H(L + \mu) \lesssim 1$. Under these conditions, for any $h \in \{0, \dots, H\}$ and $c \in \{1, \dots, N\}$, it holds that,*

$$\int \mathbb{E}\left[\|\mathsf{T}_{(c)}^h(\theta; \xi_{(c)}, Z_{(c)}^{1:h}) - \theta^\star\|^2\right] \pi_{(\gamma, H)}(\mathrm{d}\theta, \mathrm{d}\Xi) \lesssim \frac{\gamma \sigma_\star^2}{\mu} ,$$

$$\int \|\xi_{(c)} - \xi_{(c)}^\star\|^2 \pi_{(\gamma, H)}(\mathrm{d}\theta, \mathrm{d}\Xi) \lesssim \frac{L\sigma_\star^2}{\mu H} .$$

The proof is postponed to Appendix B.3. We use $\lesssim$ to omit numerical constants, which are provided in the full proof. Notably, for any agent $c$, the variances of *local iterates* after $h \leq H$ local iterations do not scale with $1/N$. However, it is crucial to highlight that the fluctuations of the control variate scale inversely with $H$. We also give derive analog variants of Corollary 4.5 and Lemma 4.6 for moments $2, 4$, and $6$ in Lemma B.3.

## 4.2. Bounding the Variance of the global iterates

We now derive an upper bound on the variance of $\theta - \theta^\star$ under the stationary distribution. In particular, we show that this variance is proportional to $1/N$, up to a higher order term in the step size. To this end, we track the relations between the covariance matrices of the global parameters and control variates, defined for any $c, c' \in [N]$ as

$$\bar{\boldsymbol{\Sigma}}^\theta \triangleq \int (\theta - \theta^\star)^{\otimes 2} \pi_{(\gamma, H)}(\mathrm{d}\theta, \mathrm{d}\Xi) ,$$

$$\bar{\boldsymbol{\Sigma}}_{(c,c')}^\xi \triangleq \int (\xi_{(c)} - \xi_{(c)}^\star)(\xi_{(c')} - \xi_{(c')}^\star)^\top \pi_{(\gamma, H)}(\mathrm{d}\theta, \mathrm{d}\Xi) ,$$

$$\bar{\boldsymbol{\Sigma}}_{(c)}^{\theta,\xi} \triangleq \int (\theta - \theta^\star)(\xi_{(c)} - \xi_{(c)}^\star)^\top \pi_{(\gamma, H)}(\mathrm{d}\theta, \mathrm{d}\Xi) .$$

We emphasize that the parameter and control variates are inherently correlated. Local gradient noise introduced in the updates of the local parameters (4) propagates to the control variates via their update (5). We refer to Lemma 5.1 for a detailed discussion on these covariance matrices. There, we provide exact first-order expansions, offering a precise characterization of their structure and interactions.

Now, we derive an upper bound on the global parameter's covariance $\bar{\boldsymbol{\Sigma}}^\theta$. To this end, define

$$\mathrm{C}^\theta = \|\bar{\boldsymbol{\Sigma}}^\theta\| , \quad \mathrm{C}^{\theta,\xi} = \frac{1}{N} \sum_{c=1}^N \|\bar{\boldsymbol{\Sigma}}_{(c)}^{\theta,\xi}\| ,$$

$$\mathrm{C}^\xi = \frac{1}{N^2} \sum_{c,c'=1}^N \|\bar{\boldsymbol{\Sigma}}_{(c,c')}^\xi\| .$$

We also define the following quantity, related to the variance of noise added by clients during local updates,

$$\varsigma^\epsilon = \frac{1}{N} \sum_{c=1}^N \sum_{h=0}^{H-1} \left\| \int \mathbb{E}[\mathcal{C}_c^h(\theta)] \pi_{(\gamma, H)}(\mathrm{d}\theta, \mathrm{d}\Xi) \right\| ,$$

where $\mathcal{C}_c^h(\theta) = \mathcal{C}_c(\mathsf{T}_{(c)}^h(\theta; \xi_{(c)}, Z_{(c)}^{1:h}))$, and $\mathcal{C}_c(\theta)$ is the covariance of the local gradient noise as defined in (3). The next lemma relates $\mathrm{C}^\theta$, $\mathrm{C}^{\theta,\xi}$ and $\mathrm{C}^\xi$. We present it in a simplified form to highlight the main dependencies.

**Lemma 4.7.** *Assume A 1, A 2, A 5. Assume the step size $\gamma$ and the number of local steps $H$ satisfy $\gamma H(L + \mu) \lesssim 1$, then*

$$\gamma \mu H \mathrm{C}^\theta \lesssim \gamma^2 H^2 L \mathrm{C}^{\theta,\xi} + \gamma^4 H^4 L^2 \mathrm{C}^\xi + \frac{\gamma^2}{N} \varsigma^\epsilon + \mathrm{r}^\theta ,$$

$$\mathrm{C}^{\theta,\xi} \lesssim \zeta_2 \mathrm{C}^\theta + \gamma^3 H^3 L^2 \mathrm{C}^\xi + \frac{\gamma}{NH} \varsigma^\epsilon + \mathrm{r}^{\theta,\xi} ,$$

$$\mathrm{C}^\xi \lesssim \zeta_2^2 \mathrm{C}^\theta + \zeta_2 \gamma H L \mathrm{C}^{\theta,\xi} + \frac{1}{NH^2} \varsigma^\epsilon + \mathrm{r}^\xi ,$$

*where $\zeta_2$ is the heterogeneity coefficient defined in A4, $\mathrm{r}^\theta$, $\mathrm{r}^{\theta,\xi}$, and $\mathrm{r}^\xi$ are higher-order terms.*

We prove this lemma and give exact expressions in Appendix C.3-Lemma C.8. Using these inequalities, we derive the next theorem, that gives an upper bound on $\mathrm{C}^\theta$.

**Theorem 4.8.** *Assume A1, A2, A3, A4, and A5. Furthermore, assume that $\gamma H L \zeta_2 \lesssim \mu$, $\gamma H(L + \mu) \lesssim 1$ and $\gamma \beta \lesssim \mu$. Then, it holds that*

$$\mathrm{C}^\theta \lesssim \frac{\gamma}{N\mu} \sigma_\star^2 + \frac{\gamma^{3/2} Q}{\mu^{5/2}} \sigma_\star^3 + \frac{\gamma^3 H Q^2}{\mu^3} \sigma_\star^4 .$$

We prove this theorem in Appendix C.3. Recall that $Q$ is the upper bound on the third derivative, which is defined in A3 and it vanishes in the quadratic case. We recover in such case the bound on the covariance of the parameter derived in (Mangold et al., 2024). A crucial feature of this result, is that the covariance of the parameters' error $\bar{\boldsymbol{\Sigma}}^\theta$ is proportional to $\gamma/N$, up to higher-order terms in the step size. To our knowledge, this is the first time the variance of SCAFFOLD with general objective function is shown to decrease with the number of clients. It is in stark contrast with existing analyses of SCAFFOLD (Mishchenko et al., 2022) where variance only scales in $\gamma$.

## 4.3. A Non-Asymptotic Rate with Linear Speed-Up

We now state our main result, showing that our bounds from Section 4.2 can be used to obtain non-asymptotic rates for SCAFFOLD. This can be achieved by using the convergence of SCAFFOLD to its stationary distribution through a synchronous coupling method.

**Theorem 4.9.** *Assume A1, A2, A3, A4, and A5. Furthermore, assume that $\gamma H L \zeta_2 \lesssim \mu$, $\gamma H(L + \mu) \lesssim 1$ and $\gamma \beta \lesssim \mu$.*

*Then, the mean squared error of SCAFFOLD's global iterates, initialized with $\theta^0 \in \mathbb{R}^d$ and $\xi_{(1)} = \cdots = \xi_{(N)} = 0 \in \mathbb{R}^d$ is*

$$
\begin{aligned}
&\mathbb{E}\left[\|\theta^T - \theta^\star\|^2\right] \\
&\lesssim \left(1 - \frac{\gamma\mu}{4}\right)^{HT} \left\{2\|\theta^0 - \theta^\star\|^2 + 2\gamma^2 H^2 \zeta_1^2 + \frac{\sigma_\star^2}{L\mu}\right\} \\
&\quad + \frac{\gamma}{N\mu}\sigma_\star^2 + \frac{\gamma^{3/2} Q}{\mu^{5/2}}\sigma_\star^3 + \frac{\gamma^3 H Q^2}{\mu^3}\sigma_\star^4 \ .
\end{aligned}
$$

To prove this theorem, we decompose $\theta^T - \theta^\star = \theta^T - \hat{\theta}^T + \hat{\theta}^T - \theta^\star$, where $\hat{\theta}^T$ is obtained by running SCAFFOLD with the same realization of noise as $\theta^T$ but starting from $\hat{\theta}^0$ in the stationary distribution. We then obtain a bound on the error by bounding $\mathbb{E}[\|\theta^T - \hat{\theta}^T\|^2]$ and $\mathbb{E}[\|\hat{\theta}^T - \theta^\star\|^2]$ separately, using Lemma 4.1 and Theorem 4.8 respectively. We give a detailed proof in Appendix D.

This theorem converts our asymptotic bound on SCAFFOLD's error in the stationary regime into a non-asymptotic bound, *where the variance term scales in $1/N$*, up to higher-order factors in $O(\gamma^{3/2} + \gamma^3 H)$. This gives the following sample and communication complexity for SCAFFOLD.

**Corollary 4.10.** *Let $\epsilon > 0$. With Theorem 4.9's assumptions, we can set $\gamma \lesssim \min(\frac{1}{L}, \frac{N\mu\epsilon^2}{\sigma_\star^2}, \frac{\mu^{5/3}\epsilon^{4/3}}{Q^{2/3}\sigma_\star^2}, \frac{L^{1/2}\mu^{3/2}\epsilon}{Q\sigma_\star^2})$ and $H \lesssim \frac{\sigma_\star^2 \min(1, \mu/\zeta_2)}{L\mu\epsilon^2} \max(\frac{1}{N}, \frac{Q^{2/3}\epsilon^{2/3}}{\mu}, \frac{QL^{1/2}\epsilon}{\mu^{1/2}})$. Then, SCAFFOLD guarantees $\mathbb{E}[\|\theta^T - \theta^\star\|^2] \leq \epsilon^2$ for $T \gtrsim \frac{L}{\mu}\max(1, \zeta_2/\mu)\log(\frac{\|\theta^0 - \theta^\star\|^2 + \zeta_1^2/L^2}{\epsilon^2})$, and the number of stochastic gradients computed by each client is*

$$
TH \lesssim \frac{\sigma_\star^2}{\mu^2\epsilon^2}\max(\frac{1}{N}, \frac{Q^{2/3}\epsilon^{2/3}}{\mu}, \frac{QL^{1/2}\epsilon}{\mu^{1/2}})\log(\frac{\psi_0}{\epsilon^2}) \ ,
$$

*where $\psi_0 = \|\theta^0 - \theta^\star\|^2 + \zeta_1^2/L^2 + \sigma_\star^2/(L\mu)$.*

We prove this corollary in Appendix D. This result combines two crucial features: (i) SCAFFOLD *has linear speed-up* up to a given number of clients: the number of gradients computed by each client scales in $1/N$; and (ii) SCAFFOLD *accelerates stochastic gradients*: the number of rounds required for convergence depends logarithmically on the desired precision $\epsilon$. In comparison, in heterogeneous settings, FEDAVG's number of communication scales polynomially in $1/\epsilon$. To our knowledge, this is the first time that SCAFFOLD is proven to have linear speed-up (without relying on global step sizes), while guaranteeing acceleration with stochastic gradients.

*Remark* 4.11. In our analysis, we show that the number of rounds scales in $\log(1/\epsilon)$, with a multiplicative factor $L/\mu$. Additionally, Hu & Huang (2023) proved that this constant can be reduced to $\sqrt{L/\mu}$, but without linear speed-up in the number of clients. It is an intriguing open question to determine whether SCAFFOLD can preserve this reduction from $L/\mu$ to $\sqrt{L/\mu}$ while guaranteeing this linear speed-up.

# 5. Explicit Expression for Bias and Variance

The analysis framework that we put in place in Section 4 is guided by the study of the covariances of the global parameters and control variates of SCAFFOLD. We now provide novel insights on the behaviour of SCAFFOLD in the stationary regime. In Section 5.1, we give exact first-order (in the step size) expression for the covariance matrices defined in Section 4.2. Surprisingly, this study uncovers that SCAFFOLD's global parameters *are still biased*, and we describe this bias in Section 5.2.

## 5.1. Variance of the Global Iterates

In SCAFFOLD, the only source of randomness comes from the stochasticity of the gradient updates. These stochastic updates then propagate in the global iterates and control variates of the algorithm. Our analysis framework allows us to give the following expressions of these covariances, as a function of the gradient's covariance at the solution $\theta^\star$.

**Lemma 5.1.** *Assume A1, A2, A3, A4, A5. Furthermore, assume that the step size $\gamma$ and number of local updates $H$ satisfy $\gamma H L \zeta_2 \lesssim \mu$ and $\gamma H(L + \mu) \lesssim 1$ and $\gamma\beta \lesssim \mu$. Then, it holds that, for $c \neq c' \in \{1, \ldots, N\}$,*

$$
\bar{\Sigma}^\theta = \frac{\gamma}{N}\mathbf{A}\mathcal{C}(\theta^\star) + O(\gamma^2 H + \gamma^{3/2}) \ ,
$$

$$
\begin{aligned}
\bar{\Sigma}^{\theta,\xi}_{(c)} &= \frac{\gamma}{N}\mathbf{A}\mathcal{C}(\theta^\star)(\nabla^2 f_{(c)}(\theta^\star) - \nabla^2 f(\theta^\star)) \\
&\quad + \frac{\gamma}{N}(\mathcal{C}_c(\theta^\star) - \mathcal{C}(\theta^\star)) + O(\gamma^2 H + \gamma^{3/2}) \ ,
\end{aligned}
$$

$$
\bar{\Sigma}^\xi_{(c,c)} = \left(1 - \frac{2}{N}\right)\frac{1}{H}\mathcal{C}_c(\theta^\star) + \frac{1}{NH}\mathcal{C}(\theta^\star) + O(\gamma) \ ,
$$

$$
\bar{\Sigma}^\xi_{(c,c')} = \frac{1}{NH}(\mathcal{C}(\theta^\star) - \mathcal{C}_c(\theta^\star) - \mathcal{C}_{c'}(\theta^\star)) + O(\gamma) \ ,
$$

*where $\mathbf{A} = \left(\mathrm{Id} \otimes \nabla^2 f(\theta^\star) + \nabla^2 f(\theta^\star) \otimes \mathrm{Id}\right)^{-1}$, $\mathcal{C}_c(\theta^\star) = \mathbb{E}[(\varepsilon^{Z_{(c)}}_{(c)}(\theta^\star))^{\otimes 2}]$ and $\mathcal{C}(\theta^\star) = \frac{1}{N}\sum_{c=1}^N \mathcal{C}_c(\theta^\star)$.*

We prove this lemma in Appendix E.2. This result confirms our finding that, in the stationary regime of SCAFFOLD, the covariances $\bar{\Sigma}^\theta$ and $\bar{\Sigma}^{\theta,\xi}_{(c)}$ both scale in $\gamma/N$. However, this is not the case for the control variates, which do not even scale in the step size $\gamma$. More remarkably, we show that, for any $c$, the covariance $\bar{\Sigma}^\xi_{(c,c)}$ of $\xi_{(c)}$ does not decrease in $1/N$. Fortunately, the covariances of pairs of distinct control variates recovers this $1/N$, which is the reason why SCAFFOLD enjoys linear speed-up.

*Remark* 5.2. We note that, in the analysis of PROXSKIP (Mishchenko et al., 2022), they use a Lyapunov function similar to (6), based on the average of the $\gamma^2 H^2 \|\xi_{(c)} - \xi^\star_{(c)}\|^2$. Lemma 5.1 shows that this Lyapunov function *cannot achieve linear speed-up*, as its terms only scale in $O(\gamma^2 H)$.

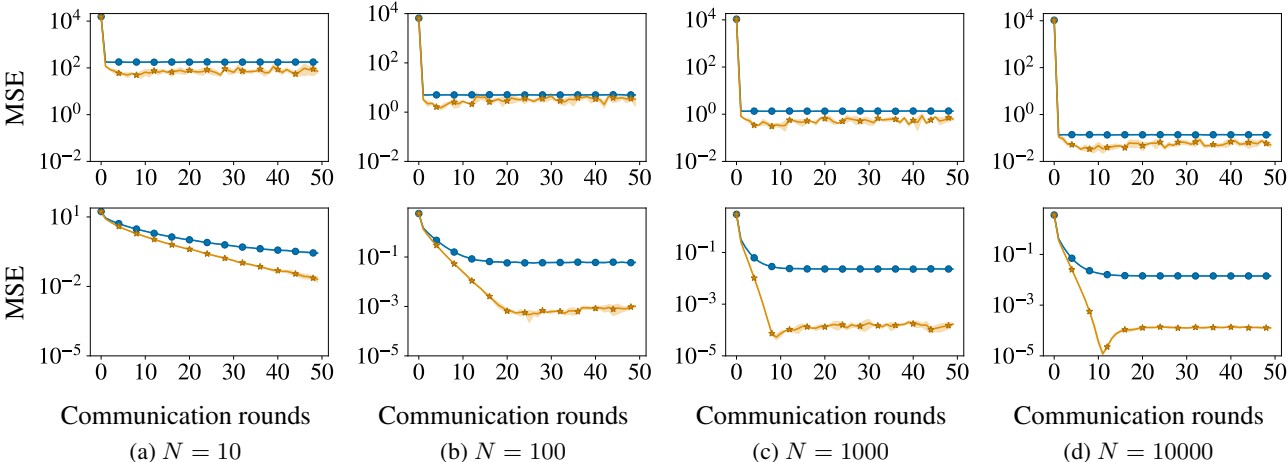

Figure 1: Mean squared error $\mathbb{E}[\|\theta^t - \theta^\star\|^2]$ as a function of the number of communications, with $H = 100$ and $\gamma = 0.05$, for linear regression (top row) and logistic regression (bottom row) problems. For each curve, we plot the average over 3 runs and the standard deviation.

## 5.2. Non-Vanishing Bias of SCAFFOLD

Quite surprisingly, our analysis highlights that SCAFFOLD is still biased. We now give an expression of SCAFFOLD's bias, i.e., the expected error in the stationary distribution

$$\bar{\boldsymbol{b}}^\theta \triangleq \int (\theta - \theta^\star)\pi_{(\gamma,H)}(\mathrm{d}\theta, \mathrm{d}\Xi) \ . \tag{9}$$

We require the fourth derivative of $f_{(c)}$ to be bounded.

**A6 (Fourth Derivative).** For $c \in \{1, \dots, N\}$, the function $f_{(c)}$ is 4 times differentiable and satisfies, for any $\theta \in \mathbb{R}^d$ and $u \in \mathbb{R}^d$, $\|\nabla^4 f_{(c)}(\theta)u^{\otimes 3}\| \leq G\|u\|^3$.

Given this assumption, we obtain the following theorem.

**Theorem 5.3.** *Assume A1, A2, A3, A4, A5, A6. Furthermore, assume that the step size $\gamma$ and number of local updates $H$ satisfy $\gamma(H-1)L\zeta_2 \lesssim \mu$ and $\gamma H(L+\mu) \lesssim 1$ and $\gamma\beta \lesssim \mu$. Then, the bias of SCAFFOLD is*

$$\bar{\boldsymbol{b}}^\theta = -\frac{\gamma}{2N}\nabla^2 f(\theta^\star)^{-1}\nabla^3 f(\theta^\star)\mathbf{A}\mathcal{C}(\theta^\star) + O(\gamma^2 H + \gamma^{3/2}) \ .$$

We refer to Appendix E.3 for a proof of this theorem. Even though SCAFFOLD eliminates heterogeneity bias, its global iterates remain biased. This bias scales with $\gamma/N$ times the local gradient's variance. It is not due to heterogeneity, but solely to the stochasticity of the local updates. In fact, we even recognize the bias of FEDAVG with homogeneous functions, as presented in Mangold et al. (2025)'s Theorem 3. We note that this bias scales with the local gradients' covariances, suggesting that SCAFFOLD may not be appropriate in problems with very noisy gradients.

## 6. Numerical Results

**Experimental setup.** We illustrate our theoretical findings on $\ell_2$ regularized linear and logistic regression. For linear regression, we use make_regression function from scikit-learn (Pedregosa et al., 2011) to generate two different datasets with $100N$ records and 20 features; to simulate heterogeneity, we use different seeds and n_informative=2 and n_informative=10 respectively. The first dataset is split evenly among the first $N/2$ clients, while the second one is split evenly across the other half of clients. For logistic regression, we repeat the same procedure with the make_classification function with two different seeds. Using this procedure, we generate a regression and a classification task, where each client has 200 records, and where the distribution is heterogeneous. In both settings, we run SCAFFOLD with $\gamma = 0.05$ and $H = 100$, $T = 100$ and $N \in \{10, 100, 1000, 10000\}$. We estimate the gradients using batches of size 10, and compare the result with FEDAVG with the same parameters. The code is available online at https://github.com/pmangold/scaffold-speed-up.

**SCAFFOLD has linear speed-up.** For each value of $N$, we run both SCAFFOLD and FEDAVG and report the results in Figure 1. As expected, SCAFFOLD consistently outperforms FEDAVG in all settings. In conformity with our theory, SCAFFOLD benefits from the presence of more clients: as the number of clients increases, the error in stationary regime decrease, both in linear (top row) and logistic (bottom row) regression.

**Linear speed-up with many clients.** Remarkably, the linear speed-up remains for number of clients gets large (up to $1,000$), suggesting that the condition on the maximal

number of clients until which the linear speed-up holds in Corollary 4.10 is not overly restrictive. Nonetheless, there is no more improvement from $N = 1,000$ to $N = 10,000$ in our logistic regression problem (bottom row): this suggest we have reached saturation, and that in this setting, increasing the number of clients does not help beyond this point. As predicted by our theory, this is not the case in linear regression (top row). Indeed, in this case, the loss function is quadratic (i.e., $Q = 0$) and the limit on the number of clients stated in Corollary 4.10 is thus infinite.

## 7. Conclusion

In this paper, we provide a novel analytical framework for the SCAFFOLD algorithm. We show that its global iterates and control variates define a Markov chain, that converges to a stationary distribution. This key property allows us to derive the first rate which shows that *SCAFFOLD achieves linear speed-up in the number of clients*. Our analysis is based on a careful examination of the covariance of SCAFFOLD's global iterates and covariance, finely tracking the propagation of noise through the algorithm's parameters.

Although our work provide novel insights on the behavior of SCAFFOLD, many questions remain open. In particular, it is yet to be understood whether SCAFFOLD can enjoy "deterministic" accelerated communication complexity as in Mishchenko et al. (2022); Hu & Huang (2023)'s analyses while preserving the desired linear speed-up. Finally, our analysis highlights that SCAFFOLD's iterates are still biased: designing novel methods that remove this residual bias is a promising direction for the development of novel stochastic federated learning methods.

### Acknowledgements

The work of P. Mangold has been supported by Technology Innovation Institute (TII), project Fed2Learn. The work of A. Dieuleveut is supported by Hi!Paris FLAG chair, and this work has benefited from French State aid managed by the Agence Nationale de la Recherche (ANR) under France 2030 program with the reference ANR-23-PEIA-005 (RE-DEEM project). The work of E. Moulines has been partly funded by the European Union (ERC-2022-SYG-OCEAN-101071601). Views and opinions expressed are however those of the author(s) only and do not necessarily reflect those of the European Union or the European Research Council Executive Agency. Neither the European Union nor the granting authority can be held responsible for them.

### Impact Statement

This paper presents work whose goal is to advance the field of Machine Learning. There are many potential societal consequences of our work, none which we feel must be specifically highlighted here.

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

# A. Preliminaries

## A.1. Strong convexity and Smoothness

We list here the inequalities that are consequences of strong convexity (A1) and smoothness (A2) of the functions that we minimize in (1). For $c \in \{1, \ldots, N\}$ and $Z_{(c)} \sim \nu_{(c)}$, A1 and A2 imply that, for any $\theta, \theta' \in \mathbb{R}^d$,

$$\mathbb{E}\left[\|\nabla F_{(c)}^{Z_{(c)}}(\theta) - \nabla F_{(c)}^{Z_{(c)}}(\theta')\|^2\right] \leq L\langle \nabla f_{(c)}(\theta) - \nabla f_{(c)}(\theta'), \theta - \theta'\rangle . \tag{10}$$

This inequality is generally referred to as co-coercivity of the gradient of $f_{(c)}$, and is proven in Theorem 2.1.5 of Nesterov (2013). Assumptions A1 and A2 also imply that, for any $\theta, \theta' \in \mathbb{R}^d$,

$$-\langle \nabla f_{(c)}(\theta) - \nabla f_{(c)}(\theta'), \theta - \theta'\rangle \leq -\mu\|\theta - \theta'\|^2 . \tag{11}$$

This second inequality is generally referred to as monotonicity of the gradient of $f_{(c)}$. Finally, smoothness of $F_{(c)}^{z_{(c)}}$, for $z_{(c)} \in \mathsf{Z}$ (A2), means that the gradient of $F_{(c)}^z$ is Lipschitz, i.e., for any $\theta, \theta' \in \mathbb{R}^d$,

$$\|\nabla F_{(c)}^{z_{(c)}}(\theta) - \nabla F_{(c)}^{z_{(c)}}(\theta')\| \leq L\|\nabla F_{(c)}^{z_{(c)}}(\theta) - \nabla F_{(c)}^{z_{(c)}}(\theta')\| . \tag{12}$$

## A.2. Iterate Operators.

We recall the operators defined in Section 4.1, that generate the local and global updates of SCAFFOLD. For $c \in \{1, \ldots, N\}$, $\theta \in \mathbb{R}^d$, $\xi_{(c)} \in \mathbb{R}^d$ and $z_{(c)} \in \mathsf{Z}$ define

$$\mathsf{T}_{(c)}(\theta; z_{(c)}, \xi_{(c)}) = \theta - \gamma\left\{\nabla F_{(c)}^{z_{(c)}}(\theta) + \xi_{(c)}\right\} ,$$

Then, set $\mathsf{T}_{(c)}^0(\theta; \xi_{(c)}, z_{(c)}) = \theta$ and define recursively for $z_{(c)}^{1:h+1} = (z_{(c)}^1, \ldots, z_{(c)}^{h+1}) \in \mathsf{Z}^{h+1}$,

$$\mathsf{T}_{(c)}^{h+1}(\theta; \xi_{(c)}, z_{(c)}^{1:h+1}) = \mathsf{T}_{(c)}(\mathsf{T}_{(c)}^h(\theta; \xi_{(c)}, z_{(c)}^{1:h}); \xi_{(c)}, z_{(c)}^{h+1}) .$$

This allows to define the global update operator, denoting $\xi_{(1:N)} = (\xi_{(1)}, \ldots, \xi_{(N)})$ and $z_{(1:N)}^{1:H} = (z_{(1)}^{1:H}, \ldots, z_{(N)}^{1:H})$

$$\mathsf{T}(\theta; \xi_{(1:N)}, z_{(1:N)}^{1:H}) = \frac{1}{N}\sum_{c=1}^{N} \mathsf{T}_{(c)}^H(\theta; \xi_{(c)}, z_{(c)}^{1:H}) .$$

Similarly, we define the operator that updates the control variates, for $c \in \{1, \ldots, N\}$, as

$$\mathsf{V}_{(c)}(\xi_{(c)}; \theta, z_{(1:N)}^{1:H}) = \xi_{(c)} + \frac{1}{\gamma H}\left(\mathsf{T}_{(c)}^H(\theta; \xi_{(c)}, z_{(c)}^{1:H}) - \mathsf{T}(\theta; \xi_{(1:N)}, z_{(1:N)}^{1:H})\right) .$$

Thus, we can define the update of the SCAFFOLD algorithm with noise $z_{(1:N)}^{1:H}$ as

$$\mathsf{S} : \left(\theta, \xi_{(1)}, \ldots, \xi_{(N)}; z_{(1:N)}^{1:H}\right) \mapsto \left(\mathsf{T}(\theta; \xi_{(1:N)}, Z), \mathsf{V}_{(1)}(\xi_{(1)}; \theta, z_{(1:N)}^{1:H}), \ldots, \mathsf{V}_{(N)}(\xi_{(N)}; \theta, z_{(1:N)}^{1:H})\right) .$$

# B. Proof of Convergence of SCAFFOLD

## B.1. Convergence of Scaffold's iterates – Proof of Lemma 4.1 and Theorem 4.2

We now analyze the convergence of SCAFFOLD's iterates. Specifically, we aim to demonstrate that, akin to FEDAVG and SGD, the iterates of SCAFFOLD (i.e., its parameters and control variates) converge to a unique stationary distribution.

To establish this result, we first show that SCAFFOLD's updates exhibit contractive behavior under certain conditions. For this purpose, we introduce the following norm, which assigns appropriate weights to each parameter and control variate,

$$\|\mathsf{X}\|_{\Lambda}^2 = \|\theta\|^2 + \frac{\gamma^2 H^2}{N}\sum_{c=1}^{N}\|\xi_{(c)}\|^2 , \tag{13}$$

where $X = (\theta, \xi_{(1)}, \ldots, \xi_{(N)})$. This can be seen as a norm on $\mathbb{R}^{(N+1)d}$ such that $\|X\|_\Lambda^2 = \langle X, \Lambda X \rangle$ for $X \in \mathbb{R}^{(N+1)d}$ and where

$$\Lambda = \mathrm{diag}\left(\mathrm{Id}_d, \frac{\gamma^2 H^2}{N}\mathrm{Id}_d, \ldots, \frac{\gamma^2 H^2}{N}\mathrm{Id}_d\right) \ ,$$

and $\mathrm{Id}_d$ is the $d \times d$ identity matrix. We now show that $\mathsf{S}$ is a contractive operator under the norm $\|\cdot\|_\Lambda$.

**Lemma 4.1** (Restated). *Assume A1 and A2. Let $Z = Z_{(1:N)}^{1:H}$ be i.i.d. random variables satisfying A5. Let the step size $\gamma > 0$ and number of local updates $H > 0$ satisfy $\gamma \le 1/(2L)$ and $\gamma H(L + \mu) \le 1$. Then, for any $\theta, \theta' \in \mathbb{R}^d$ and $\{\xi_{(c)}, \xi'_{(c)}\}_{c=1}^N \in \mathbb{R}^d$ such that $\sum_{c=1}^N \xi_{(c)} = \sum_{c=1}^N \xi'_{(c)} = 0$, it holds that*

$$\mathbb{E}\left[\|\mathsf{S}(X; Z) - \mathsf{S}(X'; Z)\|_\Lambda^2\right] \le \left(1 - \frac{\gamma\mu}{4}\right)^H \|X - X'\|_\Lambda^2 \ , \tag{8}$$

*with $X = (\theta, \xi_{(1)}, \ldots, \xi_{(N)})$, and $X' = (\theta', \xi'_{(1)}, \ldots, \xi'_{(N)})$.*

*Proof.* For readability, we define, for $\theta, \theta', \xi_{(c)}, \xi'_{(c)} \in \mathbb{R}^d$, notations for the global parameter $\theta$ update, the local parameters updates and the control variates $\xi_{(c)}$ updates as,

$$\theta^+ = \mathsf{T}(\theta; \xi_{(1:N)}, Z_{(1:N)}^{1:H}) \ , \qquad \theta_{(c)}^h = \mathsf{T}_{(c)}^h(\theta; \xi_{(c)}, Z_{(c)}^{1:h}) \ , \qquad \xi_{(c)}^+ = \mathsf{V}_{(c)}(\xi_{(c)}; \theta, Z_{(c)}^{1:H}) \ , \tag{14}$$

and similarly for $\theta'$ and $\xi'_{(c)}$,

$$\theta'^+ = \mathsf{T}(\theta'; \xi'_{(1:N)}, Z_{(1:N)}^{1:H}) \ , \qquad \theta_{(c)}'^h = \mathsf{T}_{(c)}^h(\theta'; \xi'_{(c)}, Z_{(c)}^{1:h}) \ , \qquad \xi_{(c)}'^+ = \mathsf{V}_{(c)}(\xi'_{(c)}; \theta', Z_{(c)}^{1:H}) \ . \tag{15}$$

Recall that $\theta^+ = N^{-1}\sum_{c=1}^N \theta_{(c)}^H$ and $\theta'^+ = N^{-1}\sum_{c=1}^N \theta_{(c)}'^H$. We can thus use the fact that $\sum_{c=1}^N \xi_{(c)} = 0$ and $\sum_{c=1}^N \xi'_{(c)} = 0$, as well as Lemma F.2 with $x_c = \theta_{(c)}^H + \gamma H \xi_{(c)}$ and $y_c = \theta_{(c)}'^H + \gamma H \xi'_{(c)}$ to obtain

$$\|\theta^+ - \theta'^+\|^2 = \left\|\frac{1}{N}\sum_{c=1}^N\left(\theta_{(c)}^H + \gamma H \xi_{(c)}\right) - \frac{1}{N}\sum_{c=1}^N\left(\theta_{(c)}'^H + \gamma H \xi'_{(c)}\right)\right\|^2$$

$$= \frac{1}{N}\sum_{c=1}^N\left\|\theta_{(c)}^H + \gamma H \xi_{(c)} - \theta_{(c)}'^H - \gamma H \xi'_{(c)}\right\|^2 - \frac{1}{N}\sum_{c=1}^N\left\|\gamma H\left(\xi_{(c)}^+ - \xi_{(c)}'^+\right)\right\|^2 \ , \tag{16}$$

where we used the fact that $\gamma H \xi_{(c)}^+ = \gamma H \xi_{(c)} + \theta^+ - \theta_{(c)}^H$ and $\gamma H \xi_{(c)}'^+ = \gamma H \xi'_{(c)} + \tilde{\theta}^+ - \tilde{\theta}_{(c)}^H$ in the second term. We now define the shifted parameters, for $c \in \{1, \ldots, N\}$ and $h \in \{0, \ldots, H\}$,

$$\widetilde{\theta}_{(c)}^h = \theta_{(c)}^h + \gamma h \xi_{(c)} \ , \qquad \widetilde{\theta}_{(c)}'^h = \theta_{(c)}'^h + \gamma h \xi'_{(c)} \ . \tag{17}$$

The identity (16) can be rewritten using the notations introduced in (17), which gives

$$\|\mathsf{S}(X; Z) - \mathsf{S}(X'; Z)\|_\Lambda^2 = \frac{1}{N}\sum_{c=1}^N\|\widetilde{\theta}_{(c)}^H - \widetilde{\theta}_{(c)}'^H\|^2 \ . \tag{18}$$

It remains to derive a bound on each term of this sum. We proceed by induction, on $h \in \{0, \ldots, H-1\}$ we have

$$\|\widetilde{\theta}_{(c)}^{h+1} - \widetilde{\theta}_{(c)}'^{h+1}\|^2 = \left\|\widetilde{\theta}_{(c)}^h - \widetilde{\theta}_{(c)}'^h - \gamma\left(\nabla F_{(c)}^{Z_{(c)}^{h+1}}(\theta_{(c)}^h) - \nabla F_{(c)}^{Z_{(c)}^{h+1}}(\theta_{(c)}'^h)\right)\right\|^2 \ .$$

Expanding the square and using (17), we obtain

$$\|\widetilde{\theta}_{(c)}^{h+1} - \widetilde{\theta}_{(c)}'^{h+1}\|^2$$
$$= \left\|\widetilde{\theta}_{(c)}^h - \widetilde{\theta}_{(c)}'^h\right\|^2 + \gamma^2\left\|\nabla F_{(c)}^{Z_{(c)}^{h+1}}(\theta_{(c)}^h) - \nabla F_{(c)}^{Z_{(c)}^{h+1}}(\theta_{(c)}'^h)\right\|^2 - 2\gamma\left\langle\widetilde{\theta}_{(c)}^h - \widetilde{\theta}_{(c)}'^h, \nabla F_{(c)}^{Z_{(c)}^{h+1}}(\theta_{(c)}^h) - \nabla F_{(c)}^{Z_{(c)}^{h+1}}(\theta_{(c)}'^h)\right\rangle$$

$$= \left\|\widetilde{\theta}_{(c)}^h - \widetilde{\theta}_{(c)}'^h\right\|^2 + \gamma^2 \left\|\nabla F_{(c)}^{Z_{(c)}^{h+1}}(\theta_{(c)}^h) - \nabla F_{(c)}^{Z_{(c)}^{h+1}}(\theta_{(c)}'^h)\right\|^2$$
$$- 2\gamma\left\langle\theta_{(c)}^h - \theta_{(c)}'^h, \nabla F_{(c)}^{Z_{(c)}^{h+1}}(\theta_{(c)}^h) - \nabla F_{(c)}^{Z_{(c)}^{h+1}}(\theta_{(c)}'^h)\right\rangle - 2\gamma^2 h\left\langle\xi_{(c)} - \xi_{(c)}', \nabla F_{(c)}^{Z_{(c)}^{h+1}}(\theta_{(c)}^h) - \nabla F_{(c)}^{Z_{(c)}^{h+1}}(\theta_{(c)}'^h)\right\rangle .$$

Now, using Young's inequality to bound $2\gamma^2 hab = 2(\gamma^{3/2}L^{1/2}ha)(\gamma^{1/2}L^{-1/2}b) \le \gamma^3 h^2 La^2 + \gamma L^{-1}b^2$, we get

$$- 2\gamma^2 h\left\langle\xi_{(c)} - \xi_{(c)}', \nabla F_{(c)}^{Z_{(c)}^{h+1}}(\theta_{(c)}^h) - \nabla F_{(c)}^{Z_{(c)}^{h+1}}(\theta_{(c)}'^h)\right\rangle \le \gamma^3 h^2 L\left\|\xi_{(c)} - \xi_{(c)}'\right\|^2 + \frac{\gamma}{L}\left\|\nabla F_{(c)}^{Z_{(c)}^{h+1}}(\theta_{(c)}^h) - \nabla F_{(c)}^{Z_{(c)}^{h+1}}(\theta_{(c)}'^h)\right\|^2 .$$

Plugging this in the previous inequality and using the co-coercivity of the gradient (10), we have

$$\|\widetilde{\theta}_{(c)}^{h+1} - \widetilde{\theta}_{(c)}'^{h+1}\|^2 \le \|\widetilde{\theta}_{(c)}^h - \widetilde{\theta}_{(c)}'^h\|^2 + \gamma^3 h^2 L\|\xi_{(c)} - \xi_{(c)}'\|^2 - (\gamma - \gamma^2 L)\langle\theta_{(c)}^h - \theta_{(c)}'^h, \nabla F_{(c)}^{Z_{(c)}^{h+1}}(\theta_{(c)}^h) - \nabla F_{(c)}^{Z_{(c)}^{h+1}}(\theta_{(c)}'^h)\rangle .$$

Using the fact that $\gamma \le 1/2L$ to bound $-(\gamma - \gamma^2 L) \le -\gamma/2$, taking the conditional expectation and using that $Z_{(c)}^{h+1}$ is independent of $Z_{(c)}^{1:h}$, and monotonicity of the gradient (11), we obtain

$$\mathbb{E}\left[\left\|\widetilde{\theta}_{(c)}^{h+1} - \widetilde{\theta}_{(c)}'^{h+1}\right\|^2 \,\middle|\, Z_{(c)}^{1:h}\right] \le \left\|\widetilde{\theta}_{(c)}^h - \widetilde{\theta}_{(c)}'^h\right\|^2 - \frac{\gamma\mu}{2}\left\|\theta_{(c)}^h - \theta_{(c)}'^h\right\|^2 + \gamma^3 h^2 L\|\xi_{(c)} - \xi_{(c)}'\|^2 . \tag{19}$$

Now, we remark that, for $a, b \in \mathbb{R}^d$, we have $a^2 = (a - b + b)^2 \le 2(a-b)^2 + 2b^2$, which implies that $-(a-b)^2 \le -\frac{1}{2}a^2 + b^2$. Therefore, we have

$$-\frac{\gamma\mu}{2}\left\|\theta_{(c)}^{h+1} - \tilde{\theta}_{(c)}^{h+1}\right\|^2 = -\frac{\gamma\mu}{2}\left\|\theta_{(c)}^h - \theta_{(c)}'^h - \gamma h(\xi_{(c)} - \xi_{(c)}')\right\|^2 \le -\frac{\gamma\mu}{4}\left\|\theta_{(c)}^h - \theta_{(c)}'^h\right\|^2 + \frac{\gamma^3 h^2\mu}{2}\left\|\xi_{(c)} - \xi_{(c)}'\right\|^2 .$$

Using this inequality in (19), we obtain the following inequality

$$\mathbb{E}\left[\left\|\widetilde{\theta}_{(c)}^{h+1} - \widetilde{\theta}_{(c)}'^{h+1}\right\|^2 \,\middle|\, Z_{(c)}^{1:h}\right] \le \left(1 - \frac{\gamma\mu}{4}\right)\left\|\widetilde{\theta}_{(c)}^h - \widetilde{\theta}_{(c)}'^h\right\|^2 + \left(\gamma^3 h^2\mu + \gamma^3 h^2 L\right)\left\|\xi_{(c)} - \xi_{(c)}'\right\|^2 . \tag{20}$$

Taking the expectation in the last inequality, a straightforward induction leads to

$$\mathbb{E}\left[\left\|\widetilde{\theta}_{(c)}^H - \widetilde{\theta}_{(c)}'^H\right\|^2\right] \le \left(1 - \frac{\gamma\mu}{4}\right)^H \left\|\theta - \tilde{\theta}\right\|^2 + \frac{\gamma^3 H^2(H-1)(L+\mu)}{2}\left\|\xi_{(c)} - \xi_{(c)}'\right\|^2 .$$

Consequently, whenever $\gamma H(L + \mu) \le 1$, we can sum this inequality for $c = 1$ to $N$ to obtain

$$\mathbb{E}\left[\frac{1}{N}\sum_{c=1}^N \left\|\widetilde{\theta}_{(c)}^H - \widetilde{\theta}_{(c)}'^H\right\|^2\right] \le \left(1 - \frac{\gamma\mu}{4}\right)^H \left\|\theta - \theta\right\|^2 + \frac{1}{2}\frac{\gamma^2 H^2}{N}\sum_{c=1}^N\left\|\xi_{(c)} - \xi_{(c)}'\right\|^2 \le \left(1 - \frac{\gamma\mu}{4}\right)^H \left\|\mathrm{X} - \mathrm{X}'\right\|_\Lambda^2 ,$$

where the second inequality comes from $\frac{1}{2} \cdot \gamma^2 H^2 \le (1 - \frac{\gamma\mu}{4})^H \cdot \gamma^2 H^2$. $\qquad\square$

**Theorem 4.2** (Restated). *Assume A 1, A 2, and A 5. Let $\gamma > 0$, $H > 0$, such that $\gamma \le 1/(2L)$ and $\gamma H(L + \mu) \le 1$. Let $\{\mathrm{X}^t\}_{t=0}^\infty$, with $\mathrm{X}^t = (\theta^t, \xi_{(1)}^t, \dots, \xi_{(N)}^t)$, be SCAFFOLD's iterates with step size $\gamma$ and $H$ local steps and $\mathrm{X}^0 \sim \rho$, where $\rho$ is a probability measure on $\mathcal{X}$ such that $\int \|\mathrm{X}\|_\Lambda^2 \rho(\mathrm{dX}) < \infty$. Then, the distribution $\rho\mathrm{K}_{(\gamma,H)}^t$ of $\mathrm{X}^t$ converges to a unique stationary distribution $\pi_{(\gamma,H)}$ satisfying $\int \|\mathrm{X}\|_\Lambda^2 \pi_{(\gamma,H)}(\mathrm{dX}) < \infty$, and for any $t \in \mathbb{N}$,*

$$\mathbf{W}_2^2(\rho\mathrm{K}_{(\gamma,H)}^t, \pi_{(\gamma,H)}) \le \left(1 - \frac{\gamma\mu}{4}\right)^{Ht} \mathbf{W}_2^2(\rho, \pi_{(\gamma,H)}) .$$

*Proof.* We use Douc et al. (2018, Theorem 20.3.4) with the cost function $c(\mathrm{X}, \tilde{\mathrm{X}}) = \|\mathrm{X} - \tilde{\mathrm{X}}\|_\Lambda^2$, where the norm $\|\cdot\|_\Lambda$ is defined in (13). $\qquad\square$

Note that the convergence toward the stationary distribution is geometric.

## B.2. Bound on SCAFFOLD's Global Iterates in the Stationary Distribution – Proof of Lemma 4.3 and Theorem 4.4

**Lemma 4.3** (Restated). *Assume A1, A2, A5. Let $Z = Z_{(1:N)}^{1:H}$ be i.i.d. random variables satisfying A5. Assume the step size $\gamma$ and the number of local updates $H$ satisfy $\gamma H(L + \mu) \leq 1$. Then, for all $\theta \in \mathbb{R}^d$ and $\{\xi_{(c)}\}_{c=1}^N \subset \mathbb{R}^d$ such that $\sum_{c=1}^N \xi_{(c)} = 0$,*

$$\mathbb{E}\Big[\|\mathsf{S}(\mathrm{X}; Z) - \mathrm{X}^\star\|_\Lambda^2\Big] \leq \Big(1 - \frac{\gamma\mu}{4}\Big)^H \|\mathrm{X} - \mathrm{X}^\star\|_\Lambda^2 + 2\gamma^2 H\sigma_\star^2 \,,$$

*with the global iterate vector $\mathrm{X} = (\theta, \xi_{(1)}, \ldots, \xi_{(N)})$.*

*Proof.* As in Lemma 4.1, we define, for $\vartheta, \xi \in \mathbb{R}^d$, notations for the global parameter update, the local parameter updates and the control variates updates as,

$$\vartheta^+ = \mathsf{T}(\vartheta; \xi_{(1:N)}, Z_{(1:N)}^{1:H}) \,, \qquad \vartheta_{(c)}^h = \mathsf{T}_{(c)}^h(\vartheta; \xi_{(c)}, Z_{(c)}^{1:h}) \,, \qquad \xi_{(c)}^+ = \mathsf{V}_{(c)}(\xi_{(c)}; \vartheta, Z_{(c)}^{1:H}) \,, \tag{21}$$

for $c \in \{1, \ldots, N\}$ and $h \in \{0, \ldots, H\}$. Recall that $\vartheta^+ = N^{-1} \sum_{c=1}^N \vartheta_{(c)}^H$. We can thus use the fact that $\sum_{c=1}^N \xi_{(c)} = 0$ and $\sum_{c=1}^N \xi_{(c)}^\star = 0$, as well as Lemma F.2 with $x_c = \vartheta_{(c)}^H + \gamma H\xi_{(c)}$ and $y_c = \theta^\star + \gamma H\xi_{(c)}^\star$ to obtain

$$\|\vartheta^+ - \theta^\star\|^2 = \Big\| \frac{1}{N} \sum_{c=1}^N \Big(\vartheta_{(c)}^H + \gamma H\xi_{(c)}\Big) - \frac{1}{N} \sum_{c=1}^N \Big(\theta^\star + \gamma H\xi_{(c)}^\star\Big) \Big\|^2$$

$$= \frac{1}{N} \sum_{c=1}^N \Big\| \vartheta_{(c)}^H + \gamma H\xi_{(c)} - \theta^\star - \gamma H\xi_{(c)}^\star \Big\|^2 - \frac{1}{N} \sum_{c=1}^N \Big\| \gamma H\Big(\xi_{(c)}^+ - \xi_{(c)}^\star\Big) \Big\|^2 \,, \tag{22}$$

where we used the fact that $\gamma H\xi_{(c)}^+ = \gamma H\xi_{(c)} + \vartheta_{(c)}^H - \vartheta^+$ in the second term. Define for $\vartheta \in \mathbb{R}^d$, $c \in \{1, \ldots, N\}$ and $h \in \{0, \ldots, H\}$

$$\widetilde{\vartheta}_{(c)}^h = \mathsf{T}_{(c)}^h(\vartheta; \xi_{(c)}) + \gamma h(\xi_{(c)} - \xi_{(c)}^\star) = \vartheta_{(c)}^h + \gamma h(\xi_{(c)} - \xi_{(c)}^\star) \,. \tag{23}$$

The identity in (22) can be rewritten using this expression, as well as the norm $\|\cdot\|_\Lambda$ defined in (13),

$$\|\mathsf{S}(\mathrm{X}; Z) - \mathrm{X}^\star\|_\Lambda^2 = \frac{1}{N} \sum_{c=1}^N \|\widetilde{\vartheta}_{(c)}^H - \theta^\star\|^2 \,. \tag{24}$$

It remains to derive a bound on each term of this sum, by induction on $h \in \{0, \ldots, H-1\}$. We have, for $c \in \{1, \ldots, N\}$,

$$\|\widetilde{\vartheta}_{(c)}^{h+1} - \theta^\star\|^2 = \Big\| \widetilde{\vartheta}_{(c)}^h - \theta^\star - \gamma \Big(\nabla F_{(c)}^{Z_{(c)}^{h+1}}(\vartheta_{(c)}^h) + \xi_{(c)}^\star\Big) \Big\|^2 \,.$$

Expanding the square and using (23) to write $\widetilde{\vartheta}_{(c)}^h = \vartheta_{(c)}^h + \gamma h(\xi_{(c)} - \xi_{(c)}^\star)$, we obtain

$$\|\widetilde{\vartheta}_{(c)}^{h+1} - \theta^\star\|^2 = \Big\| \widetilde{\vartheta}_{(c)}^h - \theta^\star \Big\|^2 - 2\gamma\Big\langle \widetilde{\vartheta}_{(c)}^h - \theta^\star, \nabla F_{(c)}^{Z_{(c)}^{h+1}}(\vartheta_{(c)}^h) + \xi_{(c)}^\star \Big\rangle + \gamma^2 \Big\| \nabla F_{(c)}^{Z_{(c)}^{h+1}}(\vartheta_{(c)}^h) + \xi_{(c)}^\star \Big\|^2$$

$$= \Big\| \widetilde{\vartheta}_{(c)}^h - \theta^\star \Big\|^2 + \gamma^2 \Big\| \nabla F_{(c)}^{Z_{(c)}^{h+1}}(\vartheta_{(c)}^h) + \xi_{(c)}^\star \Big\|^2$$

$$- 2\gamma\Big\langle \vartheta_{(c)}^h - \theta^\star, \nabla F_{(c)}^{Z_{(c)}^{h+1}}(\vartheta_{(c)}^h) + \xi_{(c)}^\star \Big\rangle - 2\gamma^2 h\Big\langle \xi_{(c)} - \xi_{(c)}^\star, \nabla F_{(c)}^{Z_{(c)}}(\vartheta_{(c)}^h) + \xi_{(c)}^\star \Big\rangle \,.$$

Replacing $\xi_{(c)}^\star = -\nabla f_{(c)}(\theta^\star)$, we have

$$\mathbb{E}\left[\Big\| \widetilde{\vartheta}_{(c)}^{h+1} - \theta^\star \Big\|^2 \,\Big|\, Z_{(c)}^{1:h}\right] = \Big\| \widetilde{\vartheta}_{(c)}^h - \theta^\star \Big\|^2 + \gamma^2 \mathbb{E}\left[\Big\| \nabla F_{(c)}^{Z_{(c)}^{h+1}}(\vartheta_{(c)}^h) - \nabla f_{(c)}(\theta^\star) \Big\|^2 \,\Big|\, Z_{(c)}^{1:h}\right]$$

$$- 2\gamma\Big\langle \vartheta_{(c)}^h - \theta^\star, \nabla f_{(c)}(\vartheta_{(c)}^h) - \nabla f_{(c)}(\theta^\star) \Big\rangle - 2\gamma^2 h\Big\langle \xi_{(c)} - \xi_{(c)}^\star, \nabla f_{(c)}(\vartheta_{(c)}^h) - \nabla f_{(c)}(\theta^\star) \Big\rangle \,.$$

Using the inequality $\|u+v\|^2 \le 2\|u\|^2 + \|v\|^2$ and bounding the two terms using co-coercivity (10) and A5, we can bound

$$\mathbb{E}\left[\left\|\nabla F_{(c)}^{Z_{(c)}^{h+1}}(\vartheta_{(c)}^h) - \nabla f_{(c)}(\theta^\star)\right\|^2 \;\middle|\; Z_{(c)}^{1:h}\right] = \mathbb{E}\left[\left\|\nabla F_{(c)}^{Z_{(c)}^{h+1}}(\vartheta_{(c)}^h) - \nabla F_{(c)}^{Z_{(c)}^{h+1}}(\theta^\star) + \nabla F_{(c)}^{Z_{(c)}^{h+1}}(\theta^\star) - \nabla f_{(c)}(\theta^\star)\right\|^2 \;\middle|\; Z_{(c)}^{1:h}\right]$$
$$\le 2L\langle \vartheta_{(c)}^h - \theta^\star, \nabla f_{(c)}(\vartheta_{(c)}^h) - \nabla f_{(c)}(\theta^\star)\rangle + 2\sigma_\star^2 \ .$$

Now, using Young's inequality to bound $2\gamma^2 hab = 2(\gamma^{3/2}L^{1/2}ha)(\gamma^{1/2}L^{-1/2}b) \le \gamma^3 h^2 L a^2 + \gamma L^{-1}b^2$ and co-coercivity of the gradient (10), we get

$$-2\gamma^2 h\left\langle \xi_{(c)} - \xi_{(c)}^\star, \nabla f_{(c)}(\vartheta_{(c)}^h) - \nabla f_{(c)}(\theta^\star)\right\rangle \le \gamma^3 h^2 L\|\xi_{(c)} - \xi_{(c)}^\star\|^2 + \frac{\gamma}{L}\|\nabla f_{(c)}(\vartheta_{(c)}^h) - \nabla f_{(c)}(\theta^\star)\|^2$$
$$\le \gamma^3 h^2 L\|\xi_{(c)} - \xi_{(c)}^\star\|^2 + \gamma\langle \vartheta_{(c)}^h - \theta^\star, \nabla f_{(c)}(\vartheta_{(c)}^h) - \nabla f_{(c)}(\theta^\star)\rangle \ .$$

Plugging the last two equations in the inequality that decompose the update above, we have

$$\mathbb{E}\left[\left\|\widetilde{\vartheta}_{(c)}^{h+1} - \theta^\star\right\|^2 \;\middle|\; Z_{(c)}^{1:h}\right] \le \left\|\widetilde{\vartheta}_{(c)}^h - \theta^\star\right\|^2 + \gamma^3 h^2 L\|\xi_{(c)} - \xi_{(c)}^\star\|^2$$
$$- (\gamma - 2\gamma^2 L)\langle \vartheta_{(c)}^h - \theta^\star, \nabla f_{(c)}(\vartheta_{(c)}^h) - \nabla f_{(c)}(\theta^\star)\rangle + 2\gamma^2\sigma_\star^2 \ .$$

And using the fact that $\gamma \le 1/4L$ to bound $-(\gamma - 2\gamma^2 L) \le -\gamma/2$, and the monotonocity of the gradient (11), we obtain

$$\mathbb{E}\left[\left\|\widetilde{\vartheta}_{(c)}^{h+1} - \theta^\star\right\|^2 \;\middle|\; Z_{(c)}^{1:h}\right] \le \left\|\widetilde{\vartheta}_{(c)}^h - \theta^\star\right\|^2 - \frac{\gamma\mu}{2}\left\|\vartheta_{(c)}^h - \theta^\star\right\|^2 + \gamma^3 h^2 L\|\xi_{(c)} - \xi_{(c)}^\star\|^2 + 2\gamma^2\sigma_\star^2 \ . \tag{25}$$

Now, we remark that, for $a, b \in \mathbb{R}^d$, we have $\|a\|^2 = \|a - b + b\|^2 \le 2\|a - b\|^2 + 2\|b\|^2$, which implies that $-\|a - b\|^2 \le -\frac{1}{2}\|a\|^2 + \|b\|^2$. Therefore, we have

$$-\frac{\gamma\mu}{2}\left\|\vartheta_{(c)}^h - \theta^\star\right\|^2 = -\frac{\gamma\mu}{2}\left\|\widetilde{\vartheta}_{(c)}^h - \theta^\star - \gamma h(\xi_{(c)} - \xi_{(c)}^\star)\right\|^2 \le -\frac{\gamma\mu}{4}\left\|\widetilde{\vartheta}_{(c)}^h - \theta^\star\right\|^2 + \frac{\gamma^3 h^2\mu}{2}\left\|\xi_{(c)} - \xi_{(c)}^\star\right\|^2 \ .$$

Using this inequality in (25), we obtain the following inequality

$$\mathbb{E}\left[\left\|\widetilde{\vartheta}_{(c)}^{h+1} - \theta^\star\right\|^2 \;\middle|\; Z_{(c)}^{1:h}\right] \le \left(1 - \frac{\gamma\mu}{4}\right)\left\|\widetilde{\vartheta}_{(c)}^h - \theta^\star\right\|^2 + \left(\gamma^3 h^2 L + \gamma^3 h^2\mu/2\right)\|\xi_{(c)} - \xi_{(c)}^\star\|^2 + 2\gamma^2\sigma_\star^2 \ . \tag{26}$$

Applying (26) recursively, we obtain

$$\mathbb{E}\left[\left\|\widetilde{\vartheta}_{(c)}^H - \theta^\star\right\|^2\right] \le \left(1 - \frac{\gamma\mu}{4}\right)^H \|\theta - \theta^\star\|^2 + \frac{\gamma^3 H^2(H-1)(L+\mu)}{2}\|\xi_{(c)} - \xi_{(c)}^\star\|^2 + 2\gamma^2 H\sigma_\star^2 \ . \tag{27}$$

Consequently, whenever $\gamma H(L+\mu) \le 1$, we can sum this inequality for $c = 1$ to $N$ to obtain

$$\frac{1}{N}\sum_{c=1}^N \mathbb{E}\left[\|\widetilde{\vartheta}_{(c)}^H - \theta^\star\|^2\right] \le \left(1 - \frac{\gamma\mu}{4}\right)^H \|\theta - \theta^\star\|^2 + \frac{1}{2}\frac{\gamma^2 H^2}{N}\sum_{c=1}^N \|\xi_{(c)} - \xi_{(c)}^\star\|^2 + 2\gamma^2 H\sigma_\star^2 \tag{28}$$
$$\le \left(1 - \frac{\gamma\mu}{4}\right)^H \|\mathrm{X} - \mathrm{X}^\star\|_\Lambda^2 + 2\gamma^2 H\sigma_\star^2 \ ,$$

and we get the result of the lemma by taking the expectation of (24) and plugging this bound. $\square$

**Theorem 4.4** (Restated). *Assume A1, A2 and A5. Let $\gamma > 0$ be the step size and $H > 0$ the number of local updates. Assume that $\gamma \le 1/4L$ and $\gamma H(L + \mu) \le 1$. Then, for any $T > 0$ and any $\mathrm{X}^0 \in \mathcal{X}$, the iterates and control variates of* SCAFFOLD, $\mathrm{X}^T = (\theta^T, \xi_{(1)}^T, \ldots, \xi_{(N)}^T)$, *satisfy the inequality*

$$\mathbb{E}\left[\|\mathrm{X}^T - \mathrm{X}^\star\|_\Lambda^2\right] \le \left(1 - \frac{\gamma\mu}{4}\right)^{HT}\|\mathrm{X}^0 - \mathrm{X}^\star\|_\Lambda^2 + \frac{8\gamma}{\mu}\sigma_\star^2 \ ,$$

*where $\mathrm{X}^\star$ is the global optimal vector.*

*Proof.* The proof follows by applying recursively Lemma 4.3 with the natural filtration of the process $(X^t)_{t=0}^{\infty}$. $\qquad\square$

The following corollary is a direct consequence of Theorem 4.4, and gives a crude bound on the squared error of $\theta$ in the stationary distribution.

**Corollary B.1.** *Assume A1, A2 and A5. Let $Z = Z_{(1:N)}^{1:H}$ be i.i.d. random variables satisfying A5. Let $\gamma > 0$ be the step size and $H > 0$ the number of local updates of SCAFFOLD. Assume that $\gamma \leq 1/4L$ and $\gamma H(L + \mu) \leq 1$. Then, for all $h \in \{0, \ldots, H\}$, it holds that*

$$\int \left\| \theta - \theta^\star \right\|^2 \pi_{(\gamma, H)}(\mathrm{d}\theta, \mathrm{d}\Xi) \leq \frac{8\gamma}{\mu}\sigma_\star^2 \ , \tag{29}$$

$$\frac{\gamma^2 H^2}{N} \sum_{c=1}^{N} \int \left\| \xi_{(c)} - \xi_{(c)}^\star \right\|^2 \pi_{(\gamma, H)}(\mathrm{d}\theta, \mathrm{d}\Xi) \leq \frac{8\gamma}{\mu}\sigma_\star^2 \ , \tag{30}$$

$$\frac{1}{N} \sum_{c=1}^{N} \int \mathbb{E}\left[ \left\| \mathsf{T}_{(c)}^h(\theta; \xi_{(c)}, Z_{(c)}^{1:h}) - \theta^\star \right\|^2 \right] \pi_{(\gamma, H)}(\mathrm{d}\theta, \mathrm{d}\Xi) \leq \frac{8\gamma}{\mu}\sigma_\star^2 \ , \tag{31}$$

*where $\Xi = (\xi_{(1)}, \ldots, \xi_{(N)}) \in \mathbb{R}^{N \times d}$.*

*Proof.* Inequalities (29) and (30) follow from Theorem 4.4. The third inequality (31) is obtained by unrolling (26) until $h$ similarly to (27) and summing over $c = 1$ to $N$. $\qquad\square$

## B.3. Bounds on SCAFFOLD's Local Iterates and Control Variates in the Stationary Distribution – Proof of Lemma 4.6

**Lemma B.2.** *Assume A1, A2, A5. Let $Z = Z_{(1:N)}^{1:H}$ be i.i.d. random variables satisfying A5. Assume the step size $\gamma$ and the number of local updates $H$ satisfy $\gamma H(L + \mu) \leq 1/12$ and $\gamma\beta \leq L$. Under these conditions, it holds that*

$$\int \mathbb{E}\left[\|\mathsf{T}_{(c)}^h(\theta; \xi_{(c)}, Z_{(c)}^{1:h}) - \theta^\star\|^2\right] \pi_{(\gamma, H)}(\mathrm{d}\theta, \mathrm{d}\Xi) \leq \frac{18\gamma}{\mu}\sigma_\star^2 + 3\gamma^2 H^2 \int \|\xi_{(c)} - \xi_{(c)}^\star\|^2 \pi_{(\gamma, H)}(\mathrm{d}\theta, \mathrm{d}\Xi) \ , \tag{32}$$

$$\int \|\xi_{(c)} - \xi_{(c)}^\star\|^2 \pi_{(\gamma, H)}(\mathrm{d}\theta, \mathrm{d}\Xi) \leq \frac{8(L + \mu)}{\mu H}\sigma_\star^2 + \frac{4L^2 + 2\beta}{H} \sum_{h=0}^{H-1} \int \mathbb{E}\left[\|\mathsf{T}_{(c)}^h(\theta; \xi_{(c)}, Z_{(c)}^{1:h}) - \theta^\star\|^2\right] \pi_{(\gamma, H)}(\mathrm{d}\theta, \mathrm{d}\Xi) \ . \tag{33}$$

*Proof.* Let $\theta \in \mathbb{R}^d$ and $\{\xi_{(c)}\}_{c=1}^N \subset \mathbb{R}^d$ that satisfy the constraints $\sum_{c=1}^N \xi_{(c)} = 0$. Based on the proof of Lemma 4.3, we define a notation for the local parameters and their counterpart with ideal control variates,

$$\theta_{(c)}^h = \mathsf{T}_{(c)}^h(\theta; \xi_{(c)}, Z_{(c)}^{1:h}) \tag{34}$$

$$\widetilde{\theta}_{(c)}^h = \theta_{(c)}^h + \gamma h(\xi_{(c)} - \xi_{(c)}^\star) \ . \tag{35}$$

**Bound on the local iterates.** Then, following the same lines of proof as Lemma 4.3 (see (26)) and using the fact that $\gamma H L \leq 1$, we obtain, for any $h \leq H$, and $c \in \{1, \ldots, N\}$,

$$\mathbb{E}\left[\|\widetilde{\theta}_{(c)}^h - \theta^\star\|^2\right] \leq \|\theta - \theta^\star\|^2 + \frac{\gamma^2 H^2}{2}\|\xi_{(c)} - \xi_{(c)}^\star\|^2 + 2\gamma^2 H \sigma_\star^2 \ ,$$

Since $\theta_{(c)}^h = \widetilde{\theta}_{(c)}^h + \gamma h(\xi_{(c)} - \xi_{(c)}^\star)$, this gives the inequality

$$\mathbb{E}\left[\|\theta_{(c)}^h - \theta^\star\|^2\right] \leq 2\mathbb{E}\left[\|\widetilde{\theta}_{(c)}^h - \theta^\star\|^2\right] + 2\gamma^2 h^2 \|\xi_{(c)} - \xi_{(c)}^\star\|^2$$

$$\leq 2\|\theta - \theta^\star\|^2 + 3\gamma^2 H^2 \|\xi_{(c)} - \xi_{(c)}^\star\|^2 + 4\gamma^2 H \sigma_\star^2 \ .$$

Integrating over the stationary distribution of SCAFFOLD's iterates and using (29) from Corollary B.1, we obtain (32).

**Bound on control variates.** For ease of notation, we define

$$\varepsilon_{(c)}^{h+1} = \varepsilon_{(c)}^{Z_{(c)}^{h+1}}(\theta_{(c)}^h) \ . \tag{36}$$

Let $c \in \{1, \ldots, N\}$, the control variate update can be written as

$$\xi_{(c)}^+ = \xi_{(c)} + \frac{1}{\gamma H}\left(\theta - \gamma \sum_{h=0}^{H-1} \nabla f_{(c)}(\theta_{(c)}^h) + \xi_{(c)} + \varepsilon_{(c)}^{h+1} - \theta + \frac{\gamma}{N}\sum_{i=1}^{N}\sum_{h=0}^{H-1}\nabla f_{(i)}(\theta_{(i)}^h) + \xi_{(i)} + \varepsilon_{(i)}^{h+1}\right)$$

$$= \xi_{(c)} - \frac{1}{\gamma H}\left(\gamma \sum_{h=0}^{H-1} \nabla f_{(c)}(\theta_{(c)}^h) + \xi_{(c)} + \varepsilon_{(c)}^{h+1} - \frac{\gamma}{N}\sum_{i=1}^{N}\sum_{h=0}^{H-1}\nabla f_{(i)}(\theta_{(i)}^h) + \xi_{(i)} + \varepsilon_{(i)}^{h+1}\right) .$$

Using $\sum_{i=1}^{N}\xi_{(i)} = 0$, $\sum_{i=1}^{N}\nabla f_{(i)}(\theta^\star) = 0$, $\xi_{(c)}^\star = -\nabla f_{(c)}(\theta^\star)$, and reorganizing the terms, this gives

$$\xi_{(c)}^+ - \xi_{(c)}^\star = \xi_{(c)} - \xi_{(c)}^\star - \frac{1}{NH}\sum_{i=1}^{N}\sum_{h=0}^{H-1}\left(\nabla f_{(c)}(\theta_{(c)}^h) + \xi_{(c)} - \nabla f_{(i)}(\theta_{(i)}^h) + \varepsilon_{(c)}^{h+1} - \varepsilon_{(i)}^{h+1}\right) \tag{37}$$

$$= \frac{1}{NH}\sum_{i=1}^{N}\sum_{h=0}^{H-1}\left(\left(\nabla f_{(i)}(\theta_{(i)}^h) - \nabla f_{(i)}(\theta^\star)\right) - \left(\nabla f_{(c)}(\theta_{(c)}^h) - \nabla f_{(c)}(\theta^\star)\right) + \varepsilon_{(i)}^{h+1} - \varepsilon_{(c)}^{h+1}\right) . \tag{38}$$

Taking the squared norm and expectation of (38), we obtain

$$\mathbb{E}\left[\|\xi_{(c)}^+ - \xi_{(c)}^\star\|^2\right] \leq 2\mathbb{E}\left[\left\|\frac{1}{NH}\sum_{i=1}^{N}\sum_{h=0}^{H-1}\left(\nabla f_{(i)}(\theta_{(i)}^h) - \nabla f_{(i)}(\theta^\star)\right) - \left(\nabla f_{(c)}(\theta_{(c)}^h) - \nabla f_{(c)}(\theta^\star)\right)\right\|^2\right]$$

$$+ 2\mathbb{E}\left[\left\|\frac{1}{NH}\sum_{i=1}^{N}\sum_{h=0}^{H-1}\varepsilon_{(i)}^{h+1} - \varepsilon_{(c)}^{h+1}\right\|^2\right] .$$

Using Jensen's inequality, as well as $\mathbb{E}\left[\varepsilon_{(c)}^{h+1} \,\middle|\, Z_{(1:N)}^{1:h}\right] = 0$ a.s. and $\mathbb{E}\left[\varepsilon_{(c)}^{h+1}\varepsilon_{(i)} \,\middle|\, Z_{(1:N)}^{1:h}\right] = 0$ for all $c, i \in \{1, \ldots, N\}$, $i \neq c$ and $h \in \{0, \ldots, H-1\}$, we have

$$\mathbb{E}\left[\|\xi_{(c)}^+ - \xi_{(c)}^\star\|^2\right] \leq \frac{4}{NH}\sum_{i=1}^{N}\sum_{h=0}^{H-1}\mathbb{E}\left[\|\nabla f_{(i)}(\theta_{(i)}^h) - \nabla f_{(i)}(\theta^\star)\|^2 + \|\nabla f_{(i)}(\theta_{(c)}^h) - \nabla f_{(c)}(\theta^\star)\|^2\right]$$

$$+ \frac{2}{NH^2}\sum_{i=1}^{N}\sum_{h=0}^{H-1}\mathbb{E}\left[\|\varepsilon_{(i)}^{h+1}\|^2 + \|\varepsilon_{(c)}^{h+1}\|^2\right] .$$

By Lipschitzness of the gradient (12) and smoothness of the error noise (A5),

$$\mathbb{E}\left[\|\xi_{(c)}^+ - \xi_{(c)}^\star\|^2\right] \leq \frac{4L^2}{NH}\sum_{i=1}^{N}\sum_{h=0}^{H-1}\mathbb{E}\left[\|\theta_{(i)}^h - \theta^\star\|^2 + \|\theta_{(c)}^h - \theta^\star\|^2\right]$$

$$+ \frac{2}{NH^2}\sum_{i=1}^{N}\sum_{h=0}^{H-1}\left\{\beta\mathbb{E}\left[\|\theta_{(i)}^h - \theta^\star\|^2 + \|\theta_{(c)}^h - \theta^\star\|^2\right] + 4\sigma_\star^2\right\}$$

$$\leq \frac{8}{H}\sigma_\star^2 + \frac{4L^2 + 2\beta}{NH}\sum_{i=1}^{N}\sum_{h=0}^{H-1}\mathbb{E}\left[\|\theta_{(i)}^h - \theta^\star\|^2 + \|\theta_{(c)}^h - \theta^\star\|^2\right] .$$

Integrating over the stationary distribution of SCAFFOLD's iterates and using (31) from Corollary B.1 gives inequality (33). $\qquad\square$

**Lemma 4.6** (Restated). *Assume A1, A2. Let $Z = Z_{(1:N)}^{1:H}$ be i.i.d. random variables satisfying A5. Assume the step size $\gamma$ and the number of local updates $H$ satisfy $\gamma H(L + \mu) \leq 1/12$. Under these conditions, for any $h \in \{0, \ldots, H\}$ and $c \in \{1, \ldots, N\}$, it holds that,*

$$\int \mathbb{E}\left[\|\mathsf{T}_{(c)}^h(\theta; \xi_{(c)}, Z_{(c)}^{1:h}) - \theta^\star\|^2\right]\pi_{(\gamma, H)}(\mathrm{d}\theta, \mathrm{d}\Xi) \leq \frac{28\gamma\sigma_\star^2}{\mu} , \tag{9}$$

$$\int \|\xi_{(c)} - \xi_{(c)}^\star\|^2 \pi_{(\gamma, H)}(\mathrm{d}\theta, \mathrm{d}\Xi) \leq \frac{54L\sigma_\star^2}{\mu H} . \tag{10}$$

*Proof.* **Solving the system of inequations.** We now aim to find constants $C_{(c)}^{\theta}$ and $C_{(c)}^{\xi}$, for $c \in \{1, \ldots, N\}$, such that for all $h \in \{0, \ldots, H\}$,

$$\int \mathbb{E}\left[\|\mathsf{T}_{(c)}^h(\theta; \xi_{(c)}, Z_{(c)}^{1:h}) - \theta^{\star}\|^2\right] \pi_{(\gamma, H)}(\mathrm{d}\theta, \mathrm{d}\Xi) \leq C_{(c)}^{\theta} \quad, \quad \text{and} \quad \int \|\xi_{(c)} - \xi_{(c)}^{\star}\|^2 \pi_{(\gamma, \mathrm{H})}(\mathrm{d}\theta, \mathrm{d}\Xi) \leq C_{(c)}^{\xi} \quad.$$

By the first part of the lemma, we have

$$C_{(c)}^{\theta} \leq \frac{18\gamma}{\mu}\sigma_{\star}^2 + 3\gamma^2 H^2 C_{(c)}^{\xi} \quad, \quad \text{and} \quad C_{(c)}^{\xi} \leq \frac{9\mathrm{L}}{\mu\mathrm{H}}\sigma_{\star}^2 + 12\mathrm{L}^2 C_{(c)}^{\theta} \quad.$$

Since $\gamma H L \leq 1/12$, this implies that

$$C_{(c)}^{\theta} \leq \frac{18\gamma}{\mu}\sigma_{\star}^2 + \frac{27\gamma^2 HL}{\mu}\sigma_{\star}^2 + 36\gamma^2 H^2 L^2 C_{(c)}^{\theta} \leq \frac{21\gamma}{\mu}\sigma_{\star}^2 + \frac{1}{4}C_{(c)}^{\theta} \quad,$$

$$C_{(c)}^{\xi} \leq \frac{16L}{\mu H}\sigma_{\star}^2 + \frac{12 \cdot 28\gamma L^2}{\mu}\sigma_{\star}^2 + 36\gamma^2 H^2 L^2 C_{(c)}^{\xi} \leq \frac{40\mathrm{L}}{\mu\mathrm{H}}\sigma_{\star}^2 + \frac{1}{4}C_{(c)}^{\xi} \quad,$$

and the result follows. $\qquad\square$

## B.4. Higher-order bounds

We now derive bounds on the moments of the error, up to the sixth moment.

**Lemma B.3.** *Assume A1, A2 and A5. Let $\theta \in \mathbb{R}^d$ and $\{\xi_{(c)}\}_{c=1}^{N} \subset \mathbb{R}^d$ that satisfy the constraint $\sum_{c=1}^{N} \xi_{(c)} = 0$. Define the global iterate vector $\mathrm{X} = (\theta, \xi_{(1)}, \ldots, \xi_{(N)})$, and the optimal vector $\mathrm{X}^{\star} = (\theta^{\star}, \xi_{(1)}^{\star}, \ldots, \xi_{(N)}^{\star})$. Further, let $Z = Z_{(1:N)}^{1:H}$ be a collection of i.i.d. random variables such that for any $c \in \{1, \ldots, N\}$ and $h \in \{1, \ldots, H\}$, $Z_{(c)}^h \sim \nu_{(c)}$.*

*Assume the step size $\gamma$ and the number of local updates $H$ satisfy $\gamma L \leq 1/48$, $\gamma H(L + \mu) \leq 1/24$. Then,*

$$\mathbb{E}\left[\|\mathsf{S}(\mathrm{X}; Z) - \mathrm{X}^{\star}\|_{\Lambda}^6\right]^{1/3} \leq (1 - \gamma\mu/6)^H \|\mathrm{X} - \mathrm{X}^{\star}\|_{\Lambda}^2 + 40\gamma^2 H\sigma_{\star}^2 \quad. \tag{39}$$

*Proof.* We denote, for $\theta, \xi_{(1)}, \ldots, \xi_{(N)} \in \mathbb{R}^d$, notations for the global parameter update, the local parameter updates and the control variates updates as,

$$\theta^+ = \mathsf{T}(\theta; \xi_{(1:N)}, Z_{(1:N)}^{1:H}) \quad, \qquad \theta_{(c)}^h = \mathsf{T}_{(c)}^h(\theta; \xi_{(c)}, Z_{(c)}^{1:h}) \quad, \qquad \xi_{(c)}^+ = \mathsf{V}_{(c)}(\xi_{(c)}; \theta, Z_{(c)}^{1:H}) \quad,$$

for $c \in \{1, \ldots, N\}$ and $h \in \{0, \ldots, H\}$, as well as the shifted local parameters

$$\widetilde{\theta}_{(c)}^h = \theta_{(c)}^h + \gamma h(\xi_{(c)} - \xi_{(c)}^{\star}) \quad.$$

We recall the identity from (24),

$$\|\mathsf{S}(\mathrm{X}; Z) - \mathrm{X}^{\star}\|_{\Lambda}^6 = \left(\|\theta^+ - \theta^{\star}\|^2 + \frac{\gamma^2 H^2}{N}\sum_{c=1}^{N}\|\xi_{(c)} - \xi_{(c)}^{\star}\|^2\right)^3 = \left(\frac{1}{N}\sum_{c=1}^{N}\|\widetilde{\theta}_{(c)}^H - \theta^{\star}\|^2\right)^3 \quad.$$

Thus, using Hölder's inequality, we have

$$\mathbb{E}\left[\|\mathsf{S}(\mathrm{X}; Z) - \mathrm{X}^{\star}\|_{\Lambda}^6\right]^{1/3} \leq \frac{1}{N}\sum_{c=1}^{N}\mathbb{E}\left[\|\widetilde{\theta}_{(c)}^H - \theta^{\star}\|^6\right]^{1/3} \quad. \tag{40}$$

We proceed by induction. Expanding the second power, for $h \in \{0, \ldots, H-1\}$, using $\xi_{(c)}^{\star} = -\nabla f_{(c)}(\theta^{\star})$,

$$\|\widetilde{\theta}_{(c)}^{h+1} - \theta^{\star}\|^2 = \left\|\widetilde{\theta}_{(c)}^h - \theta^{\star} - \gamma\left(\nabla F_{(c)}^{Z_{(c)}^{h+1}}(\theta_{(c)}^h) - \nabla f_{(c)}(\theta^{\star})\right)\right\|^2$$

$$= \left\|\widetilde{\theta}_{(c)}^h - \theta^{\star}\right\|^2 - 2\gamma\left\langle\widetilde{\theta}_{(c)}^h - \theta^{\star}, \nabla F_{(c)}^{Z_{(c)}^{h+1}}(\theta_{(c)}^h) - \nabla f_{(c)}(\theta^{\star})\right\rangle + \gamma^2\left\|\nabla F_{(c)}^{Z_{(c)}^{h+1}}(\theta_{(c)}^h) - \nabla f_{(c)}(\theta^{\star})\right\|^2 \quad.$$

Now, we compute the third power of this equality. We write it as $(a^2 - 2\gamma b + \gamma^2 c^2)^3$, with

$$a^2 = \|\widetilde{\theta}^h_{(c)} - \theta^\star\|^2 \ ,$$

$$-2\gamma b = -2\gamma \left\langle \widetilde{\theta}^h_{(c)} - \theta^\star, \nabla F^{Z^{h+1}_{(c)}}_{(c)}(\theta^h_{(c)}) - \nabla f_{(c)}(\theta^\star) \right\rangle \ ,$$

$$\gamma^2 c^2 = \gamma^2 \left\| \nabla F^{Z^{h+1}_{(c)}}_{(c)}(\theta^h_{(c)}) - \nabla f_{(c)}(\theta^\star) \right\|^2 \ .$$

We remark that $|b| \leq ac$, which gives

$$
\begin{aligned}
\|\widetilde{\theta}^{h+1}_{(c)} - \theta^\star\|^6 &= \left(a^2 - 2\gamma b + \gamma^2 c^2\right)^3 \\
&= a^6 - 6\gamma a^4 b + 3\gamma^2 a^4 c^2 + 12\gamma^2 a^2 b^2 - 12\gamma^3 a^2 bc^2 + 3\gamma^4 a^2 c^4 - 8\gamma^3 b^3 + 12\gamma^4 b^2 c^2 - 6\gamma^5 bc^4 + \gamma^6 c^6 \\
&\leq a^6 - 6\gamma a^4 b + 3\gamma^2 a^4 c^2 + 12\gamma^2 a^4 c^2 + 12\gamma^3 a^3 c^3 + 3\gamma^4 a^2 c^4 + 8\gamma^3 a^3 c^3 + 12\gamma^4 a^2 c^4 + 6\gamma^5 ac^5 + \gamma^6 c^6 \\
&= a^6 - 6\gamma a^4 b + 15\gamma^2 a^4 c^2 + 20\gamma^3 a^3 c^3 + 15\gamma^4 a^2 c^4 + 6\gamma^5 ac^5 + \gamma^6 c^6 \ .
\end{aligned}
\tag{41}
$$

Remark that $a$ is $\sigma(Z^{1:h}_{(c)})$-measurable. Since $\widetilde{\theta}^h_{(c)} = \theta^h_{(c)} + \gamma h(\xi_{(c)} - \xi^\star_{(c)})$, we can split the dot product $b$ similarly to Lemma 4.3's proof, using Young's inequality to bound $\langle u, v \rangle \leq 1/6\|u\|^2 + 6\|v\|^2$ for any two vectors $u, v \in \mathbb{R}^d$,

$$
\begin{aligned}
\mathbb{E}\left[-6\gamma a^4 b \;\middle|\; Z^{1:h}_{(c)}\right] &= -6\gamma a^4 \langle \widetilde{\theta}^h_{(c)} - \theta^\star, \nabla f_{(c)}(\theta^h_{(c)}) - \nabla f_{(c)}(\theta^\star) \rangle \\
&= a^4 \left( -6\gamma \langle \theta^h_{(c)} - \theta^\star, \nabla f_{(c)}(\theta^h_{(c)}) - \nabla f_{(c)}(\theta^\star) \rangle - 6\gamma^2 h \langle \xi_{(c)} - \xi^\star_{(c)}, \nabla f_{(c)}(\theta^h_{(c)}) - \nabla f_{(c)}(\theta^\star) \rangle \right) \\
&\leq a^4 \left( -6\gamma \langle \theta^h_{(c)} - \theta^\star, \nabla f_{(c)}(\theta^h_{(c)}) - \nabla f_{(c)}(\theta^\star) \rangle + 36\gamma^3 h^2 L \|\xi_{(c)} - \xi^\star_{(c)}\|^2 + \frac{\gamma}{L} \|\nabla f_{(c)}(\theta^h_{(c)}) - \nabla f_{(c)}(\theta^\star)\|^2 \right) \ .
\end{aligned}
$$

Which gives, by co-coercivity of the gradient (10),

$$\mathbb{E}\left[-6\gamma a^4 b \;\middle|\; Z^{1:h}_{(c)}\right] \leq -5\gamma a^4 \langle \theta^h_{(c)} - \theta^\star, \nabla f_{(c)}(\theta^h_{(c)}) - \nabla f_{(c)}(\theta^\star) \rangle + 36\gamma^3 h^2 L a^4 \|\xi_{(c)} - \xi^\star_{(c)}\|^2 \ .$$

Furthermore, we have, by Lipschitzness of the gradient (12), and smoothness of the error noise (A5), and using the definition $\widetilde{\theta}^h_{(c)} = \theta^h_{(c)} + \gamma h(\xi_{(c)} - \xi^\star_{(c)})$, as well as the fact that $(x + y + z)^k \leq 3^{k-1}(x^k + y^k + z^k)$ for $2 \leq k \leq 6$,

$$
\begin{aligned}
\mathbb{E}\left[\gamma^k a^{6-k} c^k \;\middle|\; Z^{1:h}_{(c)}\right] &= \gamma^k a^{6-k} \mathbb{E}\left[\left\|\nabla F^{Z^{h+1}_{(c)}}_{(c)}(\theta^h_{(c)}) - \nabla f_{(c)}(\theta^\star)\right\|^k \;\middle|\; Z^{1:h}_{(c)}\right] \\
&\leq 2^{k-1}\gamma^k a^{6-k}\left\{\mathbb{E}\left[\left\|\nabla F^{Z^{h+1}_{(c)}}_{(c)}(\theta^h_{(c)}) - \nabla F^{Z^{h+1}_{(c)}}_{(c)}(\theta^\star)\right\|^k \;\middle|\; Z^{1:h}_{(c)}\right] + \sigma^k_\star\right\} \\
&\leq 2^{k-1}\gamma^k a^{6-k}\{2L^{k-1} a^{k-2} \langle \theta^h_{(c)} - \theta^\star, \nabla f_{(c)}(\theta^h_{(c)}) - \nabla f_{(c)}(\theta^\star) \rangle + \sigma^k_\star\} \ ,
\end{aligned}
\tag{42}
$$

where we used (12) to bound A2 and $\|\nabla F^{Z^{h+1}_{(c)}}_{(c)}(\widetilde{\theta}^h_{(c)}) - \nabla F^{Z^{h+1}_{(c)}}_{(c)}(\theta^\star)\| \leq L\|\widetilde{\theta}^h_{(c)} - \theta^\star\| = La$ in the last inequality. Taking the conditional expectation of (41) and plugging (42), we have

$$
\begin{aligned}
\mathbb{E}\left[\|\widetilde{\theta}^{h+1}_{(c)} - \theta^\star\|^6 \;\middle|\; Z^{1:h}_{(c)}\right] &= \mathbb{E}\left[a^6 - 6\gamma a^4 b + 15\gamma^2 a^4 c^2 + 20\gamma^3 a^3 c^3 + 15\gamma^4 a^2 c^4 + 6\gamma^5 ac^5 + \gamma^6 c^6 \;\middle|\; Z^{1:h}_{(c)}\right] \\
&\leq a^6 - 5\gamma a^4 \left\langle \theta^h_{(c)} - \theta^\star, \nabla f_{(c)}(\theta^h_{(c)}) - \nabla f_{(c)}(\theta^\star) \right\rangle + 36\gamma^3 h^2 L a^4 \left\|\xi_{(c)} - \xi^\star_{(c)}\right\|^2 \\
&\quad + 20a^4 \sum_{k=2}^{6}(2\gamma)^k L^{k-1}\left\langle \theta^h_{(c)} - \theta^\star, \nabla f_{(c)}(\theta^h_{(c)}) - \nabla f_{(c)}(\theta^\star) \right\rangle + 20\sum_{k=2}^{6}(2\gamma\sigma_\star)^k a^{6-k} \ .
\end{aligned}
$$

Letting $\gamma L \leq 1/40$ to bound the second term, we get

$$
\begin{aligned}
\mathbb{E}\left[\|\widetilde{\theta}^{h+1}_{(c)} - \theta^\star\|^6 \;\middle|\; Z^{1:h}_{(c)}\right] &\leq a^6 - \gamma a^4 \left\langle \theta^h_{(c)} - \theta^\star, \nabla f_{(c)}(\theta^h_{(c)}) - \nabla f_{(c)}(\theta^\star) \right\rangle \\
&\quad + 36\gamma^3 h^2 L a^4 \left\|\xi_{(c)} - \xi^\star_{(c)}\right\|^2 + 20\sum_{k=2}^{6}(2\gamma\sigma_\star)^k a^{6-k} \ .
\end{aligned}
$$

As in the second-order bound, we use the monotonicity of the gradient (11) to bound $-\gamma a^4 \left\langle \theta_{(c)}^h - \theta^\star, \nabla f_{(c)}(\theta_{(c)}^h) - \nabla f_{(c)}(\theta^\star) \right\rangle \leq -\gamma \mu a^4 \|\theta_{(c)}^h - \theta^\star\|^2$, which implies, using the fact that $-\|u\|^2 \leq -\frac{1}{2}\|u+v\|^2 + \|v\|^2$ for any pair of vectors $u, v \in \mathbb{R}^d$,

$$-\gamma a^4 \left\langle \theta_{(c)}^h - \theta^\star, \nabla f_{(c)}(\theta_{(c)}^h) - \nabla f_{(c)}(\theta^\star) \right\rangle \leq -\gamma \mu a^4/2 \|\widetilde{\theta}_{(c)}^h - \theta^\star\|^2 + \gamma^3 h^2 \mu a^4 \|\xi_{(c)} - \xi_{(c)}^\star\|^2 \ .$$

Finally, we obtain

$$\mathbb{E}\left[ \|\widetilde{\theta}_{(c)}^{h+1} - \theta^\star\|^6 \ \Big| \ Z_{(c)}^{1:h} \right] \leq (1 - \gamma\mu/2)a^6 + 36\gamma^3 h^2 (\mu+L) a^4 \left\| \xi_{(c)} - \xi_{(c)}^\star \right\|^2 + 20 \sum_{k=2}^{6} (2\gamma\sigma_\star)^k a^{6-k}$$

$$\leq (1 - \gamma\mu/2)a^6 + 36\gamma^3 h^2 (\mu+L) a^4 \left\| \xi_{(c)} - \xi_{(c)}^\star \right\|^2 + 30 \sum_{k=1}^{3} (2\gamma\sigma_\star)^{2k} a^{6-2k} \ , \quad (43)$$

using for $k$ odd, $(uv)^k \leq u^{k+1}v^{k-1}/2 + u^{k-1}v^{k+1}/2$. Using Hölder inequality, we have

$$\mathbb{E}[\|\widetilde{\theta}_{(c)}^{h+1} - \theta^\star\|^6] \leq (1 - \gamma\mu/2)\mathbb{E}[a^6]^{1/3} + 36\gamma^3 h^2 (\mu+L)\mathbb{E}[a^6]^{2/3}\mathbb{E}[\|\xi_{(c)} - \xi_{(c)}^\star\|^6]^{1/3} + 30\sum_{k=1}^{3}(2\gamma\sigma_\star)^{2k}\mathbb{E}[a^6]^{1-k/3} \ .$$

Therefore, we get

$$\mathbb{E}[\|\widetilde{\theta}_{(c)}^{h+1} - \theta^\star\|^6] \leq \left( (1 - \gamma\mu/2)^{1/3}\mathbb{E}[a^6]^{1/3} + 12\gamma^3 h^2(\mu+L)\mathbb{E}[\|\xi_{(c)} - \xi_{(c)}^\star\|^6]^{1/3} + 40\gamma^2\sigma_\star^2 \right)^3 \ . \quad (44)$$

Using $(1 - \gamma\mu/2)^{1/3} \leq 1 - \gamma\mu/6$ and a straightforward induction shows that

$$\mathbb{E}[\|\widetilde{\theta}_{(c)}^H - \theta^\star\|^6]^{1/3} \leq (1 - \gamma\mu/6)\|\theta - \theta^\star\|^2 + 12\gamma^3 H^3(\mu+L)\|\xi_{(c)} - \xi_{(c)}^\star\|^2 + 40H\gamma^2\sigma_\star^2 \ .$$

Using $(1 - \gamma\mu/6) \geq 1/2$ and $\gamma H(\mu+L) \leq 1/24$ completes the proof. $\qquad\square$

**Corollary B.4.** *Assume A1, A2 and A5. Let $\gamma > 0$ be the step size and $H > 0$ the number of local updates of SCAFFOLD. Assume that $\gamma L \leq 1/48$ and $\gamma H(L+\mu) \leq 1/24$. Then, for all $h \in \{0, \dots, H\}$, and $p \in \{1, 2, 3\}$, it holds that*

$$\left( \int \left\| \theta - \theta^\star \right\|^{2p} \pi_{(\gamma,H)}(\mathrm{d}\theta, \mathrm{d}\Xi) \right)^{1/p} \leq \frac{240\gamma}{\mu}\sigma_\star^2 \ , \quad (45)$$

$$\left( \frac{1}{N} \sum_{c=1}^{N} \int \mathbb{E}\left[ \left\| \mathsf{T}_{(c)}^h(\theta; \xi_{(c)}, Z_{(c)}^{1:h}) - \theta^\star \right\|^{2p} \right] \pi_{(\gamma,H)}(\mathrm{d}\theta, \mathrm{d}\Xi) \right)^{1/p} \leq \frac{240\gamma}{\mu}\sigma_\star^2 \ , \quad (46)$$

$$\left( \frac{\gamma^2 H^2}{N} \sum_{c=1}^{N} \int \left\| \xi_{(c)} - \xi_{(c)}^\star \right\|^{2p} \pi_{(\gamma,H)}(\mathrm{d}\theta, \mathrm{d}\Xi) \right)^{1/p} \leq \frac{240\gamma}{\mu}\sigma_\star^2 \ , \quad (47)$$

*where $\Xi = (\xi_{(1)}, \dots, \xi_{(N)}) \in \mathbb{R}^{N \times d}$.*

*Proof.* By Lemma B.3, we can bound the $\Lambda$-norm of the $T$-th element of the process $(\mathrm{X}^t)_{t=0}^\infty$, as

$$\mathbb{E}\left[ \|\mathrm{X}^T - \mathrm{X}^\star\|_\Lambda^6 \right]^{1/3} \leq (1 - \gamma\mu/6)^{HT} \|\mathrm{X}^0 - \mathrm{X}^\star\|_\Lambda^2 + \sum_{t=0}^{T-1} (1 - \gamma\mu/6)^{Ht} \cdot 40\gamma^2 H\sigma_\star^2 \ .$$

Taking the limit as $T \to \infty$, we obtain

$$\lim_{T \to \infty} \mathbb{E}\left[ \|\mathrm{X}^T - \mathrm{X}^\star\|_\Lambda^6 \right]^{1/3} \leq \frac{240\gamma H}{\mu}\sigma_\star^2 \ .$$

The result follows from derivations similar to the proof of Corollary B.1 to bound the third moment of $\|\mathrm{X} - \mathrm{X}^\star\|^2$ (i.e., the case $p = 3$). The result for $p = 1$ and $p = 2$ follows by Hölder's inequality. $\qquad\square$

**Lemma B.5.** *Assume A 1, A 2 and A 5. Let $\gamma > 0$ be the step size and $H > 0$ the number of local updates of SCAFFOLD. Assume that $\gamma L \leq 1/48$, $\gamma H(L + \mu) \leq 1/24$, $\gamma H^{1/2}\beta^{1/2} \leq 1/12$ and $\gamma\beta \leq L/12$. Then, for all $h \in \{0, \ldots, H\}$,*

$$\left(\int \|\mathsf{T}_{(c)}^h(\theta; \xi_{(c)}, Z_{(c)}^{1:h}) - \theta^\star\|^6 \pi_{(\gamma,H)}(\mathrm{d}\theta, \mathrm{d}\Xi)\right)^{1/3} \leq \frac{600\gamma}{\mu}\sigma_\star^2 \ , \tag{48}$$

$$\left(\int \|\xi_{(c)} - \xi_{(c)}^\star\|^6 \pi_{(\gamma,H)}(\mathrm{d}\theta, \mathrm{d}\Xi)\right)^{1/3} \leq \frac{3000L}{H\mu}\sigma_\star^2 \ . \tag{49}$$

*Proof.* The proof follows the same lines as Lemma 4.6.

**Bound on local iterates.** Let $\theta \in \mathbb{R}^d$ and $\{\xi_{(c)}\}_{c=1}^N \subset \mathbb{R}^d$ that satisfy the constraints $\sum_{c=1}^N \xi_{(c)} = 0$. To bound the local iterates, we proceed as in (34), we define $\theta_{(c)}^h = \mathsf{T}_{(c)}^h(\theta; \xi_{(c)}, Z_{(c)}^{1:h})$ and $\widetilde{\theta}_{(c)}^h = \theta_{(c)}^h + \gamma h(\xi_{(c)} - \xi_{(c)}^\star)$. Similarly to Lemma B.2, we use Jensen's inequality to bound

$$\mathbb{E}^{1/3}\left[\|\theta_{(c)}^h - \theta^\star\|^6\right] \leq \mathbb{E}^{1/3}\left[\|\widetilde{\theta}_{(c)}^h - \theta^\star\|^6\right] + \gamma^2 H^2\|\xi_{(c)} - \xi_{(c)}^\star\|^2 \ . \tag{50}$$

Then, unrolling (44) for $h$ steps and using the fact that $\gamma H(L + \mu) \leq 1/24$, we obtain, for any $h \leq H$, and $c \in \{1, \ldots, N\}$,

$$\mathbb{E}^{1/3}\left[\|\widetilde{\theta}_{(c)}^h - \theta^\star\|^6\right] \leq \|\theta - \theta^\star\|^6 + \frac{\gamma^2 H^2}{2}\|\xi_{(c)} - \xi_{(c)}^\star\|^2 + 40\gamma^2 H\sigma_\star^2 \ . \tag{51}$$

Plugging (51) in (50), we obtain

$$\mathbb{E}^{1/3}\left[\|\theta_{(c)}^h - \theta^\star\|^6\right] \leq \|\theta - \theta^\star\|^2 + \frac{3\gamma^2 H^2}{2}\|\xi_{(c)} - \xi_{(c)}^\star\|^2 + 40\gamma^2 H\sigma_\star^2 \ . \tag{52}$$

Taking the third power of this inequality, integrating it over the stationary distribution of SCAFFOLD's iterates and using Corollary B.4, and using Jensen's inequality, we obtain

$$\int \mathbb{E}\left[\|\theta_{(c)}^h - \theta^\star\|^6\right] \pi_{(\gamma,H)}(\mathrm{d}\theta, \mathrm{d}\Xi) \leq \int \left(3^2\mathbb{E}\left[\|\theta - \theta^\star\|^6\right] + 8\gamma^6 H^6\|\xi_{(c)} - \xi_{(c)}^\star\|^6 + 3^2 \cdot 40^3 \cdot \gamma^6 H^3 \sigma_\star^6\right)\pi_{(\gamma,H)}(\mathrm{d}\theta, \mathrm{d}\Xi)$$

$$\leq 8\gamma^6 H^6 \int \|\xi_{(c)} - \xi_{(c)}^\star\|^6 \pi_{(\gamma,H)}(\mathrm{d}\theta, \mathrm{d}\Xi) + \frac{3^2 \cdot (240^3 + 1) \cdot \gamma^3}{\mu^3}\sigma_\star^6 \ , \tag{53}$$

where we used $\gamma L \leq 1/48$ and $1/L \leq 1/\mu$ to bound $40^3\gamma^3 \leq 1/\mu^3$.

**Bound on control variates.** To derive the second inequality, we start from (38),

$$\xi_{(c)}^+ - \xi_{(c)}^\star = \frac{1}{NH}\sum_{i=1}^N\sum_{h=0}^{H-1}\left(\left(\nabla f_{(i)}(\theta_{(i)}^h) - \nabla f_{(i)}(\theta^\star)\right) - \left(\nabla f_{(c)}(\theta_{(c)}^h) - \nabla f_{(c)}(\theta^\star)\right) + \varepsilon_{(i)}^{h+1} - \varepsilon_{(c)}^{h+1}\right) \ .$$

Using Jensen's inequality, we obtain

$$\mathbb{E}^{1/3}\left[\|\xi_{(c)}^+ - \xi_{(c)}^\star\|^6\right] \leq 2\mathbb{E}^{1/3}\left[\left\|\frac{1}{NH}\sum_{i=1}^N\sum_{h=0}^{H-1}\left(\nabla f_{(i)}(\theta_{(i)}^h) - \nabla f_{(i)}(\theta^\star)\right) - \left(\nabla f_{(c)}(\theta_{(c)}^h) - \nabla f_{(c)}(\theta^\star)\right)\right\|^6\right]$$

$$+ 4\mathbb{E}^{1/3}\left[\left\|\frac{1}{NH}\sum_{i=1}^N\sum_{h=0}^{H-1}\varepsilon_{(i)}^{h+1}\right\|^6\right] + 4\mathbb{E}^{1/3}\left[\left\|\frac{1}{H}\sum_{h=0}^{H-1}\varepsilon_{(c)}^{h+1}\right\|^6\right] \ .$$

To control the last two terms, we note that they are reverse martingale differences w.r.t. the filtration $\mathcal{F}^h = \sigma(Z_{(1:N)}^{1:h})$. By Burkholder's inequality (see, e.g., Osekowski (2012), Theorem 8.6) which holds due to A 5, we have

$$\mathbb{E}^{1/3}\left[\left\|\frac{1}{NH}\sum_{i=1}^N\sum_{h=0}^{H-1}\varepsilon_{(i)}^{h+1}\right\|^6\right] \leq \frac{3^2}{N^2H^2}\mathbb{E}^{1/3}\left[\left(\sum_{i=1}^N\sum_{h=0}^{H-1}\|\varepsilon_{(i)}^{h+1}\|^2\right)^3\right] \leq \frac{3^2}{N^2H^2}\sum_{i=1}^N\sum_{h=0}^{H-1}\mathbb{E}^{1/3}\left[\|\varepsilon_{(i)}^{h+1}\|^6\right] \ .$$

Using the smoothness of the error noise's moments (A5), we thus obtain

$$\mathbb{E}^{1/3}\left[\left\|\frac{1}{NH}\sum_{i=1}^{N}\sum_{h=0}^{H-1}\varepsilon_{(i)}^{h+1}\right\|^6\right] \leq \frac{3^2}{N^2H^2}\sum_{i=1}^{N}\sum_{h=0}^{H-1}\beta\mathbb{E}\left[\|\theta_{(i)}^h - \theta^\star\|^6\right]^{1/3} + \sigma_\star^2 \ .$$

Using Jensen's inequality again, and proceeding as in Lemma 4.6's proof using Lipschitzness of the gradient (12), we have

$$\mathbb{E}^{1/3}\left[\|\xi_{(c)}^+ - \xi_{(c)}^\star\|^6\right] \leq \frac{4L^2}{NH}\sum_{i=1}^{N}\sum_{h=0}^{H-1}\mathbb{E}^{1/3}\left[\|\theta_{(i)}^h - \theta^\star\|^6\right] + \mathbb{E}^{1/3}\left[\|\theta_{(c)}^h - \theta^\star\|^6\right]$$

$$+ \frac{4\cdot 3^2}{N^2H^2}\sum_{i=1}^{N}\sum_{h=0}^{H-1}\left\{\beta\mathbb{E}^{1/3}\left[\|\theta_{(i)}^h - \theta^\star\|^6\right] + \sigma_\star^2\right\} + \frac{4\cdot 3^2}{H^2}\sum_{h=0}^{H-1}\left\{\beta\mathbb{E}^{1/3}\left[\|\theta_{(c)}^h - \theta^\star\|^6\right] + \sigma_\star^2\right\}$$

$$\leq \frac{8\cdot 3^2}{H}\sigma_\star^2 + \frac{4L^2 + 4\cdot 3^2\beta}{NH^2}\sum_{i=1}^{N}\sum_{h=0}^{H-1}\left\{\mathbb{E}^{1/3}\left[\|\theta_{(i)}^h - \theta^\star\|^6\right] + \mathbb{E}^{1/3}\left[\|\theta_{(c)}^h - \theta^\star\|^6\right]\right\} \ . \tag{54}$$

Plugging (52) in (54), we obtain

$$\mathbb{E}^{1/3}\left[\|\xi_{(c)}^+ - \xi_{(c)}^\star\|^6\right] \leq \frac{72}{H}\sigma_\star^2 + \frac{8L^2 + 72\beta}{H}\left(\|\theta - \theta^\star\|^2 + \frac{3\gamma^2H^2}{2}\|\xi_{(c)} - \xi_{(c)}^\star\|^2 + 40\gamma^2H\sigma_\star^2\right)$$

$$= \frac{72}{H}\sigma_\star^2 + (40\cdot 8L^2 + 40\cdot 72\beta)\gamma^2\sigma_\star^2 + \frac{8L^2 + 72\beta}{H}\|\theta - \theta^\star\|^2 + (3\cdot 4L^2 + 3\cdot 36\beta)\gamma^2H\|\xi_{(c)} - \xi_{(c)}^\star\|^2$$

$$\leq \frac{72}{H}\sigma_\star^2 + \frac{1+5}{H}\sigma_\star^2 + \frac{8L^2 + 72\beta}{H}\|\theta - \theta^\star\|^2 + \left(\frac{1}{96} + \frac{3}{16}\right)\|\xi_{(c)} - \xi_{(c)}^\star\|^2 \ , \tag{55}$$

where we used $\gamma L \leq 1/48$, $\gamma(L+\mu)H \leq 1/24$ and $\gamma H^{1/2}\beta^{1/2} \leq 1/12$. Remark that $1/96 + 3/16 \leq 1/5$. Taking the third power of (55) and using Jensen's inequality, we obtain

$$\mathbb{E}\left[\|\xi_{(c)}^+ - \xi_{(c)}^\star\|^6\right] \leq \frac{3^2\cdot 78^3}{H^3}\sigma_\star^6 + 3^2\cdot\left(\frac{8L^2 + 72\beta}{H}\right)^3\|\theta - \theta^\star\|^6 + \frac{3^2}{5^3}\|\xi_{(c)} - \xi_{(c)}^\star\|^6 \ .$$

Integrating over $\pi_{(\gamma,H)}$, remarking that $\int\mathbb{E}\left[\|\xi_{(c)}^+ - \xi_{(c)}^\star\|^6\right]\pi_{(\gamma,H)}(\mathrm{d}\theta,\mathrm{d}\Xi) = \int\|\xi_{(c)} - \xi_{(c)}^\star\|^6\pi_{(\gamma,H)}(\mathrm{d}\theta,\mathrm{d}\Xi)$, using the fact that $3^2/5^3 \leq 1/10$, and multiplying the resulting inequality by $10/9$, we obtain

$$\int\|\xi_{(c)} - \xi_{(c)}^\star\|^6\pi_{(\gamma,H)}(\mathrm{d}\theta,\mathrm{d}\Xi) \leq \frac{10\cdot 78^3}{H^3}\sigma_\star^6 + 10\left(\frac{8L^2 + 72\beta}{H}\right)^3\int\|\theta - \theta^\star\|^6\pi_{(\gamma,H)}(\mathrm{d}\theta,\mathrm{d}\Xi)$$

$$\leq \frac{10\cdot 78^3}{H^3}\sigma_\star^6 + 10\frac{(8L^2 + 72\beta)^3}{H^3}\cdot\frac{240^3\gamma^3}{\mu^3}\sigma_\star^6 \leq \frac{10\cdot 78^3}{H^3}\sigma_\star^6 + 90\frac{8^3L^6 + 72^3\beta^3}{H^3}\cdot\frac{240^3\gamma^3}{\mu^3}\sigma_\star^6 \leq \frac{3000^3L^3}{\mu^3H^3}\sigma_\star^6 \ , \tag{56}$$

where the last inequality follows from $\gamma L \leq 1/48$, $\gamma\beta^{1/2}H^{1/2} \leq 1/12$ and $\gamma\beta \leq L/12$.

**Final bound on the local iterates.** From (53) and (56), we have

$$\int\mathbb{E}\left[\|\theta_{(c)}^h - \theta^\star\|^6\right]\pi_{(\gamma,H)}(\mathrm{d}\theta,\mathrm{d}\Xi) \leq 8\gamma^6H^6\int\|\xi_{(c)} - \xi_{(c)}^\star\|^6\pi_{(\gamma,H)}(\mathrm{d}\theta,\mathrm{d}\Xi) + \frac{3^2\cdot(240^3+1)\cdot\gamma^3}{\mu^3}\sigma_\star^6$$

$$\leq \frac{8\cdot 3000^3\gamma^6H^3L^3}{\mu^3}\sigma_\star^6 + \frac{3^2\cdot(240^3+1)\cdot\gamma^3}{\mu^3}\sigma_\star^6 \ ,$$

and the result follows from $\gamma HL \leq 1/24$, which ensures that $8\cdot 3000^3\gamma^3H^3L^3 + 3^2\cdot(240^3+1) \leq 600^3$. $\square$

## C. Bounding the Variance of SCAFFOLD

We now study the bias of the SCAFFOLD algorithm. Let $X = (\theta, \xi_{(1)}, \ldots, \xi_{(N)})$, where the global parameter and control variates are $\theta, \xi_{(1)}, \ldots, \xi_{(N)}$ is a vector in $\mathbb{R}^{(N+1)d}$ drawn from the stationary distribution $\pi_{(\gamma,H)}$. To study its expected value, we use the fact that, by definition, the $\mathsf{S}(X;Z)$ has the same distribution as $X$.

**Notations.** For $\theta \in \mathbb{R}^d$ and $\Xi = (\xi_{(1)}, \ldots, \xi_{(N)}) \in \mathbb{R}^{N \times d}$, we define the variances and covariances of parameters and control variates in the stationary distribution $\pi_{(\gamma, H)}$ as

$$\bar{\Sigma}^\theta \triangleq \int (\theta - \theta^\star)^{\otimes 2} \, \pi_{(\gamma, H)}(\mathrm{d}\theta, \mathrm{d}\Xi) \ ,$$

$$\bar{\Sigma}^\xi_{(c, c')} \triangleq \int \left(\xi_{(c)} - \xi^\star_{(c)}\right) \left(\xi_{(c')} - \xi^\star_{(c')}\right)^\top \pi_{(\gamma, H)}(\mathrm{d}\theta, \mathrm{d}\Xi) \ ,$$

$$\bar{\Sigma}^{\theta, \xi}_{(c)} \triangleq \int \left(\theta - \theta^\star\right) \left(\xi_{(c)} - \xi^\star_{(c)}\right)^\top \pi_{(\gamma, H)}(\mathrm{d}\theta, \mathrm{d}\Xi) \ ,$$

$$\bar{\Sigma}^{\xi, \theta}_{(c)} \triangleq \int \left(\xi_{(c)} - \xi^\star_{(c)}\right) \left(\theta - \theta^\star\right)^\top \pi_{(\gamma, H)}(\mathrm{d}\theta, \mathrm{d}\Xi) \ .$$

In the following, we use the following matrices and tensor, that appear in the integral remainders of our expansions

$$\bar{D}^{2,h}_{(c)}(\theta) = \int_0^1 \nabla^2 f_{(c)}(\theta^\star + t \left(\mathsf{T}^h_{(c)}(\theta; \xi_{(c)}, Z^{1:h}_{(c)}) - \theta^\star\right))\mathrm{d}t \ , \tag{57}$$

$$\bar{D}^{3,h}_{(c)}(\theta) = \int_0^1 (1 - t)\nabla^3 f_{(c)}(\theta^\star + t \left(\mathsf{T}^h_{(c)}(\theta; \xi_{(c)}, Z^{1:h}_{(c)}) - \theta^\star\right))\mathrm{d}t \ . \tag{58}$$

For conciseness, we will often use the abbreviated notations

$$\bar{D}^{2,h}_{(c)} := \bar{D}^{2,h}_{(c)}(\theta^h_{(c)}) \quad \text{and} \quad \bar{D}^{3,h}_{(c)} := \bar{D}^{3,h}_{(c)}(\theta^h_{(c)}) \ . \tag{59}$$

Following an update step of the SCAFFOLD algorithm, we obtain their updated counterparts, which reflect the adjustments made during this iteration.

$$\theta^+ = \mathsf{T}(\theta; \xi_{(1:N)}, Z^{1:H}_{(1:N)}) \ , \qquad \theta^h_{(c)} = \mathsf{T}^h_{(c)}(\theta; \xi_{(c)}, Z^{1:h}_{(c)}) \ , \qquad \xi^+_{(c)} = \mathsf{V}_{(c)}(\xi_{(c)}; \theta, Z^{1:H}_{(c)}) \ , \tag{60}$$

for $h \in \{0, \ldots, H\}$ and $c \in \{1, \ldots, N\}$. We define the noise accumulated in one round, with $\varepsilon^h_{(c)}$ as defined in (36).

$$\varepsilon^{1:H}_{(c)} = \sum_{h=1}^H \Gamma^{H-h}_{(c)} \varepsilon^h_{(c)} \ .$$

**Matrix notations.** We define the contraction matrix $\Gamma_{(c)} = \mathrm{Id} - \gamma \nabla^2 f_{(c)}(\theta^\star)$, as well as its powers, for $h \in \{0, \ldots, H\}$, average, and scaled difference between the local matrices and their average,

$$\Gamma^h_{(c)} = \left(\mathrm{Id} - \gamma \nabla^2 f_{(c)}(\theta^\star)\right)^h \ , \quad \bar{\Gamma} = \frac{1}{N} \sum_{c=1}^N \Gamma^H_{(c)} \ , \quad \Delta^\Gamma_{(c)} = \frac{1}{\gamma H}\left(\Gamma_{(c)} - \bar{\Gamma}\right) \ . \tag{61}$$

Finally, we define

$$\mathsf{C}^{1:H}_{(c)} = -\frac{1}{H} \sum_{h=0}^{H-1} \Gamma^{H-h-1}_{(c)} \ , \quad \widetilde{\mathsf{C}}^{1:H}_{(c)} = \mathrm{Id} - \frac{1}{H} \sum_{h=0}^{H-1} \Gamma^{H-h-1}_{(c)} \ , \quad \mathcal{R}^{1:H}_{(c)} = \sum_{h=0}^{H-1} \Gamma^{H-h-1}_{(c)} \bar{D}^{3,h}_{(c)} \left(\theta^h_{(c)} - \theta^\star\right)^{\otimes 2} \ . \tag{62}$$

## C.1. Expansions of local updates and control variates

First, we give explicit expansions of the local and global parameter updates.

**Lemma C.1.** *Let $\theta \in \mathbb{R}^d$ and $\Xi = (\xi_{(1)}, \ldots, \xi_{(N)}) \in \mathbb{R}^{N \times d}$. After one global update of SCAFFOLD, we obtain a global parameter $\theta^+$, $N$ control variates $\xi^+_{(c)}$ and $N \cdot H$ local iterates $\theta^h_{(c)}$ as defined in (60). These updates parameters can be expressed as*

$$\theta^H_{(c)} - \theta^\star = \Gamma^H_{(c)} (\theta - \theta^\star) + \gamma H \mathsf{C}^{1:H}_{(c)} \left(\xi_{(c)} - \xi^\star_{(c)}\right) - \gamma \mathcal{R}^{1:H}_{(c)} - \gamma \varepsilon^{1:H}_{(c)} \ , \tag{63}$$

$$\theta^+ - \theta^\star = \bar{\Gamma} (\theta - \theta^\star) + \frac{\gamma H}{N} \sum_{c=1}^N \widetilde{\mathsf{C}}^{1:H}_{(c)} \left(\xi_{(c)} - \xi^\star_{(c)}\right) - \frac{\gamma}{N} \sum_{c=1}^N \mathcal{R}^{1:H}_{(c)} - \frac{\gamma}{N} \sum_{c=1}^N \varepsilon^{1:H}_{(c)} \ . \tag{64}$$

*Proof.* Let $c \in \{1, \ldots, N\}$ and $h \in \{0, \ldots, H-1\}$. Expanding the gradient at step $h$ gives

$$\theta_{(c)}^{h+1} = \theta_{(c)}^h - \gamma \left( \nabla f_{(c)}(\theta_{(c)}^h) + \xi_{(c)} + \varepsilon_{(c)}^{h+1} \right)$$

$$= \theta_{(c)}^h - \gamma \left( \nabla f_{(c)}(\theta^\star) + \nabla^2 f_{(c)}(\theta^\star) \left( \theta_{(c)}^h - \theta^\star \right) + \bar{D}_{(c)}^{3,h+1} \left( \theta_{(c)}^h - \theta^\star \right)^{\otimes 2} + \xi_{(c)} + \varepsilon_{(c)}^{h+1} \right) \quad . \tag{65}$$

Since $\xi_{(c)}^\star = -\nabla f_{(c)}(\theta^\star)$, we obtain

$$\theta_{(c)}^{h+1} - \theta^\star = \theta_{(c)}^h - \theta^\star - \gamma \nabla^2 f_{(c)}(\theta^\star) \left( \theta_{(c)}^h - \theta^\star \right) - \gamma \left( \xi_{(c)} - \xi_{(c)}^\star \right) - \gamma \bar{D}_{(c)}^{3,h+1} \left( \theta_{(c)}^h - \theta^\star \right)^{\otimes 2} - \gamma \varepsilon_{(c)}^{h+1}$$

$$= \underbrace{\left( \mathrm{Id} - \gamma \nabla^2 f_{(c)}(\theta^\star) \right)}_{\Gamma_{(c)}} \left( \theta_{(c)}^h - \theta^\star \right) - \gamma \left( \xi_{(c)} - \xi_{(c)}^\star \right) - \gamma \bar{D}_{(c)}^{3,h} \left( \theta_{(c)}^h - \theta^\star \right)^{\otimes 2} - \gamma \varepsilon_{(c)}^{h+1} \quad .$$

We obtain the following expression for the local updates

$$\theta_{(c)}^H - \theta^\star = \Gamma_{(c)}^H (\theta - \theta^\star) - \gamma \sum_{h=0}^{H-1} \Gamma_{(c)}^{H-h-1} \left( \xi_{(c)} - \xi_{(c)}^\star \right) - \gamma \sum_{h=0}^{H-1} \Gamma_{(c)}^{H-h-1} \bar{D}_{(c)}^{3,h} \left( \theta_{(c)}^h - \theta^\star \right)^{\otimes 2} - \gamma \sum_{h=0}^{H-1} \Gamma_{(c)}^{H-h-1} \varepsilon_{(c)}^{h+1}$$

$$= \Gamma_{(c)}^H (\theta - \theta^\star) + \gamma H \mathrm{C}_{(c)}^{1:H} \left( \xi_{(c)} - \xi_{(c)}^\star \right) - \gamma \mathcal{R}_{(c)}^{1:H} - \gamma \varepsilon_{(c)}^{1:H} \quad ,$$

which gives the first identity (63). The second identity (64) follows from averaging the first one over all clients and using the fact that

$$\frac{1}{N} \sum_{c=1}^N \mathrm{C}_{(c)}^{1:H} \left( \xi_{(c)} - \xi_{(c)}^\star \right) = \frac{1}{N} \sum_{c=1}^N \widetilde{\mathrm{C}}_{(c)}^{1:H} \left( \xi_{(c)} - \xi_{(c)}^\star \right) \quad , \tag{66}$$

which follows from $\sum_{c=1}^N \xi_{(c)} - \xi_{(c)}^\star = 0$. $\qquad \square$

Based on Lemma C.1, we can give an expression for the control variate updates.

**Lemma C.2.** *Let $\theta \in \mathbb{R}^d$ and $\Xi = (\xi_{(1)}, \ldots, \xi_{(N)}) \in \mathbb{R}^{N \times d}$. After one global update of Scaffold, we obtain a global parameter $\theta^+$, $N$ control variates $\xi_{(c)}^+$ and $N \cdot H$ local iterates $\theta_{(c)}^h$ as defined in (60). The updated control variates can be expressed as*

$$\xi_{(c)}^+ - \xi_{(c)}^\star = \Delta_{(c)}^\Gamma (\theta - \theta^\star) + \widetilde{\mathrm{C}}_{(c)}^{1:H} \left( \xi_{(c)} - \xi_{(c)}^\star \right) - \frac{1}{N} \sum_{i=1}^N \widetilde{\mathrm{C}}_{(i)}^{1:H} \left( \xi_{(i)} - \xi_{(i)}^\star \right)$$

$$- \frac{1}{H} \mathcal{R}_{(c)}^{1:H} + \frac{1}{NH} \sum_{i=1}^N \mathcal{R}_{(i)}^{1:H} - \frac{1}{H} \varepsilon_{(c)}^{1:H} + \frac{1}{NH} \sum_{i=1}^N \varepsilon_{(i)}^{1:H} \quad . \tag{67}$$

*where $\mathrm{C}_{(c)}^{1:H}$, $\widetilde{\mathrm{C}}_{(c)}^{1:H}$, $\mathcal{R}_{(c)}^{1:H}$, and $\varepsilon_{(c)}^{1:H}$ are defined in (62).*

*Proof.* Let $c \in \{1, \ldots, N\}$, $\xi_{(c)}$ is updated as $\xi_{(c)}^+ = \xi_{(c)} + \frac{1}{\gamma H} \left( \theta_{(c)}^H - \theta^+ \right)$, which gives

$$\xi_{(c)}^+ = \xi_{(c)} + \frac{1}{\gamma H} \left( \Gamma_{(c)}^H - \bar{\Gamma} \right) (\theta - \theta^\star) + \mathrm{C}_{(c)}^{1:H} \left( \xi_{(c)} - \xi_{(c)}^\star \right) + \frac{1}{N} \sum_{i=1}^N \mathrm{C}_{(i)}^{1:H} \left( \xi_{(i)} - \xi_{(i)}^\star \right)$$

$$- \frac{1}{H} \mathcal{R}_{(c)}^{1:H} + \frac{1}{NH} \sum_{i=1}^N \mathcal{R}_{(i)}^{1:H} - \frac{1}{H} \varepsilon_{(c)}^{1:H} + \frac{1}{NH} \sum_{i=1}^N \varepsilon_{(i)}^{1:H}$$

$$= \xi_{(c)}^\star + \xi_{(c)} - \xi_{(c)}^\star + \Delta_{(c)}^\Gamma (\theta - \theta^\star) + \mathrm{C}_{(c)}^{1:H} \left( \xi_{(c)} - \xi_{(c)}^\star \right) + \frac{1}{N} \sum_{i=1}^N \mathrm{C}_{(i)}^{1:H} \left( \xi_{(i)} - \xi_{(i)}^\star \right)$$

$$- \frac{1}{H} \mathcal{R}_{(c)}^{1:H} + \frac{1}{NH} \sum_{i=1}^N \mathcal{R}_{(i)}^{1:H} - \frac{1}{H} \varepsilon_{(c)}^{1:H} + \frac{1}{NH} \sum_{i=1}^N \varepsilon_{(i)}^{1:H} \quad .$$

Then, remark that $\xi_{(c)} - \xi_{(c)}^\star + \mathrm{C}_{(c)}^{1:H}(\xi_{(c)} - \xi_{(c)}^\star) = \widetilde{\mathrm{C}}_{(c)}^{1:H}(\xi_{(c)} - \xi_{(c)}^\star)$ since $\widetilde{\mathrm{C}}_{(c)}^{1:H} = \mathrm{Id} + \mathrm{C}_{(c)}^{1:H}$. $\qquad \square$

## C.2. Covariance of the Parameters and Control Variates

### C.2.1. RECURSION ON COVARIANCE MATRICES

**Lemma C.3.** *Assume A1, A2 and A5. Assume the step size $\gamma$ and the number of local updates $H$ satisfy $\gamma H(L + \mu) \leq 1$. Then, it holds that*

$$\bar{\Sigma}^{\theta} = \bar{\Gamma}\bar{\Sigma}^{\theta}\bar{\Gamma} + \frac{\gamma H}{N}\sum_{c=1}^{N}\left(\bar{\Gamma}\bar{\Sigma}_{(c)}^{\theta,\xi}\widetilde{C}_{(c)}^{1:H} + \widetilde{C}_{(c)}^{1:H}\bar{\Sigma}_{(c)}^{\xi,\theta}\bar{\Gamma}\right) + \frac{\gamma^2 H^2}{N^2}\sum_{c=1}^{N}\sum_{c'=1}^{N}\widetilde{C}_{(c)}^{1:H}\bar{\Sigma}_{(c,c')}^{\xi}\widetilde{C}_{(c')}^{1:H} + \frac{\gamma^2}{N}\bar{\Sigma}^{\epsilon} + R^{\theta}\ ,$$

*where $\bar{\Sigma}^{\epsilon} = \frac{1}{N}\sum_{c=1}^{N}\mathbb{E}\left[(\varepsilon_{(c)}^{1:H})^{\otimes 2}\right]$, and $R^{\theta} = R_1^{\theta} + R_1^{\theta\top} + R_2^{\theta} + R_2^{\theta\top} + R_3^{\theta}$, with*

$$R_1^{\theta} = \frac{\gamma^2}{N^2}\sum_{c=1}^{N}\int \mathbb{E}\left[\left(\varepsilon_{(c)}^{1:H}\right)\left(\mathcal{R}_{(c)}^{1:H}\right)^{\top}\right]\pi_{(\gamma,H)}(\mathrm{d}\theta,\mathrm{d}\Xi)\ ,$$

$$R_2^{\theta} = -\frac{\gamma}{N}\sum_{c=1}^{N}\int \mathbb{E}\left[\mathcal{R}_{(c)}^{1:H}\right](\theta - \theta^{\star})^{\top}\bar{\Gamma}\pi_{(\gamma,H)}(\mathrm{d}\theta,\mathrm{d}\Xi)$$

$$-\frac{\gamma^2 H}{N^2}\sum_{c=1}^{N}\sum_{c'=1}^{N}\int \mathbb{E}\left[\mathcal{R}_{(c)}^{1:H}\right]\left(\xi_{(c')} - \xi_{(c')}^{\star}\right)^{\top}\widetilde{C}_{(c')}^{1:H}\pi_{(\gamma,H)}(\mathrm{d}\theta,\mathrm{d}\Xi)\ ,$$

$$R_3^{\theta} = \frac{\gamma^2}{N^2}\sum_{c=1}^{N}\sum_{c'=1}^{N}\int \mathbb{E}\left[\left(\mathcal{R}_{(c)}^{1:H}\right)\left(\mathcal{R}_{(c')}^{1:H}\right)^{\top}\right]\pi_{(\gamma,H)}(\mathrm{d}\theta,\mathrm{d}\Xi)\ .$$

*Proof.* Using the results from Lemma C.1, we have

$$(\theta^+ - \theta^{\star})^{\otimes 2} = \left(\bar{\Gamma}(\theta - \theta^{\star}) + \frac{\gamma H}{N}\sum_{c=1}^{N}\widetilde{C}_{(c)}^{1:H}\left(\xi_{(c)} - \xi_{(c)}^{\star}\right) - \frac{\gamma}{N}\sum_{c=1}^{N}\mathcal{R}_{(c)}^{1:H}\right)^{\otimes 2} + \frac{\gamma^2}{N^2}\left(\sum_{c=1}^{N}\varepsilon_{(c)}^{1:H}\right)^{\otimes 2}$$

$$-\frac{\gamma}{N}\sum_{c=1}^{N}\varepsilon_{(c)}^{1:H}\left(\bar{\Gamma}(\theta - \theta^{\star}) + \frac{\gamma H}{N}\sum_{c'=1}^{N}\widetilde{C}_{(c')}^{1:H}\left(\xi_{(c')} - \xi_{(c')}^{\star}\right) - \frac{\gamma}{N}\sum_{c'=1}^{N}\mathcal{R}_{(c')}^{1:H}\right)^{\top}$$

$$-\frac{\gamma}{N}\sum_{c=1}^{N}\left(\bar{\Gamma}(\theta - \theta^{\star}) + \frac{\gamma H}{N}\sum_{c'=1}^{N}\widetilde{C}_{(c')}^{1:H}\left(\xi_{(c')} - \xi_{(c')}^{\star}\right) - \frac{\gamma}{N}\sum_{c'=1}^{N}\mathcal{R}_{(c')}^{1:H}\right)^{\otimes 2}\left(\varepsilon_{(c)}^{1:H}\right)^{\top}\ .$$

Taking the expectation, and using the fact that the $Z_{(c)}$ are independent from one client to another, we obtain

$$\mathbb{E}\left[(\theta^+ - \theta^{\star})^{\otimes 2}\right] = \mathbb{E}\left[\left(\bar{\Gamma}(\theta - \theta^{\star}) + \frac{\gamma H}{N}\sum_{c=1}^{N}\widetilde{C}_{(c)}^{1:H}\left(\xi_{(c)} - \xi_{(c)}^{\star}\right) - \frac{\gamma}{N}\sum_{c=1}^{N}\mathcal{R}_{(c)}^{1:H}\right)^{\otimes 2}\right]$$

$$+\frac{\gamma^2}{N^2}\sum_{c=1}^{N}\mathbb{E}\left[\left(\varepsilon_{(c)}^{1:H}\right)^{\otimes 2}\right] + \frac{\gamma^2}{N^2}\sum_{c=1}^{N}\mathbb{E}\left[\varepsilon_{(c)}^{1:H}\left(\mathcal{R}_{(c)}^{1:H}\right)^{\top} + \mathcal{R}_{(c)}^{1:H}\left(\varepsilon_{(c)}^{1:H}\right)^{\top}\right]\ .$$

The first term can be expressed using the identity

$$\mathbb{E}\left[\left(\bar{\Gamma}(\theta - \theta^{\star}) + \frac{\gamma H}{N}\sum_{c=1}^{N}\widetilde{C}_{(c)}^{1:H}\left(\xi_{(c)} - \xi_{(c)}^{\star}\right) - \frac{\gamma}{N}\sum_{c=1}^{N}\mathcal{R}_{(c)}^{1:H}\right)^{\otimes 2}\right]$$

$$= \left(\bar{\Gamma}(\theta - \theta^{\star}) + \frac{\gamma H}{N}\sum_{c=1}^{N}\widetilde{C}_{(c)}^{1:H}\left(\xi_{(c)} - \xi_{(c)}^{\star}\right)\right)^{\otimes 2} + \frac{\gamma^2}{N^2}\mathbb{E}\left[\left(\sum_{c=1}^{N}\mathcal{R}_{(c)}^{1:H}\right)^{\otimes 2}\right]$$

$$-\frac{\gamma}{N}\sum_{c=1}^{N}\mathbb{E}\left[\mathcal{R}_{(c)}^{1:H}\right]\left(\bar{\Gamma}(\theta - \theta^{\star}) + \frac{\gamma H}{N}\sum_{c'=1}^{N}\widetilde{C}_{(c')}^{1:H}\left(\xi_{(c')} - \xi_{(c')}^{\star}\right)\right)^{\top}$$

$$- \frac{\gamma}{N} \sum_{c=1}^{N} \left( \bar{\Gamma} \left( \theta - \theta^{\star} \right) + \frac{\gamma H}{N} \sum_{c'=1}^{N} \widetilde{C}_{(c')}^{1:H} \left( \xi_{(c')} - \xi_{(c')}^{\star} \right) \right) \mathbb{E} \left[ \left( \mathcal{R}_{(c')}^{1:H} \right)^{\top} \right] \quad .$$

The first term can be expanded as

$$\left( \bar{\Gamma} \left( \theta - \theta^{\star} \right) + \frac{\gamma H}{N} \sum_{c=1}^{N} \widetilde{C}_{(c)}^{1:H} \left( \xi_{(c)} - \xi_{(c)}^{\star} \right) \right)^{\otimes 2}$$

$$= \bar{\Gamma} \left( \theta - \theta^{\star} \right)^{\otimes 2} \bar{\Gamma} + \frac{\gamma^2 H^2}{N^2} \sum_{c=1}^{N} \sum_{c'=1}^{N} \widetilde{C}_{(c)}^{1:H} \left( \xi_{(c)} - \xi_{(c)}^{\star} \right) \left( \xi_{(c')} - \xi_{(c')}^{\star} \right)^{\top} \widetilde{C}_{(c')}^{1:H}$$

$$+ \frac{\gamma H}{N} \sum_{c=1}^{N} \left\{ \bar{\Gamma} \left( \theta - \theta^{\star} \right) \left( \xi_{(c)} - \xi_{(c)}^{\star} \right)^{\top} \widetilde{C}_{(c)}^{1:H} + \widetilde{C}_{(c)}^{1:H} \left( \xi_{(c)} - \xi_{(c)}^{\star} \right) \left( \theta - \theta^{\star} \right) \bar{\Gamma} \right\} \quad ,$$

and the lemma follows by integrating over the stationary distribution of SCAFFOLD. $\qquad \square$

**Lemma C.4.** *Assume A1, A2 and A5. Assume the step size $\gamma$ and the number of local updates $H$ satisfy $\gamma H(L + \mu) \le 1$. Then, it holds that*

$$\bar{\Sigma}_{(c)}^{\theta,\xi} = \bar{\Gamma} \bar{\Sigma}^{\theta} \Delta_{(c)}^{\Gamma} + \bar{\Gamma} \bar{\Sigma}_{(c)}^{\theta,\xi} \widetilde{C}_{(c)}^{1:H} - \frac{1}{N} \sum_{i'=1}^{N} \bar{\Gamma} \bar{\Sigma}_{(i')}^{\theta,\xi} \widetilde{C}_{(i')}^{1:H} + \frac{\gamma H}{N} \sum_{i=1}^{N} \widetilde{C}_{(i)}^{1:H} \bar{\Sigma}_{(i)}^{\xi,\theta} \Delta_{(c)}^{\Gamma}$$

$$+ \frac{\gamma H}{N} \sum_{i=1}^{N} \widetilde{C}_{(i)}^{1:H} \bar{\Sigma}_{(i,c)}^{\xi} \widetilde{C}_{(c)}^{1:H} - \frac{\gamma H}{N^2} \sum_{i=1}^{N} \sum_{i'=1}^{N} \widetilde{C}_{(i)}^{1:H} \bar{\Sigma}_{(i,i')}^{\xi} \widetilde{C}_{(i')}^{1:H} + \frac{\gamma}{NH} \left( \bar{\Sigma}_{(c)}^{\epsilon} - \bar{\Sigma}^{\epsilon} \right) + R_{(c)}^{\theta,\xi} \quad ,$$

*where $\bar{\Sigma}^{\epsilon} = \frac{1}{N} \sum_{c=1}^{N} \mathbb{E} \left[ \left( \varepsilon_{(c)}^{1:H} \right)^{\otimes 2} \right]$, and $R_{(c)}^{\theta,\xi} = R_{(c),1}^{\theta,\xi} + R_{(c),2}^{\theta,\xi} + R_{(c),3}^{\theta,\xi} + R_{(c),4}^{\theta,\xi} + R_{(c),5}^{\theta,\xi}$, with*

$$R_{(c),1}^{\theta,\xi} = \frac{\gamma}{N} \int \sum_{i=1}^{N} \mathbb{E} \left[ \varepsilon_{(i)}^{1:H} \left( \frac{1}{H} \mathcal{R}_{(c)}^{1:H} - \frac{1}{NH} \sum_{i'=1}^{N} \mathcal{R}_{(i')}^{1:H} \right)^{\top} + \mathcal{R}_{(i)}^{1:H} \left( \frac{1}{H} \varepsilon_{(c)}^{1:H} - \frac{1}{NH} \sum_{i'=1}^{N} \varepsilon_{(i')}^{1:H} \right)^{\top} \right] \pi_{(\gamma,H)}(\mathrm{d}\theta, \mathrm{d}\Xi) \quad ,$$

$$R_{(c),2}^{\theta,\xi} = -\frac{\gamma}{N} \sum_{i=1}^{N} \int \mathbb{E} \left[ \mathcal{R}_{(i)}^{1:H} \right] \left( \Delta_{(c)}^{\Gamma} \left( \theta - \theta^{\star} \right) \right)^{\top} \pi_{(\gamma,H)}(\mathrm{d}\theta, \mathrm{d}\Xi) \quad ,$$

$$R_{(c),3}^{\theta,\xi} = -\frac{\gamma}{N} \sum_{i=1}^{N} \int \mathbb{E} \left[ \mathcal{R}_{(i)}^{1:H} \right] \left( \widetilde{C}_{(c)}^{1:H} \left( \xi_{(c)} - \xi_{(c)}^{\star} \right) - \frac{1}{N} \sum_{i'=1}^{N} \widetilde{C}_{(i')}^{1:H} \left( \xi_{(i')} - \xi_{(i')}^{\star} \right) \right)^{\top} \pi_{(\gamma,H)}(\mathrm{d}\theta, \mathrm{d}\Xi) \quad ,$$

$$R_{(c),4}^{\theta,\xi} = \int \mathbb{E} \left[ \left( \bar{\Gamma} \left( \theta - \theta^{\star} \right) + \frac{\gamma H}{N} \sum_{i=1}^{N} \widetilde{C}_{(i)}^{1:H} \left( \xi_{(i)} - \xi_{(i)}^{\star} \right) \right) \left( -\frac{1}{H} \mathcal{R}_{(c)}^{1:H} + \frac{1}{NH} \sum_{i'=1}^{N} \mathcal{R}_{(i')}^{1:H} \right)^{\top} \right] \pi_{(\gamma,H)}(\mathrm{d}\theta, \mathrm{d}\Xi) \quad ,$$

$$R_{(c),5}^{\theta,\xi} = \frac{\gamma}{N} \sum_{i=1}^{N} \int \mathbb{E} \left[ \mathcal{R}_{(i)}^{1:H} \left( \frac{1}{H} \mathcal{R}_{(c)}^{1:H} - \frac{1}{NH} \sum_{i'=1}^{N} \mathcal{R}_{(i')}^{1:H} \right)^{\top} \right] \pi_{(\gamma,H)}(\mathrm{d}\theta, \mathrm{d}\Xi) \quad .$$

*Proof.* Using Lemma C.1 and Lemma C.2, we have

$$\left( \theta^{+} - \theta^{\star} \right) \left( \xi_{(c)}^{+} - \xi_{(c)}^{\star} \right)^{\top} = \left( \bar{\Gamma} \left( \theta - \theta^{\star} \right) + \frac{\gamma H}{N} \sum_{i=1}^{N} \widetilde{C}_{(i)}^{1:H} \left( \xi_{(i)} - \xi_{(i)}^{\star} \right) - \frac{\gamma}{N} \sum_{i=1}^{N} \mathcal{R}_{(i)}^{1:H} - \frac{\gamma}{N} \sum_{i=1}^{N} \varepsilon_{(i)}^{1:H} \right)$$

$$\times \left( \Delta_{(c)}^{\Gamma} \left( \theta - \theta^{\star} \right) + \widetilde{C}_{(c)}^{1:H} \left( \xi_{(c)} - \xi_{(c)}^{\star} \right) - \frac{1}{N} \sum_{i'=1}^{N} \widetilde{C}_{(i')}^{1:H} \left( \xi_{(i')} - \xi_{(i')}^{\star} \right) \right.$$

$$\left. -\frac{1}{H} \mathcal{R}_{(c)}^{1:H} + \frac{1}{NH} \sum_{i'=1}^{N} \mathcal{R}_{(i')}^{1:H} - \frac{1}{H} \varepsilon_{(c)}^{1:H} + \frac{1}{NH} \sum_{i=1}^{N} \varepsilon_{(i')}^{1:H} \right) \quad .$$

Taking the expectation, we have

$$
\mathbb{E}\left[\left(\theta^+ - \theta^\star\right)\left(\xi_{(c)}^+ - \xi_{(c)}^\star\right)^\top\right]
$$

$$
= \left(\bar{\Gamma}\left(\theta - \theta^\star\right) + \frac{\gamma H}{N}\sum_{i=1}^N \widetilde{\mathrm{C}}_{(i)}^{1:H}\left(\xi_{(i)} - \xi_{(i)}^\star\right)\right)\left(\Delta_{(c)}^\Gamma\left(\theta - \theta^\star\right) + \widetilde{\mathrm{C}}_{(c)}^{1:H}\left(\xi_{(c)} - \xi_{(c)}^\star\right) - \frac{1}{N}\sum_{i'=1}^N \widetilde{\mathrm{C}}_{(i')}^{1:H}\left(\xi_{(i')} - \xi_{(i')}^\star\right)\right)^\top
$$

$$
- \frac{\gamma}{N}\sum_{i=1}^N \mathbb{E}\left[\varepsilon_{(i)}^{1:H} \times \left(-\frac{1}{H}\varepsilon_{(c)}^{1:H} + \frac{1}{NH}\sum_{i'=1}^N \varepsilon_{(i')}^{1:H}\right)^\top\right]
$$

$$
- \frac{\gamma}{N}\sum_{i=1}^N \mathbb{E}\left[\varepsilon_{(i)}^{1:H} \times \left(-\frac{1}{H}\mathcal{R}_{(c)}^{1:H} + \frac{1}{NH}\sum_{i'=1}^N \mathcal{R}_{(i')}^{1:H}\right)^\top\right] - \frac{\gamma}{N}\sum_{i=1}^N \mathbb{E}\left[\mathcal{R}_{(i)}^{1:H}\left(-\frac{1}{H}\varepsilon_{(c)}^{1:H} + \frac{1}{NH}\sum_{i'=1}^N \varepsilon_{(i')}^{1:H}\right)^\top\right]
$$

$$
+ \mathbb{E}\left[\left(\bar{\Gamma}\left(\theta - \theta^\star\right) + \frac{\gamma H}{N}\sum_{i=1}^N \widetilde{\mathrm{C}}_{(i)}^{1:H}\left(\xi_{(i)} - \xi_{(i)}^\star\right) - \frac{\gamma}{N}\sum_{i=1}^N \mathcal{R}_{(i)}^{1:H}\right)\left(-\frac{1}{H}\mathcal{R}_{(c)}^{1:H} + \frac{1}{NH}\sum_{i'=1}^N \mathcal{R}_{(i')}^{1:H}\right)^\top\right]
$$

$$
- \frac{\gamma}{N}\sum_{i=1}^N \mathbb{E}\left[\mathcal{R}_{(i)}^{1:H}\right]\left(\Delta_{(c)}^\Gamma\left(\theta - \theta^\star\right) + \widetilde{\mathrm{C}}_{(c)}^{1:H}\left(\xi_{(c)} - \xi_{(c)}^\star\right) - \frac{1}{N}\sum_{i'=1}^N \widetilde{\mathrm{C}}_{(i')}^{1:H}\left(\xi_{(i')} - \xi_{(i')}^\star\right)\right)^\top .
$$

The result follows by expanding the first term of the right hand side and integrating the resulting identity over SCAFFOLD's stationary distribution. $\qquad\square$

**Lemma C.5.** *Assume A1, A2 and A5. Assume the step size $\gamma$ and the number of local updates $H$ satisfy $\gamma H(L + \mu) \leq 1$. Then, for $c, c' \in \{1, \ldots, N\}$ such that $c \neq c'$, it holds that*

$$
\bar{\boldsymbol{\Sigma}}_{(c,c)}^\xi = \Delta_{(c)}^\Gamma \bar{\boldsymbol{\Sigma}}^\theta \Delta_{(c')}^\Gamma + \frac{1}{H^2}\bar{\boldsymbol{\Sigma}}_{(c)}^\epsilon - \frac{2}{NH^2}\bar{\boldsymbol{\Sigma}}_{(c)}^\epsilon + \frac{1}{NH^2}\bar{\boldsymbol{\Sigma}}^\epsilon
$$

$$
+ \Delta_{(c)}^\Gamma \bar{\boldsymbol{\Sigma}}_{(c)}^{\theta,\xi}\widetilde{\mathrm{C}}_{(c)}^{1:H} - \frac{1}{N}\sum_{i'=1}^N \Delta_{(c)}^\Gamma \bar{\boldsymbol{\Sigma}}_{(i')}^{\theta,\xi}\widetilde{\mathrm{C}}_{(i')}^{1:H} + \widetilde{\mathrm{C}}_{(c)}^{1:H}\bar{\boldsymbol{\Sigma}}_{(c)}^{\xi,\theta}\Delta_{(c)}^\Gamma - \frac{1}{N}\sum_{i=1}^N \widetilde{\mathrm{C}}_{(i)}^{1:H}\bar{\boldsymbol{\Sigma}}_{(i)}^{\xi,\theta}\Delta_{(c)}^\Gamma
$$

$$
+ \widetilde{\mathrm{C}}_{(c)}^{1:H}\bar{\boldsymbol{\Sigma}}_{(c,c)}^\xi \widetilde{\mathrm{C}}_{(c)}^{1:H} - \frac{1}{N}\sum_{i'=1}^N \widetilde{\mathrm{C}}_{(c)}^{1:H}\bar{\boldsymbol{\Sigma}}_{(c,i')}^\xi \widetilde{\mathrm{C}}_{(i')}^{1:H} - \frac{1}{N}\sum_{i=1}^N \widetilde{\mathrm{C}}_{(i)}^{1:H}\bar{\boldsymbol{\Sigma}}_{(i,c)}^\xi \widetilde{\mathrm{C}}_{(c)}^{1:H} + \frac{1}{N^2}\sum_{i=1}^N\sum_{i'=1}^N \widetilde{\mathrm{C}}_{(i)}^{1:H}\bar{\boldsymbol{\Sigma}}_{(i,i')}^\xi \widetilde{\mathrm{C}}_{(i')}^{1:H} + \mathrm{R}_{(c,c)}^\xi ,
$$

$$
\bar{\boldsymbol{\Sigma}}_{(c,c')}^\xi = \Delta_{(c)}^\Gamma \bar{\boldsymbol{\Sigma}}^\theta \Delta_{(c')}^\Gamma - \frac{1}{NH^2}\bar{\boldsymbol{\Sigma}}_{(c)}^\epsilon - \frac{1}{NH^2}\bar{\boldsymbol{\Sigma}}_{(c')}^\epsilon + \frac{1}{NH^2}\bar{\boldsymbol{\Sigma}}^\epsilon
$$

$$
+ \Delta_{(c)}^\Gamma \bar{\boldsymbol{\Sigma}}_{(c')}^{\theta,\xi}\widetilde{\mathrm{C}}_{(c')}^{1:H} - \frac{1}{N}\sum_{i'=1}^N \Delta_{(c)}^\Gamma \bar{\boldsymbol{\Sigma}}_{(i')}^{\theta,\xi}\widetilde{\mathrm{C}}_{(i')}^{1:H} + \widetilde{\mathrm{C}}_{(c)}^{1:H}\bar{\boldsymbol{\Sigma}}_{(c)}^{\xi,\theta}\Delta_{(c')}^\Gamma - \frac{1}{N}\sum_{i=1}^N \widetilde{\mathrm{C}}_{(i)}^{1:H}\bar{\boldsymbol{\Sigma}}_{(i)}^{\xi,\theta}\Delta_{(c')}^\Gamma
$$

$$
+ \widetilde{\mathrm{C}}_{(c)}^{1:H}\bar{\boldsymbol{\Sigma}}_{(c,c')}^\xi \widetilde{\mathrm{C}}_{(c')}^{1:H} - \frac{1}{N}\sum_{i'=1}^N \widetilde{\mathrm{C}}_{(c)}^{1:H}\bar{\boldsymbol{\Sigma}}_{(c,i')}^\xi \widetilde{\mathrm{C}}_{(i')}^{1:H} - \frac{1}{N}\sum_{i=1}^N \widetilde{\mathrm{C}}_{(i)}^{1:H}\bar{\boldsymbol{\Sigma}}_{(i,c')}^\xi \widetilde{\mathrm{C}}_{(c')}^{1:H} + \frac{1}{N^2}\sum_{i=1}^N\sum_{i'=1}^N \widetilde{\mathrm{C}}_{(i)}^{1:H}\bar{\boldsymbol{\Sigma}}_{(i,i')}^\xi \widetilde{\mathrm{C}}_{(i')}^{1:H} + \mathrm{R}_{(c,c')}^\xi ,
$$

*where* $\mathrm{R}_{(c,c')}^\xi = \mathrm{R}_{(c,c'),1}^\xi + \mathrm{R}_{(c',c),1}^{\xi\top} + \mathrm{R}_{(c,c'),2}^\xi + \mathrm{R}_{(c',c),2}^{\xi\top} + \mathrm{R}_{(\star,c'),3}^\xi + \mathrm{R}_{(\star,c),3}^{\xi\top} + \mathrm{R}_{(c,c'),4}^\xi + \mathrm{R}_{(c',c),4}^{\xi\top} + \mathrm{R}_{(c,c'),5}^\xi$, *with*

$$
\mathrm{R}_{(c,c'),1}^\xi = -\frac{1}{H}\int \Delta_{(c)}^\Gamma\left(\theta - \theta^\star\right)\mathbb{E}\left[\mathcal{R}_{(c')}^{1:H} - \frac{1}{N}\sum_{i'=1}^N \mathcal{R}_{(i')}^{1:H}\right]^\top \pi_{(\gamma,H)}(\mathrm{d}\theta, \mathrm{d}\Xi) ,
$$

$$
\mathrm{R}_{(c,c'),2}^\xi = -\frac{1}{H}\int \widetilde{\mathrm{C}}_{(c)}^{1:H}\left(\xi_{(c)} - \xi_{(c)}^\star\right)\mathbb{E}\left[\mathcal{R}_{(c')}^{1:H} - \frac{1}{N}\sum_{i'=1}^N \mathcal{R}_{(i')}^{1:H}\right]^\top \pi_{(\gamma,H)}(\mathrm{d}\theta, \mathrm{d}\Xi) ,
$$

$$
\mathrm{R}_{(\star,c'),3}^\xi = \frac{1}{NH}\int \sum_{i=1}^N \widetilde{\mathrm{C}}_{(i)}^{1:H}\left(\xi_{(i)} - \xi_{(i)}^\star\right)\mathbb{E}\left[\frac{1}{H}\mathcal{R}_{(c')}^{1:H} - \frac{1}{NH}\sum_{i'=1}^N \mathcal{R}_{(i')}^{1:H}\right]^\top \pi_{(\gamma,H)}(\mathrm{d}\theta, \mathrm{d}\Xi) ,
$$

$$
\mathrm{R}_{(c,c'),4}^\xi = \frac{1}{H^2}\int \mathbb{E}\left[\left(\mathcal{R}_{(c)}^{1:H} - \frac{1}{N}\sum_{i=1}^N \mathcal{R}_{(i)}^{1:H}\right)\left(\varepsilon_{(c')}^{1:H} - \frac{1}{N}\sum_{i=1}^N \varepsilon_{(i')}^{1:H}\right)^\top\right] \pi_{(\gamma,H)}(\mathrm{d}\theta, \mathrm{d}\Xi) ,
$$

$$R^{\xi}_{(c,c'),5} = \frac{1}{H^2} \int \mathbb{E}\left[\left(\mathcal{R}^{1:H}_{(c)} + \frac{1}{N}\sum_{i=1}^{N}\mathcal{R}^{1:H}_{(i)}\right)\left(\mathcal{R}^{1:H}_{(c')} - \frac{1}{N}\sum_{i'=1}^{N}\mathcal{R}^{1:H}_{(i')}\right)^{\top}\right]\pi_{(\gamma,H)}(\mathrm{d}\theta,\mathrm{d}\Xi) .$$

*Proof.* Recall the expression of $\xi^+_{(c)}$ from Lemma C.2, we have

$$\xi^+_{(c)} - \xi^\star_{(c)} = \Delta^{\Gamma}_{(c)}(\theta - \theta^\star) + \widetilde{C}^{1:H}_{(c)}\left(\xi_{(c)} - \xi^\star_{(c)}\right) - \frac{1}{N}\sum_{i=1}^{N}\widetilde{C}^{1:H}_{(i)}\left(\xi_{(i)} - \xi^\star_{(i)}\right)$$

$$- \frac{1}{H}\mathcal{R}^{1:H}_{(c)} + \frac{1}{NH}\sum_{i=1}^{N}\mathcal{R}^{1:H}_{(i)} - \frac{1}{H}\varepsilon^{1:H}_{(c)} + \frac{1}{NH}\sum_{i=1}^{N}\varepsilon^{1:H}_{(i)} .$$

Taking the expectation and expanding the product, we obtain, for any $c, c' \in \{1, \dots, N\}$,

$$\mathbb{E}\left[\left(\xi^+_{(c)} - \xi^\star_{(c)}\right)\left(\xi^+_{(c')} - \xi^\star_{(c')}\right)^{\top}\right]$$

$$= \Delta^{\Gamma}_{(c)}(\theta - \theta^\star)^{\otimes 2}\Delta^{\Gamma}_{(c')} + \Delta^{\Gamma}_{(c)}(\theta - \theta^\star)\left(\xi_{(c')} - \xi^\star_{(c')}\right)^{\top}\widetilde{C}^{1:H}_{(c')}$$

$$- \frac{1}{N}\sum_{i'=1}^{N}\Delta^{\Gamma}_{(c)}(\theta - \theta^\star)\left(\xi_{(i')} - \xi^\star_{(i')}\right)^{\top}\widetilde{C}^{1:H}_{(i')} - \frac{1}{H}\Delta^{\Gamma}_{(c)}(\theta - \theta^\star)\mathbb{E}\left[\mathcal{R}^{1:H}_{(c')} - \frac{1}{N}\sum_{i'=1}^{N}\mathcal{R}^{1:H}_{(i')}\right]^{\top}$$

$$+ \widetilde{C}^{1:H}_{(c)}\left(\xi_{(c)} - \xi^\star_{(c)}\right)(\theta - \theta^\star)^{\top}\Delta^{\Gamma}_{(c')} + \widetilde{C}^{1:H}_{(c)}\left(\xi_{(c)} - \xi^\star_{(c)}\right)\left(\xi_{(c')} - \xi^\star_{(c')}\right)^{\top}\widetilde{C}^{1:H}_{(c')}$$

$$- \frac{1}{N}\sum_{i'=1}^{N}\widetilde{C}^{1:H}_{(c)}\left(\xi_{(c)} - \xi^\star_{(c)}\right)\left(\xi_{(i')} - \xi^\star_{(i')}\right)^{\top}\widetilde{C}^{1:H}_{(i')} - \frac{1}{H}\widetilde{C}^{1:H}_{(c)}\left(\xi_{(c)} - \xi^\star_{(c)}\right)\mathbb{E}\left[\mathcal{R}^{1:H}_{(c')} - \frac{1}{N}\sum_{i'=1}^{N}\mathcal{R}^{1:H}_{(i')}\right]^{\top}$$

$$- \frac{1}{N}\sum_{i=1}^{N}\widetilde{C}^{1:H}_{(i)}\left(\xi_{(i)} - \xi^\star_{(i)}\right)(\theta - \theta^\star)^{\top}\Delta^{\Gamma}_{(c')} - \frac{1}{N}\sum_{i=1}^{N}\widetilde{C}^{1:H}_{(i)}\left(\xi_{(i)} - \xi^\star_{(i)}\right)\left(\xi_{(c')} - \xi^\star_{(c')}\right)^{\top}\widetilde{C}^{1:H}_{(c')}$$

$$+ \frac{1}{N^2}\sum_{i=1}^{N}\sum_{i'=1}^{N}\widetilde{C}^{1:H}_{(i)}\left(\xi_{(i)} - \xi^\star_{(i)}\right)\left(\xi_{(i')} - \xi^\star_{(i')}\right)^{\top}\widetilde{C}^{1:H}_{(i')} + \frac{1}{NH}\sum_{i=1}^{N}\widetilde{C}^{1:H}_{(i)}\left(\xi_{(i)} - \xi^\star_{(i)}\right)\mathbb{E}\left[\mathcal{R}^{1:H}_{(c')} - \frac{1}{N}\sum_{i'=1}^{N}\mathcal{R}^{1:H}_{(i')}\right]^{\top}$$

$$- \frac{1}{H}\mathbb{E}\left[\mathcal{R}^{1:H}_{(c)} - \frac{1}{N}\sum_{i=1}^{N}\mathcal{R}^{1:H}_{(i)}\right]\left((\theta - \theta^\star)^{\top}\Delta^{\Gamma}_{(c')} + \left(\xi_{(c')} - \xi^\star_{(c')}\right)^{\top}\widetilde{C}^{1:H}_{(c')} - \frac{1}{N}\sum_{i'=1}^{N}\left(\xi_{(i')} - \xi^\star_{(i')}\right)^{\top}\widetilde{C}^{1:H}_{(i')}\right)$$

$$+ \frac{1}{H^2}\mathbb{E}\left[\left(\mathcal{R}^{1:H}_{(c)} - \frac{1}{N}\sum_{i=1}^{N}\mathcal{R}^{1:H}_{(i)}\right)\left(\mathcal{R}^{1:H}_{(c')} - \frac{1}{N}\sum_{i'=1}^{N}\mathcal{R}^{1:H}_{(i')} + \varepsilon^{1:H}_{(c')} - \frac{1}{N}\sum_{i=1}^{N}\varepsilon^{1:H}_{(i')}\right)^{\top}\right]$$

$$+ \frac{1}{H^2}\mathbb{E}\left[\left(\varepsilon^{1:H}_{(c)} - \frac{1}{N}\sum_{i=1}^{N}\varepsilon^{1:H}_{(i)}\right)\left(\mathcal{R}^{1:H}_{(c')} - \frac{1}{N}\sum_{i'=1}^{N}\mathcal{R}^{1:H}_{(i')} + \varepsilon^{1:H}_{(c')} - \frac{1}{N}\sum_{i=1}^{N}\varepsilon^{1:H}_{(i')}\right)\right] .$$

Integrating over the stationary distribution, this yields

$$\bar{\Sigma}^{\xi}_{(c,c')} = \Delta^{\Gamma}_{(c)}\bar{\Sigma}^{\theta}\Delta^{\Gamma}_{(c')}$$

$$+ \Delta^{\Gamma}_{(c)}\bar{\Sigma}^{\theta,\xi}_{(c')}\widetilde{C}^{1:H}_{(c')} - \frac{1}{N}\sum_{i'=1}^{N}\Delta^{\Gamma}_{(c)}\bar{\Sigma}^{\theta,\xi}_{(i')}\widetilde{C}^{1:H}_{(i')} + \widetilde{C}^{1:H}_{(c)}\bar{\Sigma}^{\xi,\theta}_{(c)}\Delta^{\Gamma}_{(c')} - \frac{1}{N}\sum_{i=1}^{N}\widetilde{C}^{1:H}_{(i)}\bar{\Sigma}^{\xi,\theta}_{(i)}\Delta^{\Gamma}_{(c')}$$

$$+ \widetilde{C}^{1:H}_{(c)}\bar{\Sigma}^{\xi}_{(c,c')}\widetilde{C}^{1:H}_{(c')} - \frac{1}{N}\sum_{i'=1}^{N}\widetilde{C}^{1:H}_{(c)}\bar{\Sigma}^{\xi}_{(c,i')}\widetilde{C}^{1:H}_{(i')} - \frac{1}{N}\sum_{i=1}^{N}\widetilde{C}^{1:H}_{(i)}\bar{\Sigma}^{\xi}_{(i,c')}\widetilde{C}^{1:H}_{(c')} + \frac{1}{N^2}\sum_{i=1}^{N}\sum_{i'=1}^{N}\widetilde{C}^{1:H}_{(i)}\bar{\Sigma}^{\xi}_{(i,i')}\widetilde{C}^{1:H}_{(i')}$$

$$+ \frac{1}{H^2}\int \mathbb{E}\left[\left(\varepsilon^{1:H}_{(c)} - \frac{1}{N}\sum_{i=1}^{N}\varepsilon^{1:H}_{(i)}\right)\left(\varepsilon^{1:H}_{(c')} - \frac{1}{N}\sum_{i'=1}^{N}\varepsilon^{1:H}_{(i')}\right)^{\top}\right]\pi_{(\gamma,H)}(\mathrm{d}\theta,\mathrm{d}\Xi)$$

$$- \frac{1}{H} \int \mathbb{E}\left[ \Delta_{(c)}^{\Gamma} (\theta - \theta^\star) \left( \mathcal{R}_{(c')}^{1:H} - \frac{1}{N} \sum_{i'=1}^{N} \mathcal{R}_{(i')}^{1:H} \right)^\top + \left( \mathcal{R}_{(c)}^{1:H} - \frac{1}{N} \sum_{i=1}^{N} \mathcal{R}_{(i)}^{1:H} \right) (\theta - \theta^\star)^\top \Delta_{(c')}^{\Gamma} \right] \pi_{(\gamma, H)}(\mathrm{d}\theta, \mathrm{d}\Xi)$$

$$- \frac{1}{H} \int \mathbb{E}\left[ \widetilde{\mathrm{C}}_{(c)}^{1:H} \left( \xi_{(c)} - \xi_{(c)}^\star \right) \left( \mathcal{R}_{(c')}^{1:H} - \frac{1}{N} \sum_{i'=1}^{N} \mathcal{R}_{(i')}^{1:H} \right)^\top + \left( \mathcal{R}_{(c)}^{1:H} - \frac{1}{N} \sum_{i=1}^{N} \mathcal{R}_{(i)}^{1:H} \right) \left( \xi_{(c')} - \xi_{(c')}^\star \right)^\top \widetilde{\mathrm{C}}_{(c')}^{1:H} \right] \pi_{(\gamma, H)}(\mathrm{d}\theta, \mathrm{d}\Xi)$$

$$+ \frac{1}{NH} \int \mathbb{E}\left[ \sum_{i=1}^{N} \widetilde{\mathrm{C}}_{(i)}^{1:H} \left( \xi_{(i)} - \xi_{(i)}^\star \right) \left( \mathcal{R}_{(c')}^{1:H} - \frac{1}{N} \sum_{i'=1}^{N} \mathcal{R}_{(i')}^{1:H} \right)^\top \right] \pi_{(\gamma, H)}(\mathrm{d}\theta, \mathrm{d}\Xi)$$

$$+ \frac{1}{NH} \int \mathbb{E}\left[ \left( \mathcal{R}_{(c)}^{1:H} - \frac{1}{N} \sum_{i=1}^{N} \mathcal{R}_{(i)}^{1:H} \right) \sum_{i'=1}^{N} \left( \xi_{(i')} - \xi_{(i')}^\star \right)^\top \widetilde{\mathrm{C}}_{(i')}^{1:H} \right] \pi_{(\gamma, H)}(\mathrm{d}\theta, \mathrm{d}\Xi)$$

$$+ \frac{1}{H^2} \int \mathbb{E}\left[ \left( \mathcal{R}_{(c)}^{1:H} - \frac{1}{N} \sum_{i=1}^{N} \mathcal{R}_{(i)}^{1:H} \right) \left( \varepsilon_{(c')}^{1:H} - \frac{1}{N} \sum_{i=1}^{N} \varepsilon_{(i')}^{1:H} \right)^\top \right] \pi_{(\gamma, H)}(\mathrm{d}\theta, \mathrm{d}\Xi)$$

$$+ \frac{1}{H^2} \int \mathbb{E}\left[ \left( \varepsilon_{(c)}^{1:H} - \frac{1}{N} \sum_{i=1}^{N} \varepsilon_{(i)}^{1:H} \right) \left( \mathcal{R}_{(c')}^{1:H} - \frac{1}{N} \sum_{i'=1}^{N} \mathcal{R}_{(i')}^{1:H} \right)^\top \right] \pi_{(\gamma, H)}(\mathrm{d}\theta, \mathrm{d}\Xi)$$

$$+ \frac{1}{H^2} \int \mathbb{E}\left[ \left( \mathcal{R}_{(c)}^{1:H} - \frac{1}{N} \sum_{i=1}^{N} \mathcal{R}_{(i)}^{1:H} \right) \left( \mathcal{R}_{(c')}^{1:H} - \frac{1}{N} \sum_{i'=1}^{N} \mathcal{R}_{(i')}^{1:H} \right)^\top \right] \pi_{(\gamma, H)}(\mathrm{d}\theta, \mathrm{d}\Xi) \ .$$

To study the noise term, we expand

$$\left( \varepsilon_{(c)}^{1:H} - \frac{1}{N} \sum_{i=1}^{N} \varepsilon_{(i)}^{1:H} \right) \left( \varepsilon_{(c')}^{1:H} - \frac{1}{N} \sum_{i=1}^{N} \varepsilon_{(i')}^{1:H} \right)^\top$$

$$= \varepsilon_{(c)}^{1:H} \varepsilon_{(c')}^{1:H\top} - \frac{1}{N} \sum_{i=1}^{N} \varepsilon_{(i)}^{1:H} \varepsilon_{(c')}^{1:H\top} - \frac{1}{N} \sum_{i'=1}^{N} \varepsilon_{(c)}^{1:H} \varepsilon_{(i')}^{1:H\top} + \frac{1}{N^2} \sum_{i=1}^{N} \sum_{i'=1}^{N} \varepsilon_{(i)}^{1:H} \varepsilon_{(i')}^{1:H\top} \ .$$

Now we distinguish two cases. First, if $c \neq c'$, we have

$$\frac{1}{H^2} \int \mathbb{E}\left[ \left( \varepsilon_{(c)}^{1:H} - \frac{1}{N} \sum_{i=1}^{N} \varepsilon_{(i)}^{1:H} \right) \left( \varepsilon_{(c')}^{1:H} - \frac{1}{N} \sum_{i=1}^{N} \varepsilon_{(i')}^{1:H} \right)^\top \right] \pi_{(\gamma, H)}(\mathrm{d}\theta, \mathrm{d}\Xi) = - \frac{1}{NH^2} \bar{\Sigma}_{(c)}^\epsilon - \frac{1}{NH^2} \bar{\Sigma}_{(c')}^\epsilon + \frac{1}{NH^2} \bar{\Sigma}^\epsilon \ .$$

Otherwise, we have $c = c'$ and

$$\frac{1}{H^2} \int \mathbb{E}\left[ \left( \varepsilon_{(c)}^{1:H} - \frac{1}{N} \sum_{i=1}^{N} \varepsilon_{(i)}^{1:H} \right) \left( \varepsilon_{(c)}^{1:H} - \frac{1}{N} \sum_{i=1}^{N} \varepsilon_{(i')}^{1:H} \right)^\top \right] \pi_{(\gamma, H)}(\mathrm{d}\theta, \mathrm{d}\Xi) = \frac{1}{H^2} \bar{\Sigma}_{(c)}^\epsilon - \frac{2}{NH^2} \bar{\Sigma}_{(c)}^\epsilon + \frac{1}{NH^2} \bar{\Sigma}^\epsilon \ ,$$

and plugging these identities in the above equality gives the lemma. $\qquad \square$

### C.2.2. BOUND ON REMAINDER TERMS

**Lemma C.6.** *Assume A1, A2 and A5. Assume the step size $\gamma$ and the number of local updates $H$ satisfy $\gamma H(L + \mu) \leq 1/48$, $\gamma \beta^{1/2} H^{1/2} \leq 1/12$, and $\gamma \beta \leq L/12$. Then, it holds that*

$$| \operatorname{tr} \mathrm{R}^\theta | \leq \frac{1080 \gamma^{5/2} H Q}{\mu^{3/2}} \sigma_\star^3 + \frac{2 \cdot 600^2 \gamma^4 H^2 Q^2}{N \mu^2} \sigma_\star^4 \ ,$$

$$| \operatorname{tr} \mathrm{R}_{(c)}^{\theta, \xi} | \leq \frac{6000 \gamma^{3/2} Q}{\mu^{3/2}} \sigma_\star^3 + \frac{2 \cdot 600^2 \gamma^3 H Q^2}{\mu^2} \sigma_\star^4 \ ,$$

$$| \operatorname{tr} \mathrm{R}_{(c, c')}^\xi | \leq \frac{8000 \gamma^{1/2} Q}{H \mu^{3/2}} \sigma_\star^3 + \frac{4 \cdot 600^2 \gamma^2 Q^2}{\mu^2} \sigma_\star^4 \ ,$$

*where $\mathrm{R}^\theta$, $\mathrm{R}^{\theta, \xi}$ and $\mathrm{R}^\xi$ are defined in Lemma C.3, Lemma C.4 and Lemma C.5 respectively.*

*Proof.* **Bound on $R^\theta$.** We bound each of the terms from $|\operatorname{tr} R^\theta| = |2\operatorname{tr} R_1^\theta + 2\operatorname{tr} R_2^\theta + 2\operatorname{tr} R_3^\theta| \le |2\operatorname{tr} R_1^\theta| + |2\operatorname{tr} R_2^\theta| + |2\operatorname{tr} R_3^\theta|$. We have, using Cauchy-Schwarz and Hölder inequalities,

$$|\operatorname{tr} R_1^\theta| \le \frac{\gamma^2}{N^2} \sum_{c=1}^N \Big| \int \mathbb{E}\Big[\operatorname{tr}\Big(\varepsilon_{(c)}^{1:H}\Big)\Big(\mathcal{R}_{(c)}^{1:H}\Big)^\top\Big] \pi_{(\gamma,H)}(\mathrm{d}\theta, \mathrm{d}\Xi)\Big|$$

$$\le \frac{\gamma^2}{N^2} \sum_{c=1}^N \Big(\int \mathbb{E}\Big[\|\varepsilon_{(c)}^{1:H}\|^2\Big] \pi_{(\gamma,H)}(\mathrm{d}\theta, \mathrm{d}\Xi)\Big)^{1/2} \Big(\int \mathbb{E}\Big[\|\mathcal{R}_{(c)}^{1:H}\|^2\Big] \pi_{(\gamma,H)}(\mathrm{d}\theta, \mathrm{d}\Xi)\Big)^{1/2} \ .$$

By Lemma C.13 and Lemma C.14,

$$|\operatorname{tr} R_1^\theta| \le \frac{\gamma^2}{N}\Big(H^{1/2}\sigma_\star + \frac{6\gamma^{1/2}\beta^{1/2}H^{1/2}}{\mu^{1/2}}\sigma_\star\Big)\frac{600\gamma HQ}{\mu}\sigma_\star^2 = \frac{600\gamma^3 H^{3/2}Q}{N\mu}\sigma_\star^3 + \frac{6\cdot 600\gamma^{7/2}\beta^{1/2}H^{3/2}Q}{N\mu^{3/2}}\sigma_\star^3 \ .$$

Then, by Corollary B.1, Lemma C.14, and Lemma 4.6

$$|\operatorname{tr} R_2^\theta| \le \frac{\gamma}{N}\sum_{c=1}^N \Big|\int \operatorname{tr}\mathbb{E}\Big[\mathcal{R}_{(c)}^{1:H}\Big](\theta - \theta^\star)^\top \bar{\Gamma}\pi_{(\gamma,H)}(\mathrm{d}\theta, \mathrm{d}\Xi)\Big|$$

$$+ \frac{\gamma^2 H}{N^2}\sum_{c=1}^N\sum_{c'=1}^N \Big|\int \operatorname{tr}\mathbb{E}\Big[\mathcal{R}_{(c)}^{1:H}\Big]\Big(\xi_{(c')} - \xi_{(c')}^\star\Big)^\top \widetilde{C}_{(c')}^{1:H}\pi_{(\gamma,H)}(\mathrm{d}\theta, \mathrm{d}\Xi)\Big|$$

$$\le \gamma \cdot \frac{28\gamma HQ}{\mu}\sigma_\star^2 \cdot \frac{3\gamma^{1/2}}{\mu^{1/2}}\sigma_\star + \gamma^2 H \cdot \frac{28\gamma HQ}{\mu}\sigma_\star^2 \cdot \frac{8L^{1/2}}{\mu^{1/2}H^{1/2}}\sigma_\star \ ,$$

which gives, using $\gamma HL \le 1/48$ in the second inequality,

$$|\operatorname{tr} R_2^\theta| \le \frac{84Q\gamma^{5/2}H}{\mu^{3/2}}\sigma_\star^3 + \frac{224Q\gamma^3 H^{3/2}L^{1/2}}{\mu^{3/2}}\sigma_\star^3 \le \frac{90Q\gamma^{5/2}H}{\mu^{3/2}}\sigma_\star^3 \ .$$

Finally, by Lemma C.14, we obtain

$$|\operatorname{tr} R_3^\theta| \le \frac{\gamma^2}{N^2}\sum_{c=1}^N\sum_{c'=1}^N \int\Big|\mathbb{E}\Big[\operatorname{tr}\Big(\mathcal{R}_{(c)}^{1:H}\Big)\Big(\mathcal{R}_{(c')}^{1:H}\Big)^\top\Big]\pi_{(\gamma,H)}(\mathrm{d}\theta, \mathrm{d}\Xi)\Big| \le \gamma^2\frac{600^2\gamma^2 H^2 Q^2}{\mu^2}\sigma_\star^4 \ .$$

Summing these inequalities, we obtain

$$|\operatorname{tr} R^\theta| \le \frac{1200\gamma^3(\mu^{1/2} + 6\gamma^{1/2}\beta^{1/2})H^{3/2}Q}{N\mu^{3/2}}\sigma_\star^3 + \frac{180Q\gamma^{5/2}H}{\mu^{3/2}}\sigma_\star^3 + \frac{2\cdot 600^2\gamma^4 H^2 Q^2}{\mu^2}\sigma_\star^4 \ ,$$

and the result follows from $\gamma\beta^{1/2}H^{1/2} \le 1/12$ and $\gamma^{1/2}H^{1/2}\mu^{1/2} \le 1/6$.

**Bound on $R_{(c)}^{\theta,\xi}$.** We bound each term of $|\operatorname{tr} R_{(c)}^{\theta,\xi}| = |R_{(c),1}^{\theta,\xi} + \operatorname{tr} R_{(c),2}^{\theta,\xi} + \operatorname{tr} R_{(c),3}^{\theta,\xi} + \operatorname{tr} R_{(c),4}^{\theta,\xi} + \operatorname{tr} R_{(c),5}^{\theta,\xi}| \le |R_{(c),1}^{\theta,\xi}| + |\operatorname{tr} R_{(c),2}^{\theta,\xi}| + |\operatorname{tr} R_{(c),3}^{\theta,\xi}| + |\operatorname{tr} R_{(c),4}^{\theta,\xi}| + |\operatorname{tr} R_{(c),5}^{\theta,\xi}|$. By Lemma C.13, and Lemma C.14,

$$|\operatorname{tr} R_{(c),1}^{\theta,\xi}| \le \frac{\gamma}{N}\sum_{i=1}^N \Big|\int \mathbb{E}\Big[\operatorname{tr}\varepsilon_{(i)}^{1:H}\Big(\frac{1}{H}\mathcal{R}_{(c)}^{1:H} - \frac{1}{NH}\sum_{i'=1}^N \mathcal{R}_{(i')}^{1:H}\Big)^\top + \operatorname{tr}\mathcal{R}_{(i)}^{1:H}\Big(\frac{1}{H}\varepsilon_{(c)}^{1:H} - \frac{1}{NH}\sum_{i'=1}^N \varepsilon_{(i')}^{1:H}\Big)^\top\Big]\pi_{(\gamma,H)}(\mathrm{d}\theta, \mathrm{d}\Xi)\Big|$$

$$\le \gamma\Big(\Big(H^{1/2}\sigma_\star + \frac{6\gamma^{1/2}\beta^{1/2}H^{1/2}}{\mu^{1/2}}\sigma_\star\Big)\cdot\frac{2\cdot 600\gamma Q}{\mu}\sigma_\star^2\Big) + 2\gamma\Big(\Big(H^{1/2}\sigma_\star + \frac{6\gamma^{1/2}\beta^{1/2}H^{1/2}}{\mu^{1/2}}\sigma_\star\Big)\cdot\frac{600\gamma Q}{\mu}\sigma_\star^2\Big) \ ,$$

which implies $|\operatorname{tr} R_{(c),1}^{\theta,\xi}| \le \frac{2400\gamma^2 Q(\mu^{1/2} + 6\gamma^{1/2}\beta^{1/2})H^{1/2}}{\mu^{3/2}}\sigma_\star^3$. Then, using Corollary B.1, Lemma C.12, and Lemma C.14,

$$|\operatorname{tr} R_{(c),2}^{\theta,\xi}| \le \frac{\gamma}{N}\sum_{i=1}^N \Big|\int \operatorname{tr}\mathbb{E}\Big[\mathcal{R}_{(i)}^{1:H}\Big]\Big(\Delta_{(c)}^\Gamma(\theta - \theta^\star)\Big)^\top \pi_{(\gamma,H)}(\mathrm{d}\theta, \mathrm{d}\Xi)\Big| \le \gamma\cdot\frac{600\gamma Q}{\mu}\sigma_\star^2\cdot\zeta_2\cdot\frac{3\gamma^{1/2}}{\mu^{1/2}}\sigma_\star \ ,$$

which gives $|\operatorname{tr} \mathrm{R}^{\theta,\xi}_{(c),2}| \le \frac{1800\gamma^{5/2}Q\zeta_2}{\mu^{3/2}}\sigma_\star^3$. Furthermore, we have, from Lemma 4.6, Lemma C.11, and Lemma C.14,

$$|\operatorname{tr} \mathrm{R}^{\theta,\xi}_{(c),3}| \le \frac{\gamma}{N}\sum_{i=1}^{N}\Big|\int \operatorname{tr} \mathbb{E}\left[\mathcal{R}^{1:H}_{(i)}\right]\left(\widetilde{\mathrm{C}}^{1:H}_{(c)}\left(\xi_{(c)}-\xi^\star_{(c)}\right)-\frac{1}{N}\sum_{i'=1}^{N}\widetilde{\mathrm{C}}^{1:H}_{(i')}\left(\xi_{(i')}-\xi^\star_{(i')}\right)\right)^\top \pi_{(\gamma,H)}(\mathrm{d}\theta,\mathrm{d}\Xi)\Big|$$

$$\le 2\gamma \cdot \frac{600\gamma Q}{\mu}\sigma_\star^2 \cdot \frac{\gamma(H-1)L}{2}\cdot\frac{8L^{1/2}}{\mu^{1/2}H^{1/2}}\sigma_\star \ ,$$

therefore, we have $|\operatorname{tr} \mathrm{R}^{\theta,\xi}_{(c),3}| \le \frac{4800\gamma^3 L^{3/2}H^{1/2}Q}{\mu^{3/2}}\sigma_\star^3$. We also bound, using Lemma 4.6, Lemma C.10, Lemma C.11, and Lemma C.14,

$$|\mathrm{R}^{\theta,\xi}_{(c),4}| = \Big|\int \operatorname{tr} \mathbb{E}\left[\left(\bar{\Gamma}\left(\theta-\theta^\star\right)+\frac{\gamma H}{N}\sum_{i=1}^{N}\widetilde{\mathrm{C}}^{1:H}_{(i)}\left(\xi_{(i)}-\xi^\star_{(i)}\right)\right)\left(-\frac{1}{H}\mathcal{R}^{1:H}_{(c)}+\frac{1}{NH}\sum_{i'=1}^{N}\mathcal{R}^{1:H}_{(i')}\right)^\top\right]\pi_{(\gamma,H)}(\mathrm{d}\theta,\mathrm{d}\Xi)\Big|$$

$$\le \left(\frac{3\gamma^{1/2}}{\mu^{1/2}}\sigma_\star+\gamma H\cdot\frac{\gamma HL}{2}\cdot\frac{8L^{1/2}}{\mu^{1/2}H^{1/2}}\sigma_\star\right)\cdot\frac{1200\gamma Q}{\mu}\sigma_\star^2 \ ,$$

and we obtain $|\mathrm{R}^{\theta,\xi}_{(c),4}| \le \left(\frac{3\gamma^{1/2}}{\mu^{1/2}}\sigma_\star+\frac{4\gamma^2 H^2 L^{3/2}\sigma_\star}{\mu^{1/2}H^{3/2}}\right)\frac{1200\gamma Q}{\mu}\sigma_\star^2 = \frac{3600Q\gamma^{3/2}+9600Q\gamma^3 H^{3/2}L^{3/2}}{\mu^{3/2}}\sigma_\star^3$. Finally, we have, by Lemma C.14,

$$|\mathrm{R}^{\theta,\xi}_{(c),5}| \le \frac{\gamma}{N}\sum_{i=1}^{N}\Big|\int \operatorname{tr} \mathbb{E}\left[\mathcal{R}^{1:H}_{(i)}\left(\frac{1}{H}\mathcal{R}^{1:H}_{(c)}-\frac{1}{NH}\sum_{i'=1}^{N}\mathcal{R}^{1:H}_{(i')}\right)^\top\right]\pi_{(\gamma,H)}(\mathrm{d}\theta,\mathrm{d}\Xi)\Big| \le \gamma\cdot\frac{2\cdot600^2\gamma^2 HQ^2}{\mu^2}\sigma_\star^4 \ ,$$

summing these four inequalities gives

$$|\operatorname{tr} \mathrm{R}^{\theta,\xi}_{(c)}| \le \frac{2400\gamma^2 Q(\mu^{1/2}+6\gamma^{1/2}\beta^{1/2})H^{1/2}}{\mu^{3/2}}\sigma_\star^3 + \frac{1800\gamma^{5/2}Q\zeta_2}{\mu^{3/2}}\sigma_\star^3 + \frac{4800\gamma^3 L^{3/2}H^{1/2}Q}{\mu^{3/2}}\sigma_\star^3$$

$$+ \frac{3600Q\gamma^{3/2}+9600Q\gamma^3 H^{3/2}L^{3/2}}{\mu^{3/2}}\sigma_\star^3 + \frac{2\cdot600^2\gamma^3 HQ^2}{\mu^2}\sigma_\star^4 \ ,$$

and the result follows from $\gamma\beta^{1/2}H^{1/2} \le 1/12$ and $\gamma H(L+\mu) \le 1/48$.

**Bound on $\mathrm{R}^\xi_{(c,c')}$.** We bound each term of $|\operatorname{tr} \mathrm{R}^\xi_{(c,c')}| = |\operatorname{tr} \mathrm{R}^\xi_{(c,c'),1}+\operatorname{tr} \mathrm{R}^\xi_{(c',c),1}{}^\top+\operatorname{tr} \mathrm{R}^\xi_{(c,c'),2}+\operatorname{tr} \mathrm{R}^\xi_{(c',c),2}{}^\top+\operatorname{tr} \mathrm{R}^\xi_{(\star,c'),3}+\operatorname{tr} \mathrm{R}^\xi_{(\star,c),3}{}^\top+\operatorname{tr} \mathrm{R}^\xi_{(c,c'),4}+\mathrm{R}^\xi_{(c',c),4}{}^\top+\operatorname{tr} \mathrm{R}^\xi_{(c,c'),5}| \le |\operatorname{tr} \mathrm{R}^\xi_{(c,c'),1}|+|\operatorname{tr} \mathrm{R}^\xi_{(c',c),1}{}^\top|+|\operatorname{tr} \mathrm{R}^\xi_{(c,c'),2}|+|\operatorname{tr} \mathrm{R}^\xi_{(c',c),2}|+|\operatorname{tr} \mathrm{R}^\xi_{(\star,c'),3}|+|\operatorname{tr} \mathrm{R}^\xi_{(\star,c),3}{}^\top|+|\operatorname{tr} \mathrm{R}^\xi_{(c,c'),4}|+|\operatorname{tr} \mathrm{R}^\xi_{(c',c),4}|+|\operatorname{tr} \mathrm{R}^\xi_{(c,c'),5}|$. First, by Lemma C.12, and Lemma C.14,

$$|\operatorname{tr} \mathrm{R}^\xi_{(c,c'),1}| \le \frac{1}{H}\Big|\int \operatorname{tr} \Delta^\Gamma_{(c)}\left(\theta-\theta^\star\right)\mathbb{E}\left[\mathcal{R}^{1:H}_{(c')}-\frac{1}{N}\sum_{i'=1}^{N}\mathcal{R}^{1:H}_{(i')}\right]^\top \pi_{(\gamma,H)}(\mathrm{d}\theta,\mathrm{d}\Xi)\Big| \le \frac{1}{H}\cdot\zeta_2\cdot\frac{3\gamma^{1/2}}{\mu^{1/2}}\sigma_\star\cdot\frac{1200\gamma HQ}{\mu}\sigma_\star^2 \ ,$$

which gives $|\operatorname{tr} \mathrm{R}^\xi_{(c,c'),1}| \le \frac{3600\gamma^{3/2}Q}{\mu^{3/2}}\sigma_\star^3$. Then, using Lemma 4.6, Lemma C.11, and Lemma C.14, we have that

$$|\operatorname{tr} \mathrm{R}^\xi_{(c,c'),2}| \le \frac{1}{H}\Big|\int \operatorname{tr} \widetilde{\mathrm{C}}^{1:H}_{(c)}\left(\xi_{(c)}-\xi^\star_{(c)}\right)\mathbb{E}\left[\mathcal{R}^{1:H}_{(c')}-\frac{1}{N}\sum_{i'=1}^{N}\mathcal{R}^{1:H}_{(i')}\right]^\top \pi_{(\gamma,H)}(\mathrm{d}\theta,\mathrm{d}\Xi)\Big|$$

$$\le \frac{1}{H}\cdot\frac{\gamma(H-1)L}{2}\cdot\frac{8L^{1/2}}{\mu^{1/2}H^{1/2}}\sigma_\star\cdot\frac{1200\gamma HQ}{\mu}\sigma_\star^2 \ ,$$

and thus $|\operatorname{tr} \mathrm{R}^\xi_{(c,c'),2}| \le \frac{9600\gamma^2 H^{1/2}L^{3/2}Q}{\mu^{3/2}}\sigma_\star^3$. The next term can be bounded using Lemma 4.6, Lemma C.11, and Lemma C.14,

$$|\operatorname{tr} \mathrm{R}^\xi_{(\star,c'),3}| = \frac{1}{NH}\sum_{i=1}^{N}\Big|\int \operatorname{tr} \widetilde{\mathrm{C}}^{1:H}_{(i)}\left(\xi_{(i)}-\xi^\star_{(i)}\right)\mathbb{E}\left[\frac{1}{H}\mathcal{R}^{1:H}_{(c')}-\frac{1}{NH}\sum_{i'=1}^{N}\mathcal{R}^{1:H}_{(i')}\right]^\top \pi_{(\gamma,H)}(\mathrm{d}\theta,\mathrm{d}\Xi)\Big|$$

$$\leq \frac{1}{H} \cdot \frac{\gamma(H-1)L}{2} \cdot \frac{8L^{1/2}}{\mu^{1/2}H^{1/2}}\sigma_\star \cdot \frac{1200\gamma HQ}{\mu}\sigma_\star^2 \ ,$$

which implies $|\operatorname{tr} \mathrm{R}_{(\star,c'),3}^{\xi}| \leq \frac{9600\gamma^2 H^{1/2}L^{3/2}Q}{\mu^{3/2}}\sigma_\star^3$. Moreover, we have, by Lemma C.13 and Lemma C.14,

$$|\operatorname{tr} \mathrm{R}_{(c,c'),4}^{\xi}| = \frac{1}{H^2}\left| \int \operatorname{tr} \mathbb{E}\left[ \left( \mathcal{R}_{(c)}^{1:H} - \frac{1}{N}\sum_{i=1}^{N}\mathcal{R}_{(i)}^{1:H} \right) \left( \varepsilon_{(c')}^{1:H} - \frac{1}{N}\sum_{i'=1}^{N}\varepsilon_{(i')}^{1:H} \right)^\top \right] \pi_{(\gamma,H)}(\mathrm{d}\theta,\mathrm{d}\Xi) \right|$$

$$\leq \frac{1}{H^2}\cdot\frac{1200\gamma HQ}{\mu}\sigma_\star^2\cdot\left( 2H^{1/2}\sigma_\star + \frac{12\gamma^{1/2}\beta^{1/2}H^{1/2}}{\mu^{1/2}}\sigma_\star \right) \ ,$$

and thus $|\operatorname{tr} \mathrm{R}_{(c,c'),4}^{\xi}| \leq \frac{4800\gamma Q\left(\mu^{1/2}+6\gamma^{1/2}\beta^{1/2}\right)}{\mu^{3/2}H^{1/2}}\sigma_\star^3$. Finally, Lemma C.14 gives

$$|\operatorname{tr} \mathrm{R}_{(c,c'),5}^{\xi}| = \frac{1}{H^2}\left| \int \operatorname{tr} \mathbb{E}\left[ \left( \mathcal{R}_{(c)}^{1:H} - \frac{1}{N}\sum_{i=1}^{N}\mathcal{R}_{(i)}^{1:H} \right) \left( \mathcal{R}_{(c')}^{1:H} - \frac{1}{N}\sum_{i'=1}^{N}\mathcal{R}_{(i')}^{1:H} \right)^\top \right] \pi_{(\gamma,H)}(\mathrm{d}\theta,\mathrm{d}\Xi) \right|$$

$$\leq \frac{1}{H^2}\frac{4\cdot600^2\gamma^2 H^2 Q^2}{\mu^2}\sigma_\star^4 \ .$$

Combining these bounds, we obtain

$$\operatorname{tr} \mathrm{R}_{(c,c')}^{\xi} \leq \frac{7200\gamma^{3/2}\zeta_2 Q}{\mu^{3/2}}\sigma_\star^3 + \frac{19200\gamma^2 H^{1/2}L^{3/2}Q}{\mu^{3/2}}\sigma_\star^3 + \frac{19200\gamma^2 H^{1/2}L^{3/2}Q}{\mu^{3/2}}\sigma_\star^3$$

$$+ \frac{9600\gamma Q\left(\mu^{1/2}+6\gamma^{1/2}\beta^{1/2}\right)}{\mu^{3/2}H^{1/2}}\sigma_\star^3 + \frac{4\cdot600^2\gamma^2 Q^2}{\mu^2}\sigma_\star^4 \ ,$$

and we conclude using $\gamma\beta^{1/2}H^{1/2} \leq 1/12$ and $\gamma H(L+\mu) \leq 1/48$. $\qquad\square$

**Corollary C.7.** *Assume A1, A2 and A5. Assume the step size $\gamma$ and the number of local updates $H$ satisfy $\gamma H(L+\mu) \leq 1/12$. Then, it holds that*

$$\|\mathrm{R}^\theta\| + \frac{\gamma(H-1)}{N}\sum_{c=1}^{N}\|\mathrm{R}_{(c)}^{\theta,\xi}\| + \frac{\gamma^2(H-1)^2}{N^2}\sum_{c,c'=1}^{N}\|\mathrm{R}_{(c,c')}^{\xi}\| \leq \frac{15080\gamma^{5/2}HQ}{\mu^{3/2}}\sigma_\star^3 + \frac{8\cdot600^2\gamma^4 H^2 Q^2}{\mu^2}\sigma_\star^4 \ .$$

*Proof.* We have, using the results from Lemma C.6,

$$\|\mathrm{R}^\theta\| + \frac{\gamma(H-1)}{N}\sum_{c=1}^{N}\|\mathrm{R}_{(c)}^{\theta,\xi}\| + \frac{\gamma^2(H-1)^2}{N^2}\sum_{c,c'=1}^{N}\|\mathrm{R}_{(c,c')}^{\xi}\|$$

$$\leq \operatorname{tr} \mathrm{R}^\theta + \frac{\gamma(H-1)}{N}\sum_{c=1}^{N}\operatorname{tr} \mathrm{R}_{(c)}^{\theta,\xi} + \frac{\gamma^2(H-1)^2}{N^2}\sum_{c,c'=1}^{N}\operatorname{tr} \mathrm{R}_{(c,c')}^{\xi}$$

$$\leq \left( 1080\gamma^{5/2}H + 6000\gamma^{3/2}\cdot\gamma H + \frac{8000\gamma^{1/2}}{H}\cdot\gamma^2 H^2 \right)\frac{Q}{\mu^{3/2}}\sigma_\star^3 + \left( 2\gamma^4 H^2 + 2\gamma^3 H\cdot\gamma H + 4\gamma^2\cdot\gamma^2 H^2 \right)\frac{600^2 Q^2}{\mu^2}\sigma_\star^4 \ ,$$

and the result follows. $\qquad\square$

## C.3. Upper bound on covariance matrices – Proof of Lemma 4.7 Theorem 4.8

In this section, we derive an upper bound on SCAFFOLD's global iterates' error covariance $\|\bar{\mathbf{\Sigma}}^\theta\|$. To this end, we define

$$\mathrm{C}^\theta = \|\bar{\mathbf{\Sigma}}^\theta\| \ , \quad \mathrm{C}^{\theta,\xi} = \frac{1}{N}\sum_{c=1}^{N}\|\bar{\mathbf{\Sigma}}_{(c)}^{\theta,\xi}\| \ , \quad \mathrm{C}^{\xi,=} = \frac{1}{N}\sum_{c=1}^{N}\|\bar{\mathbf{\Sigma}}_{(c,c)}^{\xi}\| \ , \quad \mathrm{C}^{\xi,\neq} = \frac{1}{N(N-1)}\sum_{c\neq c'}\|\bar{\mathbf{\Sigma}}_{(c,c')}^{\xi}\| \ .$$

We also define the following quantity, relating the average norm of the noise injected at each step

$$\varsigma^\epsilon = \frac{1}{N}\sum_{c=1}^{N}\|\bar{\mathbf{\Sigma}}^\epsilon_{(c)}\| \ .$$

We now derive a system of inequations that relate all the quantities we just defined. This Lemma is a complete version of Lemma 4.7.

**Lemma C.8.** *Assume A1, A2 and A5. Assume the step size $\gamma$ and the number of local updates $H$ satisfy $\gamma H(L+\mu) \le 1/12$, then*

$$C^\theta \le (1-\gamma\mu)^H C^\theta + \gamma^2 H(H-1)LC^{\theta,\xi} + \frac{\gamma^4 H^2(H-1)^2 L^2}{4}\left(\frac{1}{N}C^{\xi,=} + \left(1-\frac{1}{N}\right)C^{\xi,\ne}\right) + \frac{\gamma^2}{N}\varsigma^\epsilon + \|R^\theta\| \ , \quad (68)$$

$$C^{\theta,\xi} \le 2\zeta_2 C^\theta + 4\gamma^3 H(H-1)^2 L^2\left(\frac{1}{N}C^{\xi,=} + \left(1-\frac{1}{N}\right)C^{\xi,\ne}\right) + \frac{4\gamma}{NH}\varsigma^\epsilon + \frac{2}{N}\sum_{c=1}^{N}\|R^{\theta,\xi}_{(c)}\| \ , \quad (69)$$

$$\frac{1}{N}C^{\xi,=} + \left(1-\frac{1}{N}\right)C^{\xi,\ne} \le 2\zeta_2^2 C^\theta + \frac{9}{NH^2}\varsigma^\epsilon + 4\zeta_2\gamma(H-1)LC^{\theta,\xi} + \frac{2}{N^2}\sum_{c,c'=1}^{N}\|R^\xi_{(c,c')}\| \ . \quad (70)$$

*Proof.* **Parameter Covariance.** Taking the operator norm of Lemma C.3 and using triangle inequality and sub-multiplicativity of the matrix operator norm, we have

$$\|\bar{\mathbf{\Sigma}}^\theta\| \le \|\bar{\Gamma}\bar{\mathbf{\Sigma}}^\theta\bar{\Gamma}\| + \frac{\gamma H}{N}\sum_{c=1}^{N}\|\bar{\Gamma}\bar{\mathbf{\Sigma}}^{\theta,\xi}_{(c)}\widetilde{C}^{1:H}_{(c)}\| + \|\widetilde{C}^{1:H}_{(c)}\bar{\mathbf{\Sigma}}^{\xi,\theta}_{(c)}\bar{\Gamma}\|$$

$$+ \frac{\gamma^2 H^2}{N^2}\sum_{c=1}^{N}\sum_{c'=1}^{N}\|\widetilde{C}^{1:H}_{(c)}\bar{\mathbf{\Sigma}}^\xi_{(c,c')}\widetilde{C}^{1:H}_{(c')}\| + \frac{\gamma^2}{N}\|\bar{\mathbf{\Sigma}}^\epsilon\| + \|R^\theta\|$$

$$\le \|\bar{\Gamma}\|\|\bar{\mathbf{\Sigma}}^\theta\|\|\bar{\Gamma}\| + \frac{\gamma H}{N}\sum_{c=1}^{N}\|\bar{\Gamma}\|\|\bar{\mathbf{\Sigma}}^{\theta,\xi}_{(c)}\|\|\widetilde{C}^{1:H}_{(c)}\| + \|\widetilde{C}^{1:H}_{(c)}\|\|\bar{\mathbf{\Sigma}}^{\xi,\theta}_{(c)}\|\|\bar{\Gamma}\|$$

$$+ \frac{\gamma^2 H^2}{N^2}\sum_{c=1}^{N}\sum_{c'=1}^{N}\|\widetilde{C}^{1:H}_{(c)}\|\|\bar{\mathbf{\Sigma}}^\xi_{(c,c')}\|\|\widetilde{C}^{1:H}_{(c')}\| + \frac{\gamma^2}{N}\|\bar{\mathbf{\Sigma}}^\epsilon\| + \|R^\theta\| \ .$$

This gives, using Lemma C.10, Lemma C.11,

$$C^\theta \le (1-\gamma\mu)^H C^\theta + \frac{\gamma^2}{N}\|\bar{\mathbf{\Sigma}}^\epsilon\| + \|R^\theta\| + \frac{\gamma H}{N}\sum_{c=1}^{N}\left\{\|\bar{\mathbf{\Sigma}}^{\theta,\xi}_{(c)}\|\cdot\frac{\gamma(H-1)L}{2} + \frac{\gamma(H-1)L}{2}\cdot C^{\theta,\xi}\right\}$$

$$+ \frac{\gamma^2 H^2}{N^2}\sum_{c=1}^{N}\frac{\gamma(H-1)L}{2}\cdot C^{\xi,=}\cdot\frac{\gamma(H-1)L}{2} + \frac{\gamma^2 H^2}{N^2}\sum_{c=1}^{N}\sum_{c'=1}^{N}\frac{\gamma(H-1)L}{2}\cdot C^{\xi,\ne}\cdot\frac{\gamma(H-1)L}{2}$$

$$\le (1-\gamma\mu)^H C^\theta + \frac{\gamma^2}{N}\|\bar{\mathbf{\Sigma}}^\epsilon\| + \|R^\theta\|$$

$$+ \gamma^2 H(H-1)LC^{\theta,\xi} + \frac{\gamma^4 H^2(H-1)^2}{4N}C^{\xi,=} + \frac{\gamma^4 H^2(H-1)^2 L^2}{4}\left(1-\frac{1}{N}\right)C^{\xi,\ne} \ .$$

**Parameter-Control Variate Covariance.** By Lemma C.4, we have

$$\|\bar{\mathbf{\Sigma}}^{\theta,\xi}_{(c)}\| \le \|\bar{\Gamma}\|\|\bar{\mathbf{\Sigma}}^\theta\|\|\Delta^\Gamma_{(c)}\| + \|\bar{\Gamma}\|\|\bar{\mathbf{\Sigma}}^{\theta,\xi}_{(c)}\|\|\widetilde{C}^{1:H}_{(c)}\|$$

$$+ \frac{1}{N}\sum_{i'=1}^{N}\|\bar{\Gamma}\|\|\bar{\mathbf{\Sigma}}^{\theta,\xi}_{(i')}\|\|\widetilde{C}^{1:H}_{(i')}\| + \frac{\gamma H}{N}\sum_{i=1}^{N}\|\widetilde{C}^{1:H}_{(i)}\|\|\bar{\mathbf{\Sigma}}^{\xi,\theta}_{(i)}\|\|\Delta^\Gamma_{(c)}\|$$

$$+ \frac{\gamma H}{N}\sum_{i=1}^{N}\|\widetilde{C}^{1:H}_{(i)}\|\|\bar{\mathbf{\Sigma}}^\xi_{(i,c)}\|\|\widetilde{C}^{1:H}_{(c)}\| + \frac{\gamma H}{N^2}\sum_{i=1}^{N}\sum_{i'=1}^{N}\|\widetilde{C}^{1:H}_{(i)}\|\|\bar{\mathbf{\Sigma}}^\xi_{(i,i')}\|\|\widetilde{C}^{1:H}_{(i')}\| + \frac{\gamma}{NH}\|\bar{\mathbf{\Sigma}}^\epsilon_{(c)} - \bar{\mathbf{\Sigma}}^\epsilon\| + \|R^{\theta,\xi}_{(c)}\| \ ,$$

Averaging this inequality for $c = 1$ to $N$ and using Lemma C.10, Lemma C.11, and Lemma C.12 gives

$$
\begin{aligned}
\mathrm{C}^{\theta,\xi} \leq{} & (1-\gamma\mu)^H \cdot \zeta_2 \cdot \|\bar{\boldsymbol{\Sigma}}^\theta\| + (1-\gamma\mu)^H \cdot \gamma(H-1)L \cdot \mathrm{C}^{\theta,\xi} + \gamma H \cdot \gamma(H-1)L \cdot \zeta_2 \cdot \mathrm{C}^{\theta,\xi} \\
& + \gamma H \cdot \gamma(H-1)L \cdot \Big(\frac{1}{N}\mathrm{C}^{\xi,=} + \Big(1-\frac{1}{N}\Big)\mathrm{C}^{\xi,\neq}\Big) \cdot \gamma(H-1)L \\
& + \frac{\gamma H}{N^2}\sum_{i=1}^{N}\sum_{i'=1}^{N}\gamma(H-1)L \cdot \Big(\frac{1}{N}\mathrm{C}^{\xi,=} + \Big(1-\frac{1}{N}\Big)\mathrm{C}^{\xi,\neq}\Big)\cdot\gamma(H-1)L + \frac{\gamma}{NH}\|\bar{\boldsymbol{\Sigma}}^\epsilon_{(c)} - \bar{\boldsymbol{\Sigma}}^\epsilon\| + \|\mathrm{R}^{\theta,\xi}_{(c)}\| \ ,
\end{aligned}
$$

which gives

$$
\begin{aligned}
\mathrm{C}^{\theta,\xi} \leq{} & \zeta_2 \cdot \mathrm{C}^\theta + \gamma(H-1)L\mathrm{C}^{\theta,\xi} + \gamma^2 H(H-1)L\zeta_2 \cdot \mathrm{C}^{\theta,\xi} + 2\gamma^3 H(H-1)^2 L^2\Big(\frac{1}{N}\mathrm{C}^{\xi,=}+\Big(1-\frac{1}{N}\Big)\mathrm{C}^{\xi,\neq}\Big) \\
& + \frac{\gamma}{NH}\|\bar{\boldsymbol{\Sigma}}^\epsilon_{(c)} - \bar{\boldsymbol{\Sigma}}^\epsilon\| + \frac{1}{N}\sum_{c=1}^{N}\|\mathrm{R}^{\theta,\xi}_{(c)}\| \ ,
\end{aligned}
$$

and the second inequality follows from $\gamma HL + \gamma^2 H(H-1)L\zeta_2 \leq 1/2$.

**Control variate covariance.** By Lemma C.5, we have

$$
\begin{aligned}
\|\bar{\boldsymbol{\Sigma}}^\xi_{(c,c')}\| \leq{} & \|\Delta^\Gamma_{(c)}\|\|\bar{\boldsymbol{\Sigma}}^\theta\|\|\Delta^\Gamma_{(c')}\| + \frac{1}{NH^2}\|\bar{\boldsymbol{\Sigma}}^\epsilon_{(c)}\| + \frac{1}{NH^2}\|\bar{\boldsymbol{\Sigma}}^\epsilon_{(c')}\| + \frac{1}{NH^2}\|\bar{\boldsymbol{\Sigma}}^\epsilon\| \\
& + \|\Delta^\Gamma_{(c)}\|\|\bar{\boldsymbol{\Sigma}}^{\theta,\xi}_{(c')}\|\|\widetilde{\mathrm{C}}^{1:H}_{(c')}\| + \frac{1}{N}\sum_{i'=1}^{N}\|\Delta^\Gamma_{(c)}\|\|\bar{\boldsymbol{\Sigma}}^{\theta,\xi}_{(i')}\|\|\widetilde{\mathrm{C}}^{1:H}_{(i')}\| + \|\widetilde{\mathrm{C}}^{1:H}_{(c)}\|\|\bar{\boldsymbol{\Sigma}}^{\xi,\theta}_{(c)}\|\|\Delta^\Gamma_{(c')}\| + \frac{1}{N}\sum_{i=1}^{N}\|\widetilde{\mathrm{C}}^{1:H}_{(i)}\|\|\bar{\boldsymbol{\Sigma}}^{\xi,\theta}_{(i)}\|\|\Delta^\Gamma_{(c')}\| \\
& + \|\widetilde{\mathrm{C}}^{1:H}_{(c)}\|\|\bar{\boldsymbol{\Sigma}}^\xi_{(c,c')}\|\|\widetilde{\mathrm{C}}^{1:H}_{(c')}\| + \frac{1}{N}\sum_{i'=1}^{N}\|\widetilde{\mathrm{C}}^{1:H}_{(c)}\|\|\bar{\boldsymbol{\Sigma}}^\xi_{(c,i')}\|\|\widetilde{\mathrm{C}}^{1:H}_{(i')}\| + \frac{1}{N}\sum_{i=1}^{N}\|\widetilde{\mathrm{C}}^{1:H}_{(i)}\|\|\bar{\boldsymbol{\Sigma}}^\xi_{(i,c')}\|\|\widetilde{\mathrm{C}}^{1:H}_{(c')}\| \\
& + \frac{1}{N^2}\sum_{i=1}^{N}\sum_{i'=1}^{N}\|\widetilde{\mathrm{C}}^{1:H}_{(i)}\|\|\bar{\boldsymbol{\Sigma}}^\xi_{(i,i')}\|\|\widetilde{\mathrm{C}}^{1:H}_{(i')}\| + \|\mathrm{R}^\xi_{(c,c')}\| \ .
\end{aligned}
$$

Averaging over all pairs $c, c' \in \{1, \ldots, N\}$ with $c \neq c'$, we have

$$
\begin{aligned}
\mathrm{C}^{\xi,\neq} \leq{} & \zeta_2 \cdot \mathrm{C}^\theta \cdot \zeta_2 + \frac{1}{NH^2}\varsigma^\epsilon + \frac{1}{NH^2}\varsigma^\epsilon + \frac{1}{NH^2}\varsigma^\epsilon + \zeta_2 \cdot \mathrm{C}^{\theta,\xi} \cdot \frac{\gamma(H-1)L}{2} + 2\zeta_2 \cdot \mathrm{C}^{\theta,\xi} \cdot \gamma(H-1)L \\
& + \frac{\gamma^2(H-1)^2 L^2}{4}\mathrm{C}^{\xi,\neq} + 2 \cdot \frac{\gamma^2(H-1)^2 L^2}{4}\Big(\frac{1}{N}\mathrm{C}^{\xi,=}+\Big(1-\frac{1}{N}\Big)\mathrm{C}^{\xi,\neq}\Big) \\
& + \frac{\gamma^2(H-1)^2 L^2}{4}\Big(\frac{1}{N}\mathrm{C}^{\xi,=}+\Big(1-\frac{1}{N}\Big)\mathrm{C}^{\xi,\neq}\Big) + \frac{1}{N(N-1)}\sum_{c \neq c'}\|\mathrm{R}^\xi_{(c,c')}\| \\
\leq{} & \zeta_2^2 \mathrm{C}^\theta + \frac{3}{NH^2}\varsigma^\epsilon + 3\zeta_2\gamma(H-1)L\mathrm{C}^{\theta,\xi} \\
& + \frac{\gamma^2(H-1)^2 L^2}{4}\mathrm{C}^{\xi,\neq} + \frac{3\gamma^2(H-1)^2 L^2}{4}\Big(\frac{1}{N}\mathrm{C}^{\xi,=}+\Big(1-\frac{1}{N}\Big)\mathrm{C}^{\xi,\neq}\Big) + \frac{1}{N(N-1)}\sum_{c \neq c'}\|\mathrm{R}^\xi_{(c,c')}\| \ .
\end{aligned}
$$

Bounding $\frac{\gamma^2(H-1)^2 L^2}{4}\mathrm{C}^{\xi,\neq} + 2 \cdot \frac{\gamma^2(H-1)^2 L^2}{4}\mathrm{C}^{\xi,\neq} + \frac{\gamma^2(H-1)^2 L^2}{4}\mathrm{C}^{\xi,\neq} \leq 1/2\mathrm{C}^{\xi,\neq}$, we obtain the third inequality of the lemma. With similar derivations, we bound the control variates' covariances

$$
\begin{aligned}
\mathrm{C}^{\xi,=} \leq{} & \zeta_2 \cdot \mathrm{C}^\theta \cdot \zeta_2 + \frac{1}{H^2}\varsigma^\epsilon + \frac{2}{NH^2}\varsigma^\epsilon + \frac{1}{NH^2}\varsigma^\epsilon + \zeta_2 \cdot \mathrm{C}^{\theta,\xi} \cdot \frac{\gamma(H-1)L}{2} + 2\zeta_2 \cdot \mathrm{C}^{\theta,\xi} \cdot \gamma(H-1)L \\
& + \frac{\gamma^2(H-1)^2 L^2}{4}\mathrm{C}^{\xi,=} + 2 \cdot \frac{\gamma^2(H-1)^2 L^2}{4}\Big(\frac{1}{N}\mathrm{C}^{\xi,=}+\Big(1-\frac{1}{N}\Big)\mathrm{C}^{\xi,\neq}\Big) \\
& + \frac{\gamma^2(H-1)^2 L^2}{4}\Big(\frac{1}{N}\mathrm{C}^{\xi,=}+\Big(1-\frac{1}{N}\Big)\mathrm{C}^{\xi,\neq}\Big) + \frac{1}{N}\sum_{c=1}^{N}\|\mathrm{R}^\xi_{(c,c)}\|
\end{aligned}
$$

$$\leq \zeta_2^2 C^\theta + \frac{4}{H^2} \varsigma^\epsilon + 3\zeta_2 \gamma (H-1) L C^{\theta,\xi}$$

$$+ \frac{\gamma^2 (H-1)^2 L^2}{4} C^{\xi,=} + \frac{3\gamma^2 (H-1)^2 L^2}{4} \left( \frac{1}{N} C^{\xi,=} + \left( 1 - \frac{1}{N} \right) C^{\xi,\neq} \right) + \frac{1}{N} \sum_{c=1}^{N} \| R_{(c,c)}^\xi \| \ .$$

Summing these two inequalities, we obtain

$$\frac{1}{N} C^{\xi,=} + \left( 1 - \frac{1}{N} \right) C^{\xi,\neq} \leq \zeta_2^2 C^\theta + \frac{8}{NH^2} \varsigma^\epsilon + 3\zeta_2 \gamma (H-1) L C^{\theta,\xi}$$

$$+ \gamma^2 (H-1)^2 L^2 \left( \frac{1}{N} C^{\xi,=} + \left( 1 - \frac{1}{N} \right) C^{\xi,\neq} \right) + \frac{1}{N^2} \sum_{c,c'=1}^{N} \| R_{(c,c')}^\xi \| \ .$$

Since $\gamma^2 (H-1)^2 L^2 \leq 1/12^2$, we obtain

$$\frac{1}{N} C^{\xi,=} + \left( 1 - \frac{1}{N} \right) C^{\xi,\neq} \leq 2\zeta_2^2 C^\theta + \frac{9}{NH^2} \varsigma^\epsilon + 4\zeta_2 \gamma (H-1) L C^{\theta,\xi} + \frac{2}{N^2} \sum_{c,c'=1}^{N} \| R_{(c,c')}^\xi \| \ ,$$

which is the third inequality of the lemma. $\qquad \square$

**Lemma C.9.** *Assume A1, A2, A3, A4, and A5. Furthermore, assume that $5\gamma(H-1)L\zeta_2 \leq \mu/2$ and $\gamma H(L+\mu) \leq 1/12$ and $\frac{\gamma\beta}{\mu} \leq 1/19$. Then, it holds that*

$$C^\theta \leq \frac{10\gamma}{N\mu} \sigma_\star^2 + \frac{2}{\gamma\mu H} \left( \| R^\theta \| + \frac{\gamma(H-1)}{N} \sum_{c=1}^{N} \| R_{(c)}^{\theta,\xi} \| + \frac{\gamma^2 (H-1)^2}{N^2} \sum_{c,c'=1}^{N} \| R_{(c,c')}^\xi \| \right) \ .$$

*Proof.* Plugging (70) in (69), we obtain

$$C^{\theta,\xi} \leq 2\zeta_2 C^\theta + 4\gamma^3 H(H-1)^2 L^2 \left( 2\zeta_2^2 C^\theta + \frac{9\varsigma^\epsilon}{NH^2} + 4\zeta_2 \gamma (H-1) L C^{\theta,\xi} + \frac{2}{N^2} \sum_{c,c'=1}^{N} \| R_{(c,c')}^\xi \| \right) + \frac{4\gamma}{NH} \varsigma^\epsilon + \frac{2}{N} \sum_{c=1}^{N} \| R_{(c)}^{\theta,\xi} \|$$

$$\leq 3\zeta_2 C^\theta + \frac{5\gamma}{NH} \varsigma^\epsilon + 16\gamma^4 H(H-1)^3 L^3 \zeta_2 C^{\theta,\xi} + \frac{8\gamma^3 H(H-1)^2 L^2}{N^2} \sum_{c,c'=1}^{N} \| R_{(c,c')}^\xi \| + \frac{2}{N} \sum_{c=1}^{N} \| R_{(c)}^{\theta,\xi} \| \ ,$$

where we used $\gamma H L \leq 1/12$ to bound $4\gamma^3 H(H-1)^2 L^2 \cdot 2\zeta_2^2 \leq \zeta_2$ and $4\gamma^3 H(H-1)^2 L^2 \cdot \frac{9}{NH^2} \leq \frac{\gamma}{NH}$. Using this inequality again, we have $16\gamma^4 H(H-1)^3 L^3 \zeta_2 \leq 1/12^2$. This allows to simplify the previous inequality, obtaining

$$C^{\theta,\xi} \leq 4\zeta_2 C^\theta + \frac{6\gamma}{NH} \varsigma^\epsilon + \frac{9\gamma^3 H(H-1)^2 L^2}{N^2} \sum_{c,c'=1}^{N} \| R_{(c,c')}^\xi \| + \frac{3}{N} \sum_{c=1}^{N} \| R_{(c)}^{\theta,\xi} \| \ . \tag{71}$$

Plugging this bound in (70), we obtain

$$\frac{1}{N} C^{\xi,=} + \left( 1 - \frac{1}{N} \right) C^{\xi,\neq} \leq 4\zeta_2^2 C^\theta + \frac{10}{NH^2} \varsigma^\epsilon + \frac{3}{N^2} \sum_{c,c'=1}^{N} \| R_{(c,c')}^\xi \| + \frac{12\zeta_2 \gamma (H-1) L}{N} \sum_{c=1}^{N} \| R_{(c)}^{\theta,\xi} \| \ , \tag{72}$$

where we used $4\zeta_2 \gamma (H-1) L \cdot 4\zeta_2 \leq 2\zeta_2^2$, $\quad 4\zeta_2 \gamma (H-1) L \cdot \frac{6\gamma}{NH} \leq \frac{1}{NH^2}$ $\quad$ and $4\zeta_2 \gamma (H-1) L \cdot \frac{9\gamma^3 H(H-1)^2 L^2}{N^2} \leq \frac{1}{N^2}$.

We now plug (71) and (72) in (68), which gives

$$C^\theta \leq (1-\gamma\mu)^H C^\theta + \left( \gamma^2 H(H-1) L + \frac{\gamma^4 H^2 (H-1)^2 L^2 \zeta_2}{4} \right) \cdot 4\zeta_2 C^\theta$$

$$+ \left( \frac{6\gamma^3 H(H-1) L}{NH} + \frac{10\gamma^4 H^2 (H-1)^2 L^2}{4NH^2} + \frac{\gamma^2}{N} \right) \varsigma^\epsilon$$

$$+ \left( \gamma^2 H(H-1)L \cdot 9\gamma^3 H(H-1)^2 L^2 + \frac{3\gamma^4 H^2 (H-1)^2 L^2}{4} \right) \frac{1}{N^2} \sum_{c,c'=1}^{N} \|\mathrm{R}_{(c,c')}^{\xi}\|$$

$$+ \left( 3\gamma^2 H(H-1)L + \frac{\gamma^4 H^2 (H-1)^2 L^2}{4} \cdot 12\zeta_2 \gamma(H-1)L \right) \frac{1}{N} \sum_{c=1}^{N} \|\mathrm{R}_{(c)}^{\theta,\xi}\| + \|\mathrm{R}^{\theta}\| \ ,$$

which can be simplified using $\gamma H L \leq 1/12$ to obtain

$$\mathrm{C}^{\theta} \leq (1-\gamma\mu)^H \mathrm{C}^{\theta} + 5\gamma^2 H(H-1)L\zeta_2 \mathrm{C}^{\theta} + \frac{2\gamma^2}{N}\varsigma^{\epsilon} + \|\mathrm{R}^{\theta}\| + \frac{\gamma(H-1)}{N}\sum_{c=1}^{N}\|\mathrm{R}_{(c)}^{\theta,\xi}\| + \frac{\gamma^2(H-1)^2}{N^2}\sum_{c,c'=1}^{N}\|\mathrm{R}_{(c,c')}^{\xi}\| \ .$$

Now, using $\gamma H \mu \leq 1$, we have $(1-\gamma\mu)^H \leq 1 - \gamma\mu H/2$. Consequently, we have $(1-\gamma\mu)^H \mathrm{C}^{\theta} + 5\gamma^2 H(H-1)\zeta_2 \mathrm{C}^{\theta} \leq 1 - \gamma H(\mu - 5\gamma(H-1)L\zeta_2$. Since we assumed $5\gamma(H-1)L\zeta_2 \leq \mu/2$, we obtain

$$\mathrm{C}^{\theta} \leq (1-\gamma\mu H/2)\mathrm{C}^{\theta} + \frac{2\gamma^2}{N}\varsigma^{\epsilon} + \|\mathrm{R}^{\theta}\| + \frac{\gamma(H-1)}{N}\sum_{c=1}^{N}\|\mathrm{R}_{(c)}^{\theta,\xi}\| + \frac{\gamma^2(H-1)^2}{N^2}\sum_{c,c'=1}^{N}\|\mathrm{R}_{(c,c')}^{\xi}\| \ .$$

We then bound the variance term using Lemma C.13, which implies that

$$\varsigma^{\epsilon} \leq H\sigma_{\star}^2 + \frac{28\gamma\beta H}{\mu}\sigma_{\star}^2 \ .$$

Plugging this bound in the previous inequality, we obtain

$$\frac{\gamma\mu H}{2}\mathrm{C}^{\theta} \leq \frac{2\gamma^2 H}{N}\sigma_{\star}^2 + \frac{56\gamma^3\beta H}{N\mu}\sigma_{\star}^2 + \|\mathrm{R}^{\theta}\| + \frac{\gamma(H-1)}{N}\sum_{c=1}^{N}\|\mathrm{R}_{(c)}^{\theta,\xi}\| + \frac{\gamma^2(H-1)^2}{N^2}\sum_{c,c'=1}^{N}\|\mathrm{R}_{(c,c')}^{\xi}\| \ ,$$

which gives the first inequality of the theorem. $\qquad\square$

**Theorem 4.8** (Restated). *Assume A1, A2, A3, A4, and A5. Furthermore, assume that $\gamma HL\zeta_2 \leq \mu/10$, $\gamma H(L+\mu) \leq 1/48$ and $\gamma\beta \leq \mu/19$. Then, it holds that*

$$\mathrm{C}^{\theta} \leq \frac{10\gamma}{N\mu}\sigma_{\star}^2 + \frac{6 \cdot 15080\gamma^{3/2}Q}{\mu^{5/2}}\sigma_{\star}^3 + \frac{48 \cdot 600^2\gamma^3 HQ^2}{\mu^3}\sigma_{\star}^4 \ .$$

*Proof.* The result follows from Lemma C.9 and Corollary C.7. $\qquad\square$

## C.4. Bounds on intermediate quantities

### C.4.1. BOUND ON MATRICES

**Lemma C.10.** *Bound on $\Gamma_{(c)}$'s powers Let $h > 0$, $\gamma \geq 0$, recall $\Gamma_{(c)} = \mathrm{Id} - \gamma\nabla^2 f_{(c)}(\theta^{\star})$. Assume A1, A2, and that $\gamma \leq 1/L$, then it holds that*

$$\|\Gamma_{(c)}^h\| \leq (1-\gamma\mu)^h \ .$$

*Proof.* Follows from A1 and A2 with $\gamma \leq 1/L$. $\qquad\square$

**Lemma C.11.** *Let $h > 0$, $\gamma \geq 0$, recall $\Gamma_{(c)} = \mathrm{Id} - \gamma\nabla^2 f_{(c)}(\theta^{\star})$. Assume A1, A2, and that $\gamma \leq 1/L$, then it holds that*

$$\|\widetilde{\mathrm{C}}_{(c)}^{1:H}\| \leq \frac{\gamma(H-1)L}{2} \ ,$$

*Proof.* Recall that $\widetilde{\mathrm{C}}_{(c)}^{1:H} = \mathrm{Id} - \frac{1}{H}\sum_{h=0}^{H-1}\Gamma_{(c)}^{H-h-1}$. Since for any (square) matrix $A$ any $\gamma > 0$ and any $k \in \mathbb{N}^*$ we get that

$$\mathrm{Id} - (I-\gamma A)^k = \gamma A \sum_{\ell=0}^{k-1}(\mathrm{Id} - \gamma A)^{\ell},$$

we obtain

$$\widetilde{\mathrm{C}}_{(c)}^{1:H} = \frac{1}{H} \sum_{h=0}^{H-1} \left( \mathrm{Id} - (\mathrm{Id} - \gamma \nabla^2 f_{(c)}(\theta^\star))^{H-h-1} \right) = \frac{\gamma}{H} \nabla^2 f_{(c)}(\theta^\star) \sum_{h=0}^{H-1} (H - h - 1)(\mathrm{Id} - \gamma \nabla^2 f_{(c)}(\theta^\star))^h \ . \quad (73)$$

Using the triangle inequality, A2 and Lemma C.10, we obtain

$$\|\widetilde{\mathrm{C}}_{(c)}^{1:H}\| = \frac{\gamma L}{H} \sum_{h=0}^{H-1} (H - h - 1)(1 - \gamma \mu)^h \ ,$$

and the lemma follows from $\sum_{h=0}^{H-1} h = \frac{H(H-1)}{2}$. $\qquad \square$

**Lemma C.12.** *Let $h > 0$, $\gamma \geq 0$, recall $\Gamma_{(c)} = \mathrm{Id} - \gamma \nabla f_{(c)}(\theta^\star)$. Assume A1, A2, A4, and that $\gamma \leq 1/L$, then it holds that*

$$\|\Delta_{(c)}^\Gamma\| \leq \zeta_2 \ .$$

*Proof.* We have, using Lemma F.1,

$$\frac{1}{\gamma H} \left( \Gamma_{(c)}^H - \bar{\Gamma} \right) = \frac{1}{\gamma H N} \sum_{i=1}^N \left( \left( \mathrm{Id} - \gamma \nabla f_{(c)}(\theta^\star) \right)^H - \left( \mathrm{Id} - \gamma \nabla f_{(i)}(\theta^\star) \right)^H \right)$$

$$= \frac{1}{HN} \sum_{i=1}^N \sum_{h=0}^H \left( \mathrm{Id} - \gamma \nabla f_{(c)}(\theta^\star) \right)^{h-1} \left( \nabla f_{(c)}(\theta^\star) - \nabla f_{(i)}(\theta^\star) \right) - \left( \mathrm{Id} - \gamma \nabla f_{(i)}(\theta^\star) \right)^{H-h-1} \ .$$

The result follows from taking the norm, using triangle inequality, Lemma C.10, and A4. $\qquad \square$

### C.4.2. BOUND ON THE NOISE TERMS

**Lemma C.13.** *Assume A1, A2 and A5. Let $\gamma > 0$, $H > 0$, such that $\gamma H(L + \mu) \leq 1/12$, then*

$$\int \mathbb{E}\left[ \|\varepsilon_{(c)}^{1:H}\|^2 \right] \pi_{(\gamma,H)}(\mathrm{d}\theta, \mathrm{d}\Xi) \leq H\sigma_\star^2 + \frac{28\gamma\beta H}{\mu} \sigma_\star^2 \ .$$

*Proof.* Since $\varepsilon_{(c)}^h$ is a martingale difference sequence, we have

$$\mathbb{E}\left[ \|\varepsilon_{(c)}^{1:H}\|^2 \right] = \sum_{h=0}^{H-1} \mathbb{E}\left[ \|\Gamma_{(c)}^{H-h-1} \varepsilon_{(c)}^{h+1}\|^2 \right] \leq \sum_{h=0}^{H-1} \|\Gamma_{(c)}^{H-h-1}\| \mathbb{E}\left[ \|\varepsilon_{(c)}^{h+1}\|^2 \right] \ . \quad (74)$$

By Lemma C.10, and A5, we have

$$\mathbb{E}\left[ \|\varepsilon_{(c)}^{1:H}\|^2 \right] \leq \sum_{h=0}^{H-1} (1 - \gamma\mu)^h \left( \sigma_\star^2 + \beta \|\mathsf{T}_{(c)}^h(\theta; \xi_{(c)}) - \theta^\star\|^2 \right) \ . \quad (75)$$

Integrating over the stationary distribution $\pi_{(\gamma,H)}$, and using Lemma 4.6 gives the result. $\qquad \square$

### C.4.3. BOUND ON THE REMAINDERS

**Lemma C.14.** *Assume A1, A2, A3, and A5. Let $\gamma > 0$, $H > 0$, such that $\gamma H(L + \mu) \leq 1/12$, then*

$$\int \mathbb{E}\left[ \|\mathcal{R}_{(c)}^{1:H}\| \right] \pi_{(\gamma,H)}(\mathrm{d}\theta, \mathrm{d}\Xi) \leq \frac{28\gamma HQ}{\mu} \sigma_\star^2 \ .$$

*If $\gamma L \leq 1/48$, $\gamma H(L + \mu) \leq 1/24$, $\gamma H^{1/2}\beta^{1/2} \leq 1/12$ and $\gamma\beta \leq L/12$, then it also holds that*

$$\int \mathbb{E}\left[ \|\mathcal{R}_{(c)}^{1:H}\|^2 \right] \pi_{(\gamma,H)}(\mathrm{d}\theta, \mathrm{d}\Xi) \leq \frac{600^2 \gamma^2 H^2 Q^2}{\mu^2} \sigma_\star^4 \ .$$

*Proof.* Taking the norm of $\mathcal{R}_{(c)}^{1:H}$, and using the triangle inequality, A3, and Lemma C.10, we have

$$\|\mathcal{R}_{(c)}^{1:H}\| \leq \sum_{h=0}^{H-1} \left\| \Gamma_{(c)}^{H-h-1} \bar{D}_{(c)}^{3,h} \left( \mathsf{T}_{(c)}^h(\theta; \xi_{(c)}, Z_{(c)}^{1:h}) - \theta^\star \right)^{\otimes 2} \right\| \leq \sum_{h=0}^{H-1} Q \| \mathsf{T}_{(c)}^h(\theta; \xi_{(c)}, Z_{(c)}^{1:h}) - \theta^\star \|^2 .$$

Integrating over the stationary distribution and taking the expectation, and using Lemma 4.6, we obtain the first inequality. The second inequality follows from similar computations, using Jensen's inequality to bound

$$\|\mathcal{R}_{(c)}^{1:H}\|^2 \leq H \sum_{h=0}^{H-1} \left\| \Gamma_{(c)}^{H-h-1} \bar{D}_{(c)}^{3,h} \left( \mathsf{T}_{(c)}^h(\theta; \xi_{(c)}, Z_{(c)}^{1:h}) - \theta^\star \right)^{\otimes 2} \right\|^2 \leq H \sum_{h=0}^{H-1} Q^2 \| \mathsf{T}_{(c)}^h(\theta; \xi_{(c)}, Z_{(c)}^{1:h}) - \theta^\star \|^4 ,$$

and the result follows from taking the expectation and integrating over SCAFFOLD's stationary distribution, then using Lemma B.5 to bound each term of the sum. $\quad\square$

## D. Non-Asymptotic Rates for SCAFFOLD – Proof of Theorem 4.9

**Theorem 4.9** (Restated). *Assume A1, A2, A3, A4, and A5. Furthermore, assume that $\gamma H L \zeta_2 \leq \mu/10$, $\gamma H(L+\mu) \leq 1/48$ and $\gamma\beta \leq \mu/19$. Then, the mean squared error of SCAFFOLD's global iterates, initialized with $\theta^0 \in \mathbb{R}^d$ and $\xi_{(1)} = \cdots = \xi_{(N)} = 0 \in \mathbb{R}^d$ is*

$$\mathbb{E}\left[\|\theta^T - \theta^\star\|^2\right] \leq \left(1 - \frac{\gamma\mu}{4}\right)^{HT} \left\{ 2\|\theta^0 - \theta^\star\|^2 + 2\gamma^2 H^2 \zeta_1^2 + \frac{64\sigma_\star^2}{L\mu} \right\}$$
$$+ \frac{20 d\gamma}{N\mu}\sigma_\star^2 + \frac{12 \cdot 15080 d\gamma^{3/2} Q}{\mu^{5/2}}\sigma_\star^3 + \frac{96 \cdot 600^2 d\gamma^3 H Q^2}{\mu^3}\sigma_\star^4 .$$

*Proof.* Let $\hat{\theta}^0 \in \mathbb{R}^d$ and $\hat{\xi}_{(1)}^0, \cdots, \hat{\xi}_{(N)}^0 \in \mathbb{R}^d$ be sampled from SCAFFOLD's stationary distribution

$$\hat{\mathrm{X}}^0 = \left(\hat{\theta}^0, \hat{\xi}_{(1)}^0, \cdots, \hat{\xi}_{(N)}^0\right) \sim \pi_{(\gamma, H)} .$$

For an i.i.d. sequence $\{Z_{(1:N)}^{t,1:H}\}_{t\in\mathbb{N}}$ determining the randomness of the algorithm, where $Z_{(c)}^{t,h} \sim \nu_{(c)}$ for $c \in \{1, \ldots, N\}$ and $h \in \{0, \ldots, H\}$, we define two sequences, starting respectively from $\mathrm{X}^0 = \left(\theta^0, \xi_{(1)}^0, \cdots, \xi_{(N)}^0\right)$ and $\hat{\mathrm{X}}^0 = \left(\hat{\theta}^0, \hat{\xi}_{(1)}^0, \cdots, \hat{\xi}_{(N)}^0\right)$, and following the recursion for $t \geq 0$,

$$\mathrm{X}^{t+1} = \left(\theta^{t+1}, \xi_{(1)}^{t+1}, \cdots, \xi_{(N)}^{t+1}\right) = \mathsf{S}\left(\mathrm{X}^t; Z_{(1:N)}^{t+1,1:H}\right) ,$$
$$\hat{\mathrm{X}}^{t+1} = \left(\hat{\theta}^{t+1}, \hat{\xi}_{(1)}^{t+1}, \cdots, \hat{\xi}_{(N)}^{t+1}\right) = \mathsf{S}\left(\hat{\mathrm{X}}^t; Z_{(1:N)}^{t+1,1:H}\right) .$$

The first sequence are the actual iterates of SCAFFOLD, while the second one is its counterpart with the same realization of noise, but initialized in the stationary distribution. By definition of the stationary distribution, all iterations of this second sequence also follow the stationary distribution, i.e., for all $t \geq 0$,

$$\hat{\mathrm{X}}^t \sim \pi_{(\gamma, H)} .$$

We can thus decompose the error in two parts

$$\mathbb{E}\left[\|\theta^T - \theta^\star\|^2\right] \leq 2\mathbb{E}[\|\theta^T - \hat{\theta}^T\|^2] + 2\mathbb{E}[\|\hat{\theta}^T - \theta^\star\|^2] , \qquad (76)$$

where we recall $\mathrm{X}^\star = \left(\theta^\star, \xi_{(1)}^\star, \ldots, \xi_{(N)}^\star\right)$ is the optimal vector. The first term is an optimization term, which determines the distance from current iterate to an iterate drawn in the stationary distribution. The second term is the variance in the stationary distribution. We now bound each of these two terms.

**Bounding the optimization term.** Using Lemma 4.1 recursively with the natural filtration of the process $\{X^t\}_{t\geq 0}$, we can bound the first term as

$$
\begin{aligned}
2\mathbb{E}[\|\theta^T - \theta^\star\|^2] &\leq 2\mathbb{E}[\|X^T - \hat{X}^T\|^2] \\
&\leq 2\left(1 - \frac{\gamma\mu}{4}\right)^{HT}\|X^0 - \hat{X}^0\|^2 \\
&\leq 4\left(1 - \frac{\gamma\mu}{4}\right)^{HT}\|X^0 - X^\star\|^2 + 4\left(1 - \frac{\gamma\mu}{4}\right)^{HT}\|\hat{X}^0 - X^\star\|^2 .
\end{aligned}
\tag{77}
$$

Integrating (77) over the stationary distribution and using Corollary B.1, we have

$$
2\mathbb{E}[\|\theta^T - \theta^\star\|^2] \leq 4\left(1 - \frac{\gamma\mu}{4}\right)^{HT}\|X^0 - X^\star\|^2 + \left(1 - \frac{\gamma\mu}{4}\right)^{HT}\frac{64\gamma}{\mu}\sigma_\star^2 .
\tag{78}
$$

**Bounding the variance term.** For the second term, we use Theorem 4.8 to bound

$$
\begin{aligned}
2\mathbb{E}[\|\hat{\theta}^T - \theta^\star\|^2] = 2\int \|\theta - \theta^\star\|^2 \pi_{(\gamma,H)}(\mathrm{d}\theta, \mathrm{d}\Xi) &\leq 2d\|\bar{\Sigma}^\theta\| \\
&\leq \frac{20d\gamma}{N\mu}\sigma_\star^2 + \frac{12\cdot 15080 d\gamma^{3/2}Q}{\mu^{5/2}}\sigma_\star^3 + \frac{96\cdot 600^2 d\gamma^3 HQ^2}{N\mu^3}\sigma_\star^4 .
\end{aligned}
\tag{79}
$$

**Final rate.** Plugging (78) and (79) in (76), we obtain

$$
\begin{aligned}
\mathbb{E}\left[\|\theta^T - \theta^\star\|^2\right] &\leq \left(1 - \frac{\gamma\mu}{4}\right)^{HT}\left(\|\theta - \theta^\star\|^2 + \frac{\gamma^2 H^2}{N}\sum_{c=1}^N \|\xi_{(c)} - \xi_{(c)}^\star\|^2 + \frac{64\gamma}{\mu}\sigma_\star^2\right) \\
&\quad + \frac{20d\gamma}{N\mu}\sigma_\star^2 + \frac{16\cdot 15080\gamma^{3/2}Q}{\mu^{5/2}}\sigma_\star^3 + \frac{96\cdot 600^2 d\gamma^3 HQ^2}{N\mu^3}\sigma_\star^4 ,
\end{aligned}
$$

and the result follows by taking $\xi_{(c)} = 0$ for all $c\in\{1,\dots,N\}$ and using the fact that $\xi_{(c)}^\star = -\nabla f_{(c)}(\theta^\star)$. $\qquad\square$

**Corollary 4.10** (Restated). *Let $\epsilon > 0$. With Theorem 4.9's assumptions, we can set $\gamma \lesssim \min(\frac{1}{L}, \frac{N\mu\epsilon^2}{\sigma_\star^2}, \frac{\mu^{5/3}\epsilon^{4/3}}{Q^{2/3}\sigma_\star^2}, \frac{L^{1/2}\mu^{3/2}\epsilon}{Q\sigma_\star^2})$ and $H \lesssim \frac{\sigma_\star^2\min(1,\mu/\zeta_2)}{L\mu\epsilon^2}\max(\frac{1}{N}, \frac{Q^{2/3}\epsilon^{2/3}}{\mu}, \frac{QL^{1/2}\epsilon}{\mu^{1/2}})$. Then, SCAFFOLD guarantees $\mathbb{E}[\|\theta^T - \theta^\star\|^2] \leq \epsilon^2$ for $T \gtrsim \frac{L}{\mu}\max(1, \zeta_2/\mu)\log(\frac{\|\theta^0 - \theta^\star\|^2 + \zeta_1^2/L^2}{\epsilon^2})$, and the number of stochastic gradients computed by each client is*

$$
TH \lesssim \frac{\sigma_\star^2}{\mu^2\epsilon^2}\max\left(\frac{1}{N}, \frac{Q^{2/3}\epsilon^{2/3}}{\mu}, \frac{QL^{1/2}\epsilon}{\mu^{1/2}}\right)\log\left(\frac{\psi_0}{\epsilon^2}\right) ,
$$

*where $\psi_0 = \|\theta^0 - \theta^\star\|^2 + \zeta_1^2/L^2 + \sigma_\star^2/(L\mu)$.*

*Proof.* By Theorem 4.9, we have

$$
\mathbb{E}\left[\|\theta^T - \theta^\star\|^2\right] \lesssim \left(1 - \frac{\gamma\mu}{4}\right)^{HT}\left\{\|\theta^0 - \theta^\star\|^2 + \gamma^2 H^2\zeta_1 + \frac{\gamma\sigma_\star^2}{\mu}\right\} + \frac{\gamma}{N\mu}\sigma_\star^2 + \frac{\gamma^{3/2}Q}{\mu^{5/2}}\sigma_\star^3 + \frac{\gamma^3 HQ^2}{\mu^3}\sigma_\star^4 .
$$

For the last three terms to be smaller than $\epsilon^2$, we require

$$
\gamma \lesssim \min\left(\frac{1}{L}, \frac{N\mu\epsilon^2}{\sigma_\star^2}, \frac{\mu^{5/3}\epsilon^{4/3}}{Q^{2/3}\sigma_\star^2}, \frac{L^{1/2}\mu^{3/2}\epsilon}{Q\sigma_\star^2}\right) ,
$$

where the first two conditions follow from the terms in $\gamma$ and $\gamma^{3/2}$ and the last one follows from the term in $\gamma^3 H$, using the property $\gamma H \leq 1/L$. To choose $H$, we remark that, in Theorem 4.9, we require $\gamma HL \lesssim 1$ and $\gamma HL\zeta_2 \lesssim \mu$. Thus, choosing the largest step size possible, we have

$$
H \lesssim \frac{1}{\gamma L}\min(1, \mu/\zeta_2) \lesssim \frac{\sigma_\star^2\min(1, \mu/\zeta_2)}{L\mu\epsilon^2}\max\left(\frac{1}{N}, \frac{Q^{2/3}\epsilon^{2/3}}{\mu}, \frac{QL^{1/2}\epsilon}{\mu^{1/2}}\right) .
\tag{80}
$$

This gives

$$\mathbb{E}\left[\|\theta^T - \theta^\star\|^2\right] \lesssim \left(1 - \frac{\gamma\mu}{4}\right)^{HT}\left\{\|\theta^0 - \theta^\star\|^2 + \gamma^2 H^2 \zeta_1^2 + \frac{\gamma\sigma_\star^2}{\mu}\right\} + \epsilon^2 \ .$$

Now, we choose $\gamma$ and $H$ as big as possible, which gives

$$T \gtrsim \frac{L}{\mu}\max(1, \zeta_2/\mu)\log\left(\frac{\|\theta^0 - \theta^\star\|^2 + \zeta_1^2/L^2 + \sigma_\star^2/(L\mu)}{\epsilon^2}\right) \ , \tag{81}$$

such that $\mathbb{E}\left[\|\theta^T - \theta^\star\|^2\right] \leq \epsilon^2$. Since each client computes $TH$ gradients, the result follows from (80) and (81). $\qquad\square$

## E. Bias of SCAFFOLD

We now give first-order expression of the bias of SCAFFOLD. For $\theta \in \mathbb{R}^d$ and $\Xi = (\xi_{(1)}, \ldots, \xi_{(N)}) \in \mathbb{R}^{N \times d}$, we define the bias in the stationary distribution of the parameters and control variates as

$$\bar{\boldsymbol{b}}^\theta \triangleq \int (\theta - \theta^\star)\,\pi_{(\gamma,H)}(\mathrm{d}\theta, \mathrm{d}\Xi) \ , \qquad b \triangleq \int \left(\xi_{(c)} - \xi_{(c)}^\star\right)\pi_{(\gamma,H)}(\mathrm{d}\theta, \mathrm{d}\Xi) \ .$$

Alike the tensors defined in (57) and (58), we define the following tensor that will be used to expand the gradients to third order,

$$\bar{D}_{(c)}^{4,h}(\theta) = \int_0^1 (1-t)^2 \nabla^4 f_{(c)}(\theta^\star + t\left(\mathsf{T}_{(c)}^h(\theta; \xi_{(c)}, Z_{(c)}^{1:h}) - \theta^\star\right))\mathrm{d}t \ . \tag{82}$$

As in Appendix C, we will often denote $\bar{D}_{(c)}^{4,h} = \bar{D}_{(c)}^{4,h}(\theta_{(c)}^h)$ for conciseness.

**Lemma E.1.** *Assume the step size $\gamma$ and the number of local updates $H$ satisfy $\gamma H(L+\mu) \leq 1/12$. Under these conditions, it holds that*

$$\Gamma_{(c)}^h = \mathrm{Id} - \gamma h \nabla^2 f_{(c)}(\theta^\star) + O(\gamma^2 H^2) \ ,$$
$$\bar{\Gamma} = \mathrm{Id} - \gamma H \nabla^2 f(\theta^\star) + O(\gamma^2 H^2) \ ,$$
$$\widetilde{\mathsf{C}}_{(c)}^{1:H} = \frac{\gamma(H-1)}{2}\nabla^2 f_{(c)}(\theta^\star) + O(\gamma^2 H) \ .$$

*Proof.* The first equality follows from expanding $\Gamma_{(c)} = \left(\mathrm{Id} - \gamma \nabla^2 f_{(c)}(\theta^\star)\right)^h$ using the Binomial theorem and the fact that $\mathrm{Id}$ and $\nabla^2 f_{(c)}(\theta^\star)$ commute. Then, terms of higher order can be bounded by bounding the remainder terms using the exponential series and the fact that $\nabla^2 f_{(c)}(\theta^\star) \preccurlyeq L$ with $\gamma H L \leq 1$. The second equality follows from the first one with $h = H$ and $\nabla^2 f(\theta^\star) = \frac{1}{N}\sum_{c=1}^N \nabla^2 f_{(c)}(\theta^\star)$. The last identity follows from (73) and Lemma C.10. $\qquad\square$

### E.1. Bias on the Control Variates

**Lemma E.2** (Bias of Control Variates). *Assume A1, A2 and A5. Let $c \in \{1, \ldots, N\}$, $Z = Z_{(1:N)}^{1:H}$ be i.i.d. random variables. Assume the step size $\gamma$ and the number of local updates $H$ satisfy $\gamma H(L + \mu) \leq 1/12$. Under these conditions, control variates' bias satisfies*

$$\bar{\boldsymbol{b}}_{(c)}^\xi = (\nabla^2 f_{(c)}(\theta^\star) - \nabla^2 f(\theta^\star))\bar{\boldsymbol{b}}^\theta + O(\gamma) \ , \tag{83}$$

*Proof.* Let $\theta \in \mathbb{R}^d$ and $\xi_{(1)}, \ldots, \xi_{(N)} \in \mathbb{R}^d$. For $c \in \{1, \ldots, N\}$ and $h \in \{0, \ldots, H\}$, define

$$\theta_c^h = \mathsf{T}_{(c)}^h(\theta; \xi_{(c)}) = \theta - \gamma \sum_{\ell=0}^{h-1}\left\{\nabla f_{(c)}(\mathsf{T}_{(c)}^\ell(\theta; \xi_{(c)})) + \varepsilon_{(c)}^{\ell+1}(Z_{(c)}^{\ell+1}) + \xi_{(c)}\right\} \ , \tag{84}$$

$$\xi_{(c)}^+ = \mathsf{V}_{(c)}(\xi_{(c)}; \theta, Z_{(1:N)}^{1:H}) = \xi_{(c)} + \frac{1}{\gamma H}\left(\mathsf{T}_{(c)}^H(\theta; \xi_{(c)}, Z_{(c)}^{1:H}) - \mathsf{T}(\theta; \xi_{(1:N)}, Z_{(1:N)}^{1:H})\right) \ . \tag{85}$$

First, we derive a first-order expansion of the local updates error. Using Corollary B.1 to bound the remainder term, we have

$$\theta_c^h - \theta^\star = \theta - \theta^\star - \gamma \sum_{\ell=0}^{h-1} \left\{ \nabla f_{(c)}(\theta^\star) + \varepsilon_{(c)}^{\ell+1}(Z_{(c)}^{\ell+1}) + \xi_{(c)} + O(\gamma^{1/2}) \right\} \tag{86}$$

$$= \theta - \theta^\star - \gamma h(\xi_{(c)} - \xi_{(c)}^\star) - \gamma \sum_{\ell=0}^{h-1} \varepsilon_{(c)}^{\ell+1}(Z_{(c)}^{\ell+1}) + O(\gamma^{3/2}h) \ . \tag{87}$$

Then, we recall the expression of the control variates updates

$$\xi_{(c)}^+ = \xi_{(c)} + \frac{1}{\gamma H} \left( \mathsf{T}_{(c)}^H(\theta; \xi_{(c)}, Z_{(c)}^{1:H}) - \mathsf{T}(\theta; \xi_{(1:N)}, Z_{(1:N)}^{1:H}) \right) \tag{88}$$

$$= \xi_{(c)} - \frac{1}{H} \left( \sum_{h=0}^{H} \nabla f_{(c)}(\theta_c^h) + \varepsilon_{(c)}^{\ell+1}(Z_{(c)}^{\ell+1}) + \xi_{(c)} - \frac{1}{N} \sum_{i=0}^{N} \sum_{h=0}^{H} \nabla f_{(i)}(\theta_i^h) + \varepsilon_{(i)}^{\ell+1}(Z_{(i)}^{\ell+1}) + \xi_{(i)} \right) \ . \tag{89}$$

Taking the conditional expectation, and expanding the gradients we have

$$\mathbb{E}\left[\xi_{(c)}^+\right] = -\frac{1}{H} \left( \sum_{h=0}^{H} \nabla f_{(c)}(\theta^\star) + \nabla^2 f_{(c)}(\theta^\star) \mathbb{E}\left[\theta_c^h - \theta^\star\right] - \frac{1}{N} \sum_{i=0}^{N} \sum_{h=0}^{H} \nabla^2 f_{(i)}(\theta^\star) \mathbb{E}\left[\theta_i^h - \theta^\star\right] + O(\gamma) \right)$$

$$= -\frac{1}{H} \left( \sum_{h=0}^{H} -\xi_{(c)}^\star + \nabla^2 f_{(c)}(\theta^\star) \mathbb{E}\left[\theta_c^h - \theta^\star\right] - \frac{1}{N} \sum_{i=0}^{N} \sum_{h=0}^{H} \nabla^2 f_{(i)}(\theta^\star) \mathbb{E}\left[\theta_i^h - \theta^\star\right] + O(\gamma) \right) \ .$$

Since $\mathbb{E}\left[\theta_i^h - \theta^\star\right] = \theta - \theta^\star + O(\gamma H)$, we have

$$\mathbb{E}\left[\xi_{(c)}^+ - \xi_{(c)}^\star\right] = (\nabla^2 f_{(c)}(\theta^\star) - \nabla^2 f(\theta^\star))(\theta - \theta^\star) + O(\gamma) \ , \tag{90}$$

and the result of the lemma follows. $\qquad\square$

### E.2. Expression of the Parameter's Variance – Proof of Lemma 5.1

**Lemma 5.1** (Restated). *Assume A1, A2, A3, A4, A5. Furthermore, assume that the step size $\gamma$ and number of local updates $H$ satisfy $\gamma H L \zeta_2 \le \mu/10$ and $\gamma H (L + \mu) \le 1/48$ and $\gamma \beta \le \mu/19$. Then, it holds that, for $c \ne c' \in \{1, \ldots, N\}$,*

$$\bar{\mathbf{\Sigma}}^\theta = \frac{\gamma}{N} \mathbf{A} \mathcal{C}(\theta^\star) + O(\gamma^2 H + \gamma^{3/2}) \ ,$$

$$\bar{\mathbf{\Sigma}}_{(c)}^{\theta,\xi} = \frac{\gamma}{N} \mathbf{A} \mathcal{C}(\theta^\star)(\nabla^2 f_{(c)}(\theta^\star) - \nabla^2 f(\theta^\star)) + \frac{\gamma}{N} (\mathcal{C}_c(\theta^\star) - \mathcal{C}(\theta^\star)) + O(\gamma^2 H + \gamma^{3/2}) \ ,$$

$$\bar{\mathbf{\Sigma}}_{(c,c)}^\xi = \left(1 - \frac{2}{N}\right) \frac{1}{H} \mathcal{C}_c(\theta^\star) + \frac{1}{NH} \mathcal{C}(\theta^\star) + O(\gamma) \ ,$$

$$\bar{\mathbf{\Sigma}}_{(c,c')}^\xi = \frac{1}{NH} (\mathcal{C}(\theta^\star) - \mathcal{C}_c(\theta^\star) - \mathcal{C}_{c'}(\theta^\star)) + O(\gamma) \ ,$$

*where $\mathbf{A} = \left(\mathrm{Id} \otimes \nabla^2 f(\theta^\star) + \nabla^2 f(\theta^\star) \otimes \mathrm{Id}\right)^{-1}$, $\mathcal{C}_c(\theta^\star) = \mathbb{E}[(\varepsilon_{(c)}^{Z_{(c)}}(\theta^\star))^{\otimes 2}]$ and $\mathcal{C}(\theta^\star) = \frac{1}{N} \sum_{c=1}^{N} \mathcal{C}_c(\theta^\star)$.*

*Proof.* Lemma C.9 gives $\bar{\mathbf{\Sigma}}^\theta = O(\gamma)$. Then, by Lemma 4.7-(69) and Lemma 4.6, it holds that $\frac{1}{N} \sum_{c=1}^{N} \|\bar{\mathbf{\Sigma}}_{(c)}^{\theta,\xi}\| = O(\gamma)$. Finally, Lemma 4.6 ensures that $\bar{\mathbf{\Sigma}}_{(c,c')}^\xi = O(1/H)$ for all $c, c' \in \{1, \ldots, N\}$.

We recall the expression from Lemma C.3,

$$\mathbf{\Sigma}^\theta = \bar{\Gamma} \bar{\mathbf{\Sigma}}^\theta \bar{\Gamma} + \frac{\gamma H}{N} \sum_{c=1}^{N} \left( \bar{\Gamma} \bar{\mathbf{\Sigma}}_{(c)}^{\theta,\xi} \widetilde{\mathrm{C}}_{(c)}^{1:H} + \widetilde{\mathrm{C}}_{(c)}^{1:H} \bar{\mathbf{\Sigma}}_{(c)}^{\xi,\theta} \bar{\Gamma} \right) + \frac{\gamma^2 H^2}{N^2} \sum_{c=1}^{N} \sum_{c'=1}^{N} \widetilde{\mathrm{C}}_{(c)}^{1:H} \bar{\mathbf{\Sigma}}_{(c,c')}^\xi \widetilde{\mathrm{C}}_{(c')}^{1:H} + \frac{\gamma^2}{N} \bar{\mathbf{\Sigma}}^\epsilon + \mathrm{R}^\theta \ .$$

By Lemma E.1, to expand the matrices $\bar{\Gamma}$ and $\widetilde{\mathrm{C}}_{(c)}^{1:H}$ for $c \in \{1, \ldots, N\}$, and using $\gamma H = O(1)$, we thus have

$$\bar{\mathbf{\Sigma}}^\theta = \bar{\mathbf{\Sigma}}^\theta - \gamma H \nabla^2 f(\theta^\star) \bar{\mathbf{\Sigma}}^\theta - \gamma H \bar{\mathbf{\Sigma}}^\theta \nabla^2 f(\theta^\star) + \frac{\gamma^2}{N} \bar{\mathbf{\Sigma}}^\epsilon + O(\gamma^3 H^2) + O(\gamma^{5/2} H) \ , \tag{91}$$

where we also used Lemma C.6 to obtain $R^\theta = O(\gamma^{5/2}H)$. Finally, we expand $\bar{\Sigma}^\epsilon$ using A5 and Corollary B.1, which gives

$$\bar{\Sigma}^\epsilon = H\mathcal{C}(\theta^\star) + O(\gamma H) \ .$$

Plugging this equation in (91) and reorganizing the terms gives the result.

**Covariance of $\theta$ and $\xi_{(c)}$.** From Lemma C.4, recall

$$\bar{\Sigma}^{\theta,\xi}_{(c)} = \bar{\Gamma}\bar{\Sigma}^\theta\Delta^\Gamma_{(c)} + \bar{\Gamma}\bar{\Sigma}^{\theta,\xi}_{(c)}\widetilde{C}^{1:H}_{(c)} - \frac{1}{N}\sum_{i'=1}^{N}\bar{\Gamma}\bar{\Sigma}^{\theta,\xi}_{(i')}\widetilde{C}^{1:H}_{(i')} + \frac{\gamma H}{N}\sum_{i=1}^{N}\widetilde{C}^{1:H}_{(i)}\bar{\Sigma}^{\xi,\theta}_{(i)}\Delta^\Gamma_{(c)}$$

$$+ \frac{\gamma H}{N}\sum_{i=1}^{N}\widetilde{C}^{1:H}_{(i)}\bar{\Sigma}^\xi_{(i,c)}\widetilde{C}^{1:H}_{(c)} - \frac{\gamma H}{N^2}\sum_{i=1}^{N}\sum_{i'=1}^{N}\widetilde{C}^{1:H}_{(i)}\bar{\Sigma}^\xi_{(i,i')}\widetilde{C}^{1:H}_{(i')} + \frac{\gamma}{NH}\left(\bar{\Sigma}^\epsilon_{(c)} - \bar{\Sigma}^\epsilon\right) + R^{\theta,\xi}_{(c)} \ ,$$

which gives

$$\bar{\Sigma}^{\theta,\xi}_{(c)} = \bar{\Sigma}^\theta\Delta^\Gamma_{(c)} + \frac{\gamma}{NH}\left(\bar{\Sigma}^\epsilon_{(c)} - \bar{\Sigma}^\epsilon\right) + O(\gamma^2 H) + O(\gamma^{3/2}) \ ,$$

and the result follows.

**Covariance of control variates.** Similarly, we obtain

$$\bar{\Sigma}^\xi_{(c,c)} = \Delta^\Gamma_{(c)}\bar{\Sigma}^\theta\Delta^\Gamma_{(c')} + \frac{1}{H^2}\bar{\Sigma}^\epsilon_{(c)} - \frac{2}{NH^2}\bar{\Sigma}^\epsilon_{(c)} + \frac{1}{NH^2}\bar{\Sigma}^\epsilon + O(\gamma^2 H + \gamma^{3/2}) \ ,$$

$$\bar{\Sigma}^\xi_{(c,c')} = \Delta^\Gamma_{(c)}\bar{\Sigma}^\theta\Delta^\Gamma_{(c')} - \frac{1}{NH^2}\bar{\Sigma}^\epsilon_{(c)} - \frac{1}{NH^2}\bar{\Sigma}^\epsilon_{(c')} + \frac{1}{NH^2}\bar{\Sigma}^\epsilon + O(\gamma^2 H + \gamma^{3/2}) \ ,$$

and the last two identities follow. $\qquad\square$

### E.3. Bias on the Parameters – Proof of Theorem 5.3

**Lemma E.3.** *Assume A1, A2 and A5. Let $Z = Z^{1:H}_{(1:N)}$ be i.i.d. random variables. Assume the step size $\gamma$ and the number of local updates $H$ satisfy $\gamma H(L + \mu) \leq 1/12$. Under these conditions, it holds that*

$$\int \mathbb{E}\left[\left(\theta^h_{(c)} - \theta^\star\right)^{\otimes 2}\right]\pi_{(\gamma,H)}(\mathrm{d}\theta, \mathrm{d}\Xi) = \int\left(\theta - \theta^\star\right)^{\otimes 2}\pi_{(\gamma,H)}(\mathrm{d}\theta, \mathrm{d}\Xi) + U^h_{(c)} \ ,$$

*where $U^h_{(c)} = O(\gamma^2 H)$.*

*Proof.* To this end, we expand the gradient in $\theta^h_{(c)} - \theta^\star = \theta - \gamma\sum_{\ell=0}^{h-1}\left\{\nabla f_{(c)}(\theta^\ell_{(c)}) + \xi_{(c)} + \varepsilon^{\ell+1}_{(c)}\right\} - \theta^\star$, which gives

$$\left(\theta^h_{(c)} - \theta^\star\right)^{\otimes 2} = \left(\theta - \theta^\star - \gamma h(\xi_{(c)} - \xi^\star_{(c)}) - \gamma\varepsilon^{1:h}_{(c)} - \gamma\sum_{\ell=0}^{h-1}\bar{D}^{2,\ell}_{(c)}\left(\theta^\ell_{(c)} - \theta^\star\right)\right)^{\otimes 2} \ .$$

Expanding the square, we get the result with $U^h_{(c)}$ given by

$$U^h_{(c)} = -\gamma h\int\left(\theta - \theta^\star\right)\left(\xi_{(c)} - \xi^\star_{(c)} + \frac{1}{h}\sum_{\ell=0}^{h-1}\mathbb{E}\left[\bar{D}^{2,\ell}_{(c)}\left(\theta^\ell_{(c)} - \theta^\star\right)\right]\right)^\top\pi_{(\gamma,H)}(\mathrm{d}\theta, \mathrm{d}\Xi)$$

$$+ \gamma^2 h^2\int\left(\xi_{(c)} - \xi^\star_{(c)} + \frac{1}{h}\sum_{\ell=0}^{h-1}\bar{D}^{2,\ell}_{(c)}\left(\theta^\ell_{(c)} - \theta^\star\right)\right)^{\otimes 2}\pi_{(\gamma,H)}(\mathrm{d}\theta, \mathrm{d}\Xi) + \gamma^2\int\mathbb{E}\left[\left(\varepsilon^{1:h}_{(c)}\right)^{\otimes 2}\right]\pi_{(\gamma,H)}(\mathrm{d}\theta, \mathrm{d}\Xi)$$

$$+ \gamma^2\sum_{\ell=0}^{h-1}\int\mathbb{E}\left[\bar{D}^{2,\ell}_{(c)}\left(\theta^\ell_{(c)} - \theta^\star\right)\left(\varepsilon^{1:h}_{(c)}\right)^\top + \left(\varepsilon^{1:h}_{(c)}\right)\bar{D}^{2,\ell}_{(c)}\left(\theta^\ell_{(c)} - \theta^\star\right)^\top\right]\pi_{(\gamma,H)}(\mathrm{d}\theta, \mathrm{d}\Xi) \ ,$$

which satisfies $U^h_{(c)} = O(\gamma^2 h)$ by Corollary B.1, Lemma 4.6, Corollary B.4, Lemma B.5 and $\gamma HL \leq 1$. $\qquad\square$

**Lemma E.4.** *Assume A1, A2 and A5. Let $Z = Z_{(1:N)}^{1:H}$ be i.i.d. random variables. Assume the step size $\gamma$ and the number of local updates $H$ satisfy $\gamma H(L + \mu) \leq 1/12$. Under these conditions, it holds that*

$$\bar{b}^\theta = -\frac{1}{2N}\nabla^2 f(\theta^\star)^{-1}\nabla^3 f(\theta^\star)\bar{\Sigma}^\theta + O(\gamma^2 H + \gamma^{3/2}) \ .$$

*Proof.* By definition of the local updates, we have, for $h \in \{0, \ldots, H-1\}$, assuming $\theta_{(c)}^{h+1}$ is $\mathcal{F}_c^h$-measurable,

$$\mathbb{E}\left[\theta_{(c)}^{h+1} - \theta^\star \mid \mathcal{F}_c^h\right] = \theta_{(c)}^h - \theta^\star - \gamma\nabla f_{(c)}(\theta_{(c)}^h) - \gamma\xi_{(c)} \ .$$

Like in (65), we expand the gradient, but for one more order, and use $\xi_{(c)}^\star = -\nabla f_{(c)}(\theta^\star)$,

$$\mathbb{E}\left[\theta_{(c)}^{h+1} - \theta^\star \mid \mathcal{F}_c^h\right] = \theta_{(c)}^h - \theta^\star - \gamma\nabla^2 f_{(c)}(\theta^\star)\left(\theta_{(c)}^h - \theta^\star\right) - \frac{\gamma}{2}\nabla^3 f_{(c)}(\theta^\star)\left(\theta_{(c)}^h - \theta^\star\right)^{\otimes 2}$$
$$- \frac{\gamma}{2}\bar{D}_{(c)}^{4,h+1}\left(\theta_{(c)}^h - \theta^\star\right)^{\otimes 3} - \gamma\left(\xi_{(c)} - \xi_{(c)}^\star\right) \ .$$

Taking the expectation, unrolling this equality and averaging the result over $c = 1$ to $N$, we obtain

$$\mathbb{E}\left[\theta^+ - \theta^\star\right] = \bar{\Gamma}\left(\theta - \theta^\star\right) + \frac{\gamma H}{N}\sum_{c=1}^N \widetilde{C}_{(c)}^{1:H}\left(\xi_{(c)} - \xi_{(c)}^\star\right)$$
$$- \frac{\gamma}{2N}\sum_{c=1}^N\sum_{h=0}^{H-1}\Gamma_{(c)}^{H-h-1}\nabla^3 f_{(c)}(\theta^\star)\mathbb{E}\left[\left(\theta_{(c)}^h - \theta^\star\right)^{\otimes 2}\right] - \frac{\gamma}{2N}\sum_{c=1}^N\sum_{h=0}^{H-1}\Gamma_{(c)}^{H-h-1}\mathbb{E}\left[\bar{D}_{(c)}^{4,h}\left(\theta_{(c)}^h - \theta^\star\right)^{\otimes 3}\right] \ .$$

Integrating over the stationary distribution of SCAFFOLD and using Lemma E.3, we obtain

$$\bar{b}^\theta = \bar{\Gamma}\bar{b}^\theta + \frac{\gamma H}{N}\sum_{c=1}^N \widetilde{C}_{(c)}^{1:H}\bar{b}_{(c)}^\xi - \frac{\gamma}{N}\sum_{c=1}^N\sum_{h=0}^{H-1}\Gamma_{(c)}^{H-h-1}\nabla^3 f_{(c)}(\theta^\star)\left(\bar{\Sigma}^\theta + U_{(c)}^h\right) + W \ ,$$

where $W = -\frac{\gamma}{2N}\sum_{c=1}^N\sum_{h=0}^{H-1}\Gamma_{(c)}^{H-h-1}\bar{D}_{(c)}^{4,h}\left(\theta_{(c)}^h - \theta^\star\right)^{\otimes 3}$ satisfies $W = O(\gamma^{5/2}H)$ by A6 and Lemma B.5. Plugging in the expansions from Lemma E.1, we obtain

$$\bar{b}^\theta = \left(\mathrm{Id} - \gamma H\nabla^2 f(\theta^\star) + O(\gamma^2 H^2)\right)\bar{b}^\theta + \frac{\gamma H}{N}\sum_{c=1}^N\left(\frac{\gamma(H-1)}{2}\nabla^2 f_{(c)}(\theta^\star) + O(\gamma^2 H)\right)\bar{b}_{(c)}^\xi$$
$$- \frac{\gamma}{2N}\sum_{c=1}^N\sum_{h=0}^{H-1}\left(\mathrm{Id} + O(\gamma H)\right)\nabla^3 f_{(c)}(\theta^\star)\left(\bar{\Sigma}^\theta + U_{(c)}^h\right) + W \ ,$$

which gives

$$\gamma H\nabla^2 f(\theta^\star)\bar{b}^\theta = \frac{\gamma^2 H(H-1)}{2N}\sum_{c=1}^N\nabla^2 f_{(c)}(\theta^\star)\bar{b}_{(c)}^\xi - \frac{\gamma}{2N}\sum_{c=1}^N\sum_{h=0}^{H-1}\nabla^3 f_{(c)}(\theta^\star)\bar{\Sigma}^\theta + O(\gamma^3 H^2 + \gamma^{5/2}H) \ .$$

The result follows by multiplying by $(\gamma H\nabla^2 f_{(c)}(\theta^\star))^{-1}$ on both sides, and using Lemma E.2 to bound $\bar{b}_{(c)}^\xi$. $\qquad\square$

**Theorem 5.3** (Restated). *Assume A1, A2, A3, A4, A5, A6. Furthermore, assume that the step size $\gamma$ and number of local updates $H$ satisfy $\gamma(H-1)L\zeta_2 \leq \mu/10$ and $\gamma H(L+\mu) \leq 1/12$ and $\gamma\beta \leq \mu/19$. Then, the bias of SCAFFOLD is*

$$\bar{b}^\theta = -\frac{\gamma}{2N}\nabla^2 f(\theta^\star)^{-1}\nabla^3 f(\theta^\star)\mathbf{A}\mathcal{C}(\theta^\star) + O(\gamma^2 H + \gamma^{3/2}) \ .$$

*Proof.* The result follows by plugging the expression of $\bar{\Sigma}^\theta$ from Lemma 5.1 in Lemma E.4. $\qquad\square$

# F. Useful Lemmas

**Lemma F.1** (Matrix Product Coupling). *For any matrix-valued sequences $(M_k)_{k \in \mathbb{N}}$, $(M'_k)_{k \in \mathbb{N}}$ and for any $K \in \mathbb{N}$, it holds that*

$$\prod_{k=1}^{K} M_k - \prod_{k=1}^{K} M'_k = \sum_{k=1}^{K} \left\{ \prod_{\ell=1}^{k-1} M_\ell \right\} \left( M_k - M'_k \right) \left\{ \prod_{\ell=k+1}^{M} M'_\ell \right\} \ .$$

**Lemma F.2** (Projection). *Let $N > 0$, $\mathbf{x} = (x_1, \ldots, x_N)$ and $\mathbf{y} = (y_1, \ldots, y_N)$ with $x_c, y_c \in \mathbb{R}^d$ for $c \in \{1, \ldots, N\}$. We define $\bar{\mathbf{x}} = (\bar{x}, \ldots, \bar{x})$ and $\bar{\mathbf{y}} = (\bar{y}, \ldots, \bar{y})$ with $\bar{x} = N^{-1} \sum_{c=1}^{N} x_c$ and $\bar{y} = N^{-1} \sum_{c=1}^{N} y_c$. It holds that*

$$\| \bar{\mathbf{x}} - \bar{\mathbf{y}} \|^2 = \| \mathbf{x} - \mathbf{y} \|^2 - \| (\bar{\mathbf{x}} - \mathbf{x}) - (\bar{\mathbf{y}} - \mathbf{y}) \|^2 \ ,$$

*where $\| \cdot \|$ is the $\ell_2$-norm over $\mathbb{R}^{Nd}$. Since $\| \bar{\mathbf{x}} - \bar{\mathbf{y}} \|^2 = N \| \bar{x} - \bar{y} \|$, we also have*

$$\| \bar{x} - \bar{y} \|^2 = \frac{1}{N} \sum_{c=1}^{N} \{ \| x_c - y_c \|^2 - \| (\bar{x} - x_c) - (\bar{y} - y_c) \|^2 \} \ .$$

*Proof.* Expanding the norm, we have

$$\begin{aligned}
\| \bar{\mathbf{x}} - \bar{\mathbf{y}} \|^2 &= \| \mathbf{x} - \mathbf{y} + \bar{\mathbf{x}} - \mathbf{x} - \bar{\mathbf{y}} + \mathbf{y} \|^2 \\
&= \| \mathbf{x} - \mathbf{y} \|^2 + 2 \langle \mathbf{x} - \mathbf{y}, \bar{\mathbf{x}} - \mathbf{x} - \bar{\mathbf{y}} + \mathbf{y} \rangle + \| \bar{\mathbf{x}} - \mathbf{x} - \bar{\mathbf{y}} + \mathbf{y} \|^2 \\
&= \| \mathbf{x} - \mathbf{y} \|^2 + 2 \langle \bar{\mathbf{x}} - \bar{\mathbf{y}}, \bar{\mathbf{x}} - \mathbf{x} - \bar{\mathbf{y}} + \mathbf{y} \rangle - \| \bar{\mathbf{x}} - \mathbf{x} - \bar{\mathbf{y}} + \mathbf{y} \|^2 \ .
\end{aligned}$$

Then, we notice that

$$2 \langle \bar{\mathbf{x}} - \bar{\mathbf{y}}, \bar{\mathbf{x}} - \mathbf{x} - \bar{\mathbf{y}} + \mathbf{y} \rangle = 2 \sum_{c=1}^{N} \langle \bar{x} - \bar{y}, \bar{x} - x_i - \bar{y} + y_i \rangle = 2 \langle \bar{x} - \bar{y}, N(\bar{x} - \bar{y}) + \sum_{c=1}^{N} \{ y_i - x_i \} \rangle \ .$$

And we have $2 \langle \bar{x} - \bar{y}, N(\bar{x} - \bar{y}) + \sum_{c=1}^{N} \{ y_i - x_i \} \rangle = 0$ since $N\bar{x} = \sum_{c=1}^{N} x_i$ and $N\bar{y} = \sum_{c=1}^{N} y_i$. $\square$

