# OpenReview forum: "Scaffold with Stochastic Gradients: New Analysis with Linear Speed-Up"
_ICML.cc/2025/Conference — ICML 2025 poster_

### Official Review · Reviewer_cy3M · 2025-03-11

**Overall Recommendation:** 3

**Summary:**

This paper presents a novel analysis of the SCAFFOLD algorithm, a popular method in federated learning designed to address client heterogeneity. The authors show that the global parameters and control variates of SCAFFOLD form a Markov chain that converges to a stationary distribution, which allows them to establish that SCAFFOLD achieves linear speed-up with respect to the number of clients, up to higher-order terms in the step size. The analysis reveals that SCAFFOLD retains a small higher-order bias due to stochastic updates. The paper derives new non-asymptotic convergence rates and highlights that SCAFFOLD’s global parameters’ variance decreases inversely with the number of clients, enabling scalable and efficient federated learning. Additionally, the authors provide precise characterizations of the algorithm’s variance and bias in the stationary regime.

**Claims And Evidence:**

The claims made in this submission are generally supported by clear and convincing evidence.

**Essential References Not Discussed:**

N/A

**Experimental Designs Or Analyses:**

In the experimental setup, using the same step size $\gamma$ for both SCAFFOLD and FedAvg without tuning them individually does not seem very reasonable.

**Methods And Evaluation Criteria:**

The proposed methods and evaluation criteria generally make sense.

**Other Comments Or Suggestions:**

N/A

**Other Strengths And Weaknesses:**

Strengths:
1. The paper provides a novel Markov chain-based analysis of SCAFFOLD under stochastic gradient updates, which has not been explored in prior work. This originality strengthens the theoretical understanding of federated learning in stochastic settings.
2. The result showing linear speed-up for SCAFFOLD in stochastic settings without relying on restrictive assumptions (e.g., global step sizes or quadratic objectives) is a significant theoretical contribution.
3. The identification and quantification of higher-order bias caused by stochastic updates in SCAFFOLD is a new insight that extends the understanding of bias correction mechanisms in federated learning.

Weaknesses:
1. A key significance of the original SCAFFOLD by Karimireddy et al. (2020) is its independence from the heterogeneity of data distributions. However, the convergence rate of the one-learning-rate SCAFFOLD in this paper depends on the level of data heterogeneity $\zeta_1$ (Assumption 4 and Theorem 4.8).
2. The analytical framework is somewhat restricted, as it only applies to strongly convex objectives and assumes full client participation.
3. While it is understandable that this paper is primarily theoretical, the experiments appear relatively simple. Moreover, certain aspects of the experimental setup, such as step size tuning, are not well conducted (see comments in Experimental Designs Or Analyses).

**Questions For Authors:**

1. Can SCAFFOLD, without a global learning rate, handle partial client participation effectively?
2. Do the theoretical results in this paper imply linear speedup for FedAvg when using only a local learning rate?
3. Does the algorithmic framework strongly rely on convexity assumptions? Can it be extended to non-convex settings?

**Relation To Broader Scientific Literature:**

N/A

**Theoretical Claims:**

I do not carefully check the correctness of any proofs.

---

> ### Author Rebuttal · Authors · 2025-03-31
>
> Thank you for the positive evaluation of our paper!
> We are happy that you found that the Markov chain-based analysis of Scaffold is original, and that "this originality strengthens the theoretical understanding of federated learning in stochastic settings," that proving that Scaffold has linear speed-up without global step-size is a "significant theoretical contribution," and that our analysis "extends the understanding of bias correction mechanisms in federated learning".
>
> We address your concerns below.
>
> **"A key significance of the original SCAFFOLD by Karimireddy et al. (2020) is its independence from the heterogeneity of data distributions. However, the convergence rate of the one-learning-rate SCAFFOLD in this paper depends on the level of data heterogeneity   (Assumption 4 and Theorem 4.8)."**
> Thank you for allowing us to clarify this point. This is also the case in the original Scaffold analysis (see Remark 10 in [1]), and in analyses of related algorithms like Scaffnew [2].
> Similarly to [1, 2], heterogeneity appears in our results with a multiplicative factor $(1 - \gamma \mu/4)^{HT}$, and thus does not prevent convergence. It only has a small impact on the convergence rate, as it only affects logarithmic terms in the algorithm's complexity (as in [1, 2]).
> We insist that this measure of heterogeneity is not an additional assumption, as Assumption 4 always holds as long as a unique global optimum exists. We will replace this assumption and state it as a consequence of strong convexity, which implies that a unique global minimum exists.
>
> [1] Karimireddy et al. (2020). Scaffold: Stochastic controlled averaging for federated learning.
>
> [2] Mishchenko et al. (2022). Proxskip: Yes! local gradient steps provably lead to communication acceleration! finally!.
>
> **"The analytical framework is somewhat restricted, as it only applies to strongly convex objectives."**
> Analyzing strongly convex and smooth objective functions is a common approach in federated learning, as it allows for rigorous theoretical guarantees.
> While our focus is on this setting, we believe our methodology could serve as a foundation for extending convergence analysis to weaker assumptions, making this a promising direction for future research.
>
> **"While it is understandable that this paper is primarily theoretical, the experiments appear relatively simple. Moreover, certain aspects of the experimental setup, such as step size tuning, are not well conducted (see comments in Experimental Designs Or Analyses)."**
> As noted by the reviewer, our experiments primarily serve an illustrative purpose. We selected a representative step size and number of local iterations to demonstrate the key theoretical insights in practical settings.
> If the reviewer thinks this limitation is crucial for acceptance of our paper, we are happy to provide more results with more thorough hyperparameter tuning in the final version of the paper.
>
> We now answer your questions:
>
> **"1. Can SCAFFOLD, without a global learning rate, handle partial client participation effectively?"**
> Thank you for this insightful question. We believe that results similar to ours would still hold with partial participation and that the analytical framework we propose can be extended to this setting.
> Nonetheless, we refrained from including it in the paper, as we believe that it would divert attention from the core of this paper's study, which is the linear speed-up phenomenon.
>
> "**2.  Do the theoretical results in this paper imply linear speedup for FedAvg when using only a local learning rate?"**
> Yes, indirectly. The results on the stationary distribution $\pi^{(\gamma, H)}$ (that is, Lemma 4.6, Theorem 4.7, and the variance part of Theorem 4.8) would still hold when setting $\xi_c = 0$ throughout the analysis.
> This substitution would directly imply that FedAvg exhibits linear speed-up whenever SCAFFOLD does.
>
>
> **"3.  Does the algorithmic framework strongly rely on convexity assumptions? Can it be extended to non-convex settings?"**
> This is an exciting direction for future research. While we expect that viewing SCAFFOLD as a Markov chain and establishing convergence to a stationary distribution could be established without strong convexity, doing so would require fundamentally different analytical tools.
> Assuming that, without strong convexity, a stationary distribution would still exist, our current analysis would not allow to study it as is.
> Building an analysis in this setting (e.g., following ideas from [1]) is thus a promising direction for future research.
>
> [1] Yu et al. (2021). An analysis of constant step size SGD in the non-convex regime: Asymptotic normality and bias.

---

> > ### Comment · Reviewer_cy3M · 2025-04-02
> >
> > Thank the authors for the response. Could authors further explain why the original Scaffold analysis requires the heterogeneity assumption? I did not find the discussion in your paper. Additionally, I remain unconvinced by using the same step sizes for the tested algorithms, as it seems hard to justify what is a *representative* step size. Nevertheless, I appreciate the the theoretical contributions of this paper.

---

> > > ### Author Response · Authors · 2025-04-03
> > >
> > > Thank you for allowing us to clarify further. In the original Scaffold paper, like in our results, the convergence rate for the strongly-convex case (first case in their Theorem VII) an optimization error that decreases exponentially.
> > > Denoting $\mu$ the strong convexity constant, $L$ the smoothness constant, and $T$ the number of  term that decreases exponentially scales in $ \widetilde{D} \exp( - (\mu / L ) T ) $, where, using the notations from our manuscript,
> > >
> > > $$\widetilde{D} = \|\| \theta^0 - \theta^\star \|\|^2 + \frac{1}{N L^2} \sum_{c=1}^N \|\| \xi_c^0 - \nabla f_c(\theta^\star) \|\|^2 $$
> > >
> > > Taking $\xi_c^0 = 0$, the second term becomes
> > >
> > > $$\frac{1}{N L^2} \sum_{c=1}^N \|\| \xi_c^0 - \nabla f_c(\theta^\star) \|\|^2 = \frac{1}{N L^2} \sum_{c=1}^N \|\| \nabla f_c(\theta^\star) \|\|^2 = \frac{1}{L^2} \zeta_1^2$$
> > >
> > > which is exactly the term that we have in Theorem 4.8 when taking maximal step size.
> > >
> > > Again, we stress that the way we measure heterogeneity is not an assumption, but rather a **definition** of $\zeta_1^2 = \frac{1}{N} \sum_{c=1}^N \|\| \nabla f_c(\theta^\star) \|\|^2$. This quantity $\zeta_1$ is always defined, as long as the optimal point $\theta^\star$ is defined, which is always the case for strongly convex functions.
> > >
> > > Regarding experiments, the goal was to show that in a two numerical examples, Scaffold has linear speed-up, while FedAvg does not in the case where there is heterogeneity, as it suffers from heterogeneity bias. If you have specific suggestions as to how we could improve these experiments, we will gladly provide more numerical results and add them in the final version of the manuscript.
> > >
> > > Thank you again for your thorough review and your valuable insights.

---

### Official Review · Reviewer_LPDP · 2025-03-11

**Overall Recommendation:** 4

**Summary:**

The paper proposes an analysis for the Scaffold algorithm, a popular method for dealing with data heterogeneity in federated learning.  The authors first show that the global parameters and control variates define a Markov chain that converges to a stationary distribution in the Wasserstein distance.  Leveraging this result, they prove that Scaffold achieves linear speed-up in the number of clients up to higher-order terms in the step size.  The analysis also reveals that Scaffold retains a higher-order bias, similar to FedAvg, that does not decrease as the number of clients increases.

**Claims And Evidence:**

I think this is a great paper. I agree with the contributions and claims. Viewing scaffold through the lenses of MC is a great idea.

**Essential References Not Discussed:**

None

**Experimental Designs Or Analyses:**

This section is not needed (see my previous comments).

**Methods And Evaluation Criteria:**

In my opinion the computation part is superfluous. It essentially confirms the theoretical results. It better should or otherwise there are flaws in the math.
They also make a claim that it's better than FedAvg. This has already been established numerically in many other papers.

The value of this paper is in theory and thus Section 6 should be removed since it doesn't add anything interesting.

**Other Comments Or Suggestions:**

None

**Other Strengths And Weaknesses:**

None

**Questions For Authors:**

1. What is the purpose of Section 6?

**Relation To Broader Scientific Literature:**

See my comments above (MC and scaffold).

**Theoretical Claims:**

No, I did not. I checked the math in the main body, including all of the statements.

---

> ### Author Rebuttal · Authors · 2025-03-31
>
> Thank you for your thorough and constructive feedback. We greatly appreciate your recognition of our paper’s contribution and the innovative perspective provided by viewing Scaffold using Markov chain formalism. Below, we carefully address each of your remarks:
>
> **Claims and Evidence:**
> We appreciate your positive acknowledgment of our contributions and claims, particularly regarding the theoretical framing of Scaffold using Markov chain analysis.
>
> **Methods and Evaluation Criteria:**
> Thank you for your insightful comment on the experimental section. Our computational experiments primarily validate our theoretical findings rather than establish novel empirical superiority over existing methods like FedAvg, whose empirical performance is already well-documented in the literature. However, we respectfully suggest retaining Section 6, at least in a condensed form. Indeed, numerical validations, even when theoretically expected, offer essential practical insights into the assumptions and conditions underpinning our theoretical results. We will explicitly clarify the purpose of these experiments in the revised manuscript to underscore their complementary role.
>
> Considering your feedback, we will condense the experimental section, and stress its role as an illustration of our theoretical results. This adjustment aligns with the theoretical focus of our paper while providing an experimental component for practical reference.

---

> > ### Comment · Reviewer_LPDP · 2025-04-01
> >
> > Thanks for providing the answers. I have no further questions and comments.

---

### Official Review · Reviewer_QhCL · 2025-03-13

**Overall Recommendation:** 3

**Summary:**

This paper investigates the convergence properties of Scaffold, a federated learning method designed to reduce variance among clients. By analyzing the global iterates and control variates from the perspective of a Markov chain, the study establishes a novel non-asymptotic convergence rate for Scaffold with respect to the total number of clients.

**Claims And Evidence:**

I think the claims in the submission are mostly clear. The paper well illustrates the key contributions as stated in the introduction with sufficient evidence.

However, regarding Remark 2.1, I find the claim that "While this yields the desired linear speed-up by... and increasing the global one, it essentially reduces the algorithm to mini-batch SGD" somewhat confusing. To my understanding, this is not entirely accurate because federated algorithms involve locally iterative training. Reducing the local learning rate and performing multiple steps of local training does not equate to simply increasing the batch size in mini-batch SGD. The iterative nature of local updates in federated learning introduces additional dynamics and are not equal to standard mini-batch SGD.

**Essential References Not Discussed:**

There is a (recent) paper that also explores FL using a Markov chain framework:

Sun, Z., Zhang, Z., Xu, Z., Joshi, G., Sharma, P., & Wei, E. (2024). Debiasing Federated Learning with Correlated Client Participation.

**Experimental Designs Or Analyses:**

1. The authors discuss the non-vanishing bias of Scaffold in Section 5 and claim that Scaffold can eliminate heterogeneity bias. However, they do not provide further analysis or discussion on heterogeneity bias in the experimental section. It would be beneficial to include empirical results or insights that demonstrate how Scaffold addresses heterogeneity bias in practice.
2. The authors claim that Scaffold achieves linear speedup based on the results in Figure 1. However, this claim is not particularly convincing to me, as linear speedup with respect to the number of clients is not unique to Scaffold—FedAvg also exhibits this property. Therefore, the linear speedup of Scaffold does not stand out as a novel or exciting result.

**Methods And Evaluation Criteria:**

The main focus of this paper is on studying the convergence properties of existing methods. Consequently, it does not introduce any significant new methods or define evaluation criteria for comparative analysis.

**Other Comments Or Suggestions:**

The legends in Figure 1 are missing.

**Other Strengths And Weaknesses:**

Strengths: The paper is well written, and the theoretical part is mostly clear to follow.

Most Weaknesses have been discussed in previous sections, particularly about the theoretical claims, the claims of the paper and the empirical analysis.

**Questions For Authors:**

Please refer to previous sections. Addressing the issues in previous part would help me better understand and potentially change my evaluation of the paper.

**Relation To Broader Scientific Literature:**

This paper may contribute to the privacy-preserving related machine learning applications.

**Theoretical Claims:**

I have reviewed all the theoretical results in the main paper and briefly examined the theoretical results in the appendix. I have several questions regarding this:
1. I believe the theoretical results in this paper rely on some strong assumptions, such as the bounded third and fourth derivatives. While I do not think these assumptions necessarily make the paper less convincing, it is true that achieving better convergence results heavily depends on them. If these assumptions are also commonly applied in other convergence analysis works related to Markov chains or other perspectives, I would suggest the authors highlight their necessity and reference existing related studies to make it clear.
2. The improved convergence rate with linear speedup in terms of the number of clients $N$ seems to rely on specific conditions, as highlighted in Corollary 4.9. These conditions include $\gamma \lesssim ...$, $H  \lesssim ...$ and $N \lesssim...$, this implies that the better rate seems achievable only when the number of local steps $H$ and the number of total clients $N$ are constrained within certain bounds. I think such conditions about $H$ and $N$ appear to be less prevalent in related works, such as Mangold et al. (2024b) and Hu & Huang (2023) referenced in your paper.
3. Moreover, one of the key motivation behind Scaffold is to address the issue of client drift, which occurs when a large number of local training steps cause clients with heterogeneous data distributions to converge toward their local optima rather than the global optimum. However, in this paper, the condition $H  \lesssim ...$ (which, to my understanding, is not required in the original Scaffold paper),  somewhat contradicts this motivation.

---

> ### Author Rebuttal · Authors · 2025-03-31
>
> Thank you for your thorough review of our paper and for your insightful comments.
> We are happy that you found our paper "well written, and the theoretical part is mostly clear to follow."
>
> **"I believe the theoretical results in this paper rely on some strong assumptions, such as the bounded third and fourth derivatives. [...] If these assumptions are also commonly applied in other convergence analysis works related to Markov chains or other perspectives, I would suggest the authors highlight their necessity and reference existing related studies to make it clear."**
> Thank you for allowing us to clarify.
> These assumptions are not required to establish the *convergence* results (namely Theorem 4.2 and 4.4), which hold under standard assumptions ($\mu$ strong convexity and $L$-smoothness).
> They are necessary to bound higher-order terms in subsequent results (e.g., Theorems 4.7 and 4.8): this is needed to derive our refined results.
> Higher-order derivative assumptions are standard in the literature, they are employed in the analysis of SGD in [1,2,3], using similar ideas of viewing SGD as a Markov chain.
>
>
> [1] Dieuleveut et al. (2020). Bridging the Gap between Constant Step Size Stochastic Gradient Descent and Markov Chains.
>
> [2] Allmeier et al. (2024). Computing the bias of constant-step stochastic approximation with markovian noise.
>
> [3] Sheshukova et al. (2024). Nonasymptotic analysis of stochastic gradient descent with the richardson-romberg extrapolation.
>
> **"The improved convergence rate with linear speedup in terms of the number of clients seems to rely on specific conditions, as highlighted in Corollary 4.9 ($\gamma \lesssim ...$, $H \lesssim ...$, and $N \lesssim ...$).  [...] I think such conditions about and appear to be less prevalent in related works, such as Mangold et al. (2024b) and Hu & Huang (2023)."**
> Corollary 4.9 provides the sample and communication complexity of SCAFFOLD. The bound on \\(H\\) arises from the necessity to simultaneously control errors in both the parameters and the control variates, explaining the discrepancy between our analysis and that of Mangold et al. (2024b). The bound on \\(\\gamma\\) is standard in the stochastic optimization literature, typically required for bounding the error of the final iterate. Finally, the bound on \\(N\\) emerges from the need to manage higher-order terms: convergence still holds for values of \\(N\\) exceeding this bound, albeit without linear speed-up. If the reviewer believes it would clarify our main message, we can provide an alternative formulation of this lemma without imposing this condition.
>
>
> **"Moreover, one of the key motivation behind Scaffold is to address ...]. However, in this paper, the condition [..] contradicts this motivation."**
> A similar condition appears in the main theorem (Theorem VII) of Karimireddy et al. (2020), which requires \\(\eta_g \\gamma H L \le 1\\), with \\(\\eta_g\\) denoting the global step size. Since Karimireddy et al. mandate the global step size to satisfy \\(\\eta_g \\ge 1\\), their result implicitly enforces the condition \\(\\gamma H L \\le 1\\), matching the assumption in our paper. To our knowledge, no existing analysis of SCAFFOLD (with a deterministic communication scheme) avoids this assumption. If the reviewer is aware of any such analysis, we would gladly include it in our discussion.
>
> **"1. The authors discuss the non-vanishing bias of Scaffold in Section 5 and claim that Scaffold can eliminate heterogeneity bias...."**
> Our analysis of SCAFFOLD's bias facilitates a direct comparison with that of FedAvg, as characterized, for instance, by Mangold et al. (2024b). Specifically, we demonstrate that the bias can be decomposed into heterogeneity and stochasticity components. Regarding the empirical validation of this finding, we note that related experimental evidence has already been provided, for example, by Karimireddy et al. (2020). Consequently, our own experiments primarily focus on investigating the linear speed-up phenomenon.
>
>  **"2. The authors claim that Scaffold achieves linear speedup based on the results in Figure 1. [...]."**
> We respectfully maintain that our result does represent a notable advancement: demonstrating that SCAFFOLD achieves linear speed-up has been an open problem in the literature for several years. Although linear speed-up is well-established for FedAvg, extending this result to SCAFFOLD is considerably more challenging due to noise propagation through the control variates into all parameters. To establish linear speed-up in this context, one must carefully track covariances among all pairs of control variates and demonstrate that these covariances scale as \\(1/N\\), a task that is technically intricate and previously unresolved.

---

> > ### Comment · Reviewer_QhCL · 2025-04-07
> >
> > Thank you to the authors for the rebuttal. Most of my concerns have been addressed. Sorry for my late reply, as I originally replied to the “Official Comment” button.
> >
> > While I still find the condition of $N \lesssim...$ a little bit unclear (I understand the necessity of $\gamma \lesssim ...$, $H  \lesssim ...$ as many papers need such condition). Could the authors please elaborate further on how “the bound on $N$ emerges from the need to manage higher-order terms”?
> >
> > I would be glad to raise my score if this concern is clarified.

---

> > > ### Author Response · Authors · 2025-04-08
> > >
> > > Thank you very much for the follow-up. Regarding the condition on $N$, it can be removed, which results in the following modification of Corollary 4.9.
> > > In this version of the Corollary, we do not impose any condition on $N$. Instead, the number of gradients computed by each client scales in $\tfrac{\sigma_\star^2 }{\mu^2 \epsilon^2}
> > > \max(\tfrac{1}{N}, \tfrac{Q^{2/3} \epsilon^{2/3}}{\mu}, \tfrac{Q \epsilon}{L^{1/2}\epsilon^{1/2}})$ (up to logarithmic terms), which decreases in $N$ while $\max(\tfrac{Q^{2/3} \epsilon^{2/3}}{\mu}, \tfrac{Q \epsilon}{L^{1/2}\epsilon^{1/2}}) \le 1/N$.
> > > We will replace the current Corollary 4.9 by this updated version.
> > >
> > > ------
> > >
> > > What we meant by "the bound on N emerges from the need to manage higher-order terms" is that linear speed-up holds until this given threshold on $N$ (but this does not prevent convergence of the algorithm). This is due to the need to bound the terms in $\gamma$, $\gamma^{3/2}$ and $\gamma^3$ in Theorem 4.8's result by $\epsilon^2$, which requires that $\gamma \lesssim{} \min(\frac{N \mu \epsilon^2}{\sigma_\star^2}, \frac{\epsilon^{4/3} \mu^{5/3}}{Q^{2/3} \sigma_\star^2}, \frac{{L}^{1/2} \mu^{3/2} \epsilon}{  Q \sigma_\star^2})$.
> > >
> > > Thank you again for your insightful remarks, which will greatly help to improve the clarity of our paper.
> > >
> > > -----
> > >
> > > **Corollary 4.9.** Let $\epsilon > 0$.
> > > Under Theorem 4.8's assumptions, we can set $\gamma \lesssim{} \min(\frac{N \mu \epsilon^2}{\sigma_\star^2}, \frac{\epsilon^{4/3} \mu^{5/3}}{Q^{2/3} \sigma_\star^2}, \frac{{L}^{1/2} \mu^{3/2} \epsilon}{  Q \sigma_\star^2})$ and $H
> > > \lesssim \frac{\sigma_\star^2 \min( 1, {\mu}/{\zeta})}{\mu L \epsilon^2} \max(\frac{1}{N}, \frac{Q^{2/3} \epsilon^{2/3}}{\mu}, \frac{Q \epsilon}{L^{1/2}\epsilon^{1/2}})$.
> > > Then, Scaffold guarantees $\mathbb{E}[ \|\| \theta^{T} - \theta^\star \|\|^2] \le \epsilon^2$ for $T \gtrsim
> > > \frac{L}{\mu} \max(1, \zeta/\mu)
> > > \log\left(
> > > \frac{\|\| \theta^{0} - \theta^\star \|\|^2 + \zeta^2 / L^2}{\epsilon^2}
> > > \right)$, and the number of stochastic gradients computed by each client is
> > > $$\\# \text{grad per client}
> > > \lesssim \tfrac{\sigma_\star^2 }{\mu^2 \epsilon^2}
> > > \max(\tfrac{1}{N}, \tfrac{Q^{2/3} \epsilon^{2/3}}{\mu}, \tfrac{Q \epsilon}{L^{1/2}\epsilon^{1/2}})
> > > \log\left(
> > > \tfrac{\|\| \theta^{0} - \theta^\star \|\|^2 + \zeta^2 / L^2}{\epsilon^2}
> > > \right) \enspace.$$
> > >
> > > **Proof.**
> > > The condition on $\gamma$ stems from the conditions $\frac{\gamma}{N \mu} \sigma_\star^2 \le \epsilon^2$ and $\frac{\gamma^{3/2} Q}{\mu^{5/2}} \sigma_\star^3 \le \epsilon^2$, as well as $\frac{\gamma^3 H Q^2}{\mu^3} \sigma_\star^4 \le \frac{\gamma^2 Q^2}{\mu^3 L} \sigma_\star^4 \le \epsilon^2$.
> > > Then the condition on $H$ comes from $\gamma H L \lesssim 1$ and $\gamma H L \zeta_2 \lesssim \mu$.
> > >
> > > Finally, setting $T$ such that $(1 - \gamma \mu/4)^{HT} \log(\|\|\theta^0 - \theta^\star \|\|^2 + \gamma^2 H^2 \zeta_1^2 ) \le \epsilon^2$ gives the communication complexity.
> > > The sample complexity comes from the fact that each client computes $TH$ gradients.

---

### Official Review · Reviewer_muGE · 2025-03-17

**Overall Recommendation:** 4

**Summary:**

This paper studies the convergence of the SCAFFOLD algorithm under the assumptions of (a) strong convexity, (b) smoothness, (c) first-order similarity (i.e. the average norm of the difference between the gradients on each client and the avg function is bounded), (d) second-order similarity (like the former, but for Hessians), and (e) second-order smoothness (so the Hessian is Lipschitz). The original analysis of SCAFFOLD does exhibit a linear speedup, but only when using a global stepsize (i.e. not straightforward aggregation). This paper gives convergence guarantees showing that SCAFFOLD does exhibit a linear speedup in its main terms, albeit at the cost of the additional assumptions (d) and (e). The paper also derives approximate expressions for the covariates of SCAFFOLD at convergence.

After reviewing the authors' response clarifying their focus on studying SCAFFOLD with global stepsize η=1 (which aligns with standard usage in implementations), I maintain my accept recommendation.

**Claims And Evidence:**

The paper presents novel theory that provably shows SCAFFOLD does enjoy linear speedup in the number of nodes, albeit at the cost of added assumptions (in particular, the Lipschitz Hessian assumption and the second-order similarity ones). There are additional higher-order terms that do not enjoy a linear speedup, but this is common to all other analysis of both SCAFFOLD and FedAvg.

1. "In this paper, we aim to study the SCAFFOLD algorithm as it is commonly used. Thus, contrarily to (Karimireddy et al., 2020; Yang et al., 2021), we do not consider two-sided step sizes. While this yields the desired linear speed-up by dividing the local step size by √N , and increasing the global one, it essentially reduces the algorithm to mini-batch SGD, and does not give much insights on SCAFFOLD itself. Thus, we consider in Table 1 the rate of SCAFFOLD without global step size" One of the key flexibilities of using two stepsizes is to be able to interpolate between minibatch SGD and local SGD, as argued in [1]-- I don't see why removing this is any good idea. I also find it a bit strange to say "does not give much insights on SCAFFOLD itself" when the SCAFFOLD paper uses a global stepsize as an integral part of the algorithm! "in practical implementations, there is no global step size" This does not seem to be true. Looking at GitHub repos implementing SCAFFOLD, many use a global learning rate [e.g. 2, 3]; Moreover, [4] argued that server stepsizes play a crucial role in federated learning algorithms. Nevertheless, I think understanding the algorithm with global stepsize $1$ is still useful.
2. The fact that SCAFFOLD still suffers from bias due to stochastic updates is not very surprising. After all, there is no correction term applied on the basis of which stochastic gradient we are using-- the correction terms are indexed by the clients.

[1] Woodworth, Patel & Srebro (2020) Minibatch vs Local SGD for Heterogeneous Distributed Learning.
[2] https://github.com/KarhouTam/SCAFFOLD-PyTorch/blob/master/src/server/scaffold.py
[3] https://github.com/BalajiAI/Federated-Learning/blob/main/src/algorithms/SCAFFOLD/server.py
[4] Charles & Konečný (2020) On the Outsized Importance of Learning Rates in Local Update Methods.

**Essential References Not Discussed:**

N/A.

**Experimental Designs Or Analyses:**

N/A.

**Methods And Evaluation Criteria:**

This is a theoretical paper, and the included examples are toy to demonstrate the theory. However, I think the paper would benefit significantly from including some experiments on neural networks to see if we can expect the same linear speedup to hold in the nonconvex setting.

**Other Comments Or Suggestions:**

Line 1094: "Bouding" should be Bounding.

**Other Strengths And Weaknesses:**

1. (Strength) The Markov chain point-of-view is not very common in the federated learning literature, and using it here turns out to be very useful and insightful. I think this connection can be helpful to the community going forward.
2. (Strength) The two-tiered approach of first relying on a convergence guarantee with no linear speedup, then one with linear speedup, is also quite nice.
3. (Strength) The approach of studying the convergence of the algorithm by essentially studying its algorithmic stability (Lemma B.1) is also very creatively applied here. It's rare to see this in the literature.
4. (Weakness) Virtually all of the results require using local stepsizes that scale with $1/H$, this is a tiny stepsize and does not reflect practice. It also means we can not recover results in the i.i.d. regime, which require large local stepsizes.

**Questions For Authors:**

1. Given that you do assume second-order similarity, the rate you obtain for SCAFFOLD is $\frac{L}{\mu} \frac{\zeta_2}{\mu}$ when $\zeta_2 > \mu$-- given that $\zeta_2 \leq L$ necessarily, this rate is always larger than $\frac{\zeta_2^2}{\mu^2}$, which is the rate obtained by several algorithms in the second-order similarity setting (e.g. [5]). This adds yet another dimension to the comparison: sometimes, $\zeta_2^2/\mu^2$ can be even smaller than $\sqrt{L/\mu}$. I am not 100% sure if any of the papers on the second-order similarity assumption show a linear speedup; Do you think it's possible to get a rate like $\frac{\zeta_2^2}{\mu^2}$ here?
2. Can the Performance Estimation Problem [6] be used to derive an algorithm-specific lower bound for SCAFFOLD that can shed light on what the tightest rate we can expect to be?

[5] Khaled & Jin (2022) Faster Federated Optimization Under Second-Order Similarity.
[6] https://francisbach.com/computer-aided-analyses/

**Relation To Broader Scientific Literature:**

There is plenty of literature on variance-reduced methods like SCAFFOLD for reducing the heterogeneity bias in federated learning; This paper advances the analysis of SCAFFOLD and the methods here can potentially be used for other algorithms as well. The analysis in the present paper borrows a lot from [5], but I think it's sufficiently different that it is still a significant contribution.

[5] Mangold, Paul, et al. "Refined Analysis of Federated Averaging's Bias and Federated Richardson-Romberg Extrapolation." arXiv preprint arXiv:2412.01389 (2024).

**Theoretical Claims:**

I have checked some proofs in section B, particularly the proofs of Lemma B.1. to Theorem B.4. I did not check the other proofs.

---

> ### Author Rebuttal · Authors · 2025-03-31
>
> Thank you for the positive evaluation of our paper! We appreciate that you found the Markov chain point of view "very useful and insightful", and "helpful to the community going forward", and that you found our theory based on algorithmic stability "very creatively applied here" and "rare to see in the literature."
>
>
> We address your concerns below.
>
> **" “in practical implementations, there is no global step size” This does not seem to be true. Looking at GitHub repos implementing SCAFFOLD, many use a global learning rate [e.g. 2, 3]"**
> Thank you for allowing us to clarify this point. Our statement refers to the experimental section in the original SCAFFOLD paper, which states: "We always use global step-size $\eta_g = 1$".
>
> While it is true that implementations of this algorithm allow for a global step size different from one, the default setting in all experiments we found is $\eta_g = 1$.
> From the references you cite, [2] consistently uses a global step size $\eta_g = 1$ in its experiments, and [3] only provides an implementation of the algorithm without using it in specific tasks.
> As an additional example, the FLamby dataset's benchmarks [FLamby] also sets the global step size to one across various optimization tasks.
> We will add these references to the statement in the paper to make this as clear as possible.
>
> [2] https://github.com/KarhouTam/SCAFFOLD-PyTorch/blob/master/src/server/scaffold.py
>
> [3] https://github.com/BalajiAI/Federated-Learning/blob/main/src/algorithms/SCAFFOLD/server.py
>
> [FLamby] https://github.com/owkin/FLamby/tree/main/flamby
>
>
> **"One of the key flexibilities of using two stepsizes is to be able to interpolate between minibatch SGD and local SGD, as argued in [1]-- I don’t see why removing this is any good idea."**
> **"Moreover, [4] argued that server stepsizes play a crucial role in federated learning algorithms. Nevertheless, I think understanding the algorithm with global stepsize $1$ is still useful."**
> We acknowledge this point and agree that server step sizes can play an important role.
> However, as stated above, using a server step size departs from standard usage of Scaffold, motivating our choice of studying the algorithm with the global step size set to one.
> Precisely studying the impact of the choice of global/local step sizes on Scaffold (in terms of convergence rate, bias, and linear speed-up) could be an interesting extension of the analysis framework we propose here, that goes beyond the scope of the current paper.
>
> **"Virtually all of the results require using local stepsizes that scale with $1/H$, this is a tiny stepsize and does not reflect practice. It also means we can not recover results in the i.i.d. regime, which require large local stepsizes."**
> Thank you for raising this point.
> Note that the condition was already present in the original Scaffold paper.
> Removing the condition $\gamma H L \lesssim{} 1$ in the convergence of Scaffold to a stationary distribution this question is highly non-trivial.
> In fact, it is not clear that Scaffold converges to a stationary distribution for larger H: this is a very interesting open problem to solve in future work.
>
> **"1. I am not 100% sure if any of the papers on the second-order similarity assumption show a linear speedup; Do you think it’s possible to get a rate like $\zeta_2^2 / \mu^2$ here?""**
> To our knowledge, our result is the first to show that a method that corrects heterogeneity (at least without global step size) has linear speed-up. Reducing $L \zeta_2 / \mu^2$ to $\zeta_2^2 / \mu^2$ would require to remove the conditions $\gamma H L \lesssim{} 1$ which, as stated above, is a difficult problem.
>
> **"2. Can the Performance Estimation Problem [6] be used to derive an algorithm-specific lower bound for SCAFFOLD that can shed light on what the tightest rate we can expect to be?""**
> Performance estimation ideas are completely orthogonal to our work.
> To perform this kind of analyses, profound investigation is required. In particular, PEP relies on "interpolation conditions", which, to extend to the federated case, would require to choose an heterogeneity measure and adapt interpolation conditions accordingly.
> Nonetheless, they constitute an interesting perspective for future work, with the goal of achieving the best possible rate for Scaffold.

---

> > ### Comment · Reviewer_muGE · 2025-04-01
> >
> > Thanks for your response, I maintain my score.

---

### Decision · Program_Chairs · 2025-05-01

**Decision:**

Accept (poster)

**Comment:**

This paper provides a new convergence analysis for SCAFFOLD, a popular method in federated learning. The primary focus is studying SCAFFOLD with a single stepsize ("without global step size"). This corresponds to the variant of SCAFFOLD primarily used in practice and improves over the original paper, which introduced an additional stepsize for technical reasons.

The reviewers commended the approach from a Markov point of view, which might be interesting for some readers (although this approach is not novel, it is less common in the FL literature).

The discussion between authors and reviewers clarified some aspects of the paper. In particular, the key differences (and similarities) between the convergence results presented in this work vs. the original SCAFFOLD paper and the justification for the particular assumptions used in this work (discussion with Reviewer QhCL) were discussed. We ask the authors to incorporate these aspects in the final version of the paper.